# Osteocyte mitochondria regulate angiogenesis of transcortical vessels

Peng Liao [1,2,10], Long Chen[3,10], Hao Zhou[4,10], Jiong Mei[1], Ziming Chen[5,6], Bingqi Wang[1,2], Jerry Q. Feng[7], Guangyi Li[1], Sihan Tong[1,2], Jian Zhou[1,2], Siyuan Zhu[8], Yu Qian[9], Yao Zong[5,6], Weiguo Zou [1,2,3], Hao Li[1,2], Wenkan Zhang[4], Meng Yao[1,2], Yiyang Ma[1,2], Peng Ding [1,2], Yidan Pang [1,2], Chuan Gao[1,2], Jialun Mei[1,2], Senyao Zhang [1,2], Changqing Zhang [1,2] ✉, Delin Liu [1,2] ✉, Minghao Zheng [5,6] ✉ & Junjie Gao [1,2] ✉

Transcortical vessels (TCVs) provide effective communication between bone marrow vascular system and external circulation. Although osteocytes are in close contact with them, it is not clear whether osteocytes regulate the homeostasis of TCVs. Here, we show that osteocytes maintain the normal network of TCVs by transferring mitochondria to the endothelial cells of TCV. Partial ablation of osteocytes causes TCV regression. Inhibition of mitochondrial transfer by conditional knockout of *Rhot1* in osteocytes also leads to regression of the TCV network. By contrast, acquisition of osteocyte mitochondria by endothelial cells efficiently restores endothelial dysfunction. Administration of osteocyte mitochondria resultes in acceleration of the angiogenesis and healing of the cortical bone defect. Our results provide new insights into osteocyte-TCV interactions and inspire the potential application of mitochondrial therapy for bone-related diseases.

Blood vessels are widely distributed in bone and necessary for bone and bone marrow homeostasis[1]. The vascular network has been shown to play a critical role in controlling bone development[2] and wound healing[3] and provides a microenvironment for the differentiation and maturation of hematopoietic and immune cells in bone marrow[1,3]. Capillaries originate from bone marrow, traverse to cortical bone, named transcortical vessels (TCVs)[4], consist of endothelial cells lining the narrow canal inside of the cortical bone[5,6]. TCV formation relies on the sprout protrusion of endothelial cells in cortical bone canals[7–9].

TCVs connect the endosteum and periosteum and provide access that effectively promotes communication between the inner cavity and outer shell of bone[4]. It has been reported that over 80% of arterial, 59% of venous blood and several kinds of immune cells pass through the TCVs to go to the other side of the bone[4], suggesting that TCVs provide critical routes between the medullary environment and outer bone surface for hematopoietic cells and components transportation as well as bone cells and immune cell communication[5,6]. Although TCVs constitute a crucial part of the bone vascular system for bone and bone

[1]Department of Orthopaedics, Shanghai Sixth People's Hospital Affiliated to Shanghai Jiao Tong University School of Medicine, Shanghai, China. [2]Institute of Microsurgery on Extremities, Shanghai Sixth People's Hospital Affiliated to Shanghai Jiao Tong University School of Medicine, Shanghai, China. [3]State Key Laboratory of Cell Biology, Shanghai Institute of Biochemistry and Cell Biology, CAS Center for Excellence in Molecular Cell Science, Chinese Academy of Sciences, University of Chinese Academy of Sciences, Shanghai, China. [4]Department of Orthopedics, The Second Affiliated Hospital of Zhejiang University School of Medicine, Hangzhou, China. [5]Centre for Orthopaedic Research, Medical School, The University of Western Australia, Nedlands, Western Australia, Australia. [6]Perron Institute for Neurological and Translational Science, Nedlands, Western Australia, Australia. [7]Shanxi Medical University School and Hospital of Stomatology, Shanxi Province Key Laboratory of Oral Diseases Prevention and New Materials, Taiyuan, China. [8]Department of General Surgery, Shanghai Sixth People's Hospital Affiliated to Shanghai Jiao Tong University School of Medicine, Shanghai, China. [9]Department of Orthopedics, The First Affiliated Hospital of Zhejiang Chinese Medical University (Zhejiang Provincial Hospital of Chinese Medicine), Hangzhou, China. [10]These authors contributed equally: Peng Liao, Long Chen, Hao Zhou. ✉e-mail: zhangcq@sjtu.edu.cn; liudelin_doc@126.com; minghao.zheng@uwa.edu.au; colingjj@163.com

marrow homeostasis, the regulatory factors that determine TCV formation and maintenance are still unknown.

In cortical bone, osteocytes comprise 90–95% of all bone cells and are extensively embedded in the cortical bone matrix[10]. Once the premature osteocytes are embedded in cortical bone, they undergo a dramatic transformation from a polygonal cell to a cell extending abundant dendrites toward the mineralizing front, which is followed by the appearance of dendrites extending to either the vascular space or bone surface[10]. Dendrites of osteocytes form an interconnected network between osteocytes and provides a route for intercellular signal transmission and organelles communication[10,11]. Considering that osteocytes are the dominant cells embedded in mineralized cortical bone and play important roles in regulating bone homeostasis, it is rational that osteocytes might act as the critical mediator for the ingrowth and maintenance of TCVs. Although morphological evidence of the association between osteocytes and endothelial cells in cortical bone has been demonstrated[4,12], whether osteocytes, the major cell components of cortical bone, could mediate the vascularization process of TCVs is still unknown.

Here, we reveal a crucial role of osteocytes in TCV vascularization. Osteocytes maintain a normal TCV network by transferring mitochondria to endothelial cells in cortical bone. The acquisition of osteocyte mitochondria maintains normal endothelial functions by alleviating oxidative stress, promoting cell proliferation, advancing tube formation, and restoring the migration capability of endothelial cells. The pro-angiogenic effect of osteocyte mitochondria on endothelial cells can be mimicked by supplementation with D-sphingosine and abolished after the inhibition of osteocytes sphingosine kinase 1 (SPHK1). These findings revealed a unique mechanism involved in the vascular homeostasis of cortical bone and potentially open up a therapeutic approach for bone diseases associated with vascular damage.

## Results

### Osteocytes connect to endothelial cells of TCV

To observe the structural association between osteocytes and TCVs, we employed scanning electron microscopy (SEM), and revealed that osteocytes extend dendrites with inflated endfeet-like structures to the TCV (Fig. 1a). In addition, transmission electron microscopy (TEM) imaging of mouse femur cortical bone also showed the dendritic connection between lacunae-chambered osteocytes and endothelial cells in the vessel canals of mineralized bone (Fig. 1b). Next, to further investigate the interactions between osteocytes and endothelial cells of TCV, we used an Evans Blue injection assay (Fig. 1c) and immunostaining of the endothelial cell marker CD31 (Fig. 1d) to visualize blood vessels in cortical bone. Confocal imaging of coronal sections in mouse femurs showed a large number of dendrites stretching from osteocytes to the nearby the TCV and in close contact with endothelial cells in cortical bone (Fig. 1c, d). Quantitative analysis from CD31-labeled confocal images (Fig. 1d) showed abundant dendrites of osteocytes around TCVs (Fig. 1e, f). In detail, 60.27% of the total dendrites were distributed on the vessel nearside (Fig. 1f, g), among which 74.20% were in contact with endothelial cells (Fig. 1f, h), establishing complex dendritic contact between osteocytes and endothelial cells. We next quantified the angular deflection of the osteocyte long axis in femur cortical bone, and showed that osteocytes adjacent to TCVs presented greater axis deflection to align with TCVs rather than the bone axis, than osteocytes that were distant to TCVs (Fig. 1i). These results suggested that osteocytes are closely connected to endothelial cells in cortical bone with inflated endfeet abutting on the TCV.

### Osteocytes regulate the vascularization of TCVs

To investigate the effect of osteocytes on TCVs, we employed a *Dmp1*[Cre] transgenic mouse lineage to specifically target osteocytes. By crossing with the diphtheria toxin subunit A (*DTA*) mouse lineage[13], we generated heterozygous *Dmp1*[Cre]-*DTA*[ki/wt] mice in which osteocytes were partially ablated[14]. Partial ablation of osteocytes was evidenced by the immunofluorescence staining of cortical bone. Quantification analysis showed that the absolute number of osteocytes were significantly decreased in *Dmp1*[Cre]-*DTA*[ki/wt] mice in comparison with control *DTA*[ki/wt] mice. (Supplementary Fig. 1a, b).

Next, we investigated the alteration of TCVs after ablating the osteocytes in *Dmp1*[Cre]-*DTA*[ki/wt] mice. Considering that the high percentage of canals in cortical bone are occupied by vessels, high-resolution micro computed tomography (μCT) was utilized to visualize the perpendicular canal system to the long axis of cortical bone to indicate the morphology of the TCV system[8,15,16]. To examine the morphological changes of TCVs after the absence of osteocytes, high-resolution μCT scanning and analysis were utilized to compare the morphological characteristics of canals within femur cortical bone between *DTA*[ki/wt] and *Dmp1*[Cre]-*DTA*[ki/wt] mice. The results suggested that the TCV morphology in *Dmp1*[Cre]-*DTA*[ki/wt] mice appears to be very crude with noncontinuous straight trackers that lack interlacing, whereas TCVs in the control *DTA*[ki/wt] mice were more complicated with continuous intersecting vertical and horizontal trackers to form a blood vessel network (Fig. 2a). Next, immunofluorescence staining of the endothelial cell marker CD31 and confocal imaging were employed to validate the changes in TCV morphology in mouse femurs after osteocyte removal. The results showed a significant decrease in osteocyte number near the blood vessels (Fig. 2b, c). In control mice (*DTA*[ki/wt]) femurs, CD31-labeled TCVs passed through cortical bone in a divaricate manner with complex directionality, whereas TCVs in *Dmp1*[Cre]-*DTA*[ki/wt] mice showed a disrupted manner with insufficient interconnection (Fig. 2b). Quantitative comparisons between the control and *Dmp1*[Cre]-*DTA*[ki/wt] mice showed a decreased number of vascular branches in *Dmp1*[Cre]-*DTA*[ki/wt] mice (Fig. 2d). These data demonstrated that absence of osteocytes led to a significant alteration in TCV network.

As it has been reported that osteoclastic activity plays a role in mediating TCVs formation[4], to answer if osteocyte ablation induced TCV regression is mediated by the induction of osteoclast activities in *Dmp1*[Cre]-*DTA*[ki/wt] mice. We performed zoledronic acid injection to inhibit the activities of osteoclasts in 4-week-old *Dmp1*[Cre]-*DTA*[ki/wt] mice. Lower limbs were collected for osteoclast staining and TCVs assessment. Using tartrate-resistant acid phosphatase (TRAP) and alkaline phosphatase (ALP) staining on Dmp1[Cre]-*DTA*[ki/wt] mice cortical bone, we showed that ablation of osteocytes in cortical bone causes a significant decrease of osteoblast number (Supplementary Fig. 1c, d) and increase of osteoclast number (Supplementary Fig. 1e, f). Administration of zoledronic acid for 2 weeks revealed decrease of osteoclast number in cortical bone in comparison with the PBS injection mice (Supplementary Fig. 1g, h). The CD31 immunofluorescence staining showed that zoledronic acid does not alter the TCVs number nor the TCVs branches points as compared to the control (Supplementary Fig. 1i, j). These results indicated that the decreased TCVs when osteocytes were ablated is not due to the induction of osteoclast activities. Together, these results showed that osteocytes regulate the vascularization of TCVs directly in the manner that is independent of osteoclasts.

To better understand the role of osteocytes in regulating TCVs, bulk RNA sequencing (RNA-seq) of mouse femur cortical bone was utilized. Gene Ontology (GO) analysis and gene set enrichment analysis (GSEA) showed that angiogenesis was one of the top two downregulated terms in *Dmp1*[Cre]-*DTA*[ki/wt] mice compared with control *DTA*[ki/wt] mice (Fig. 2e, f). In addition, by searching all the significantly downregulated terms in the GO analysis data, we also found a broad range of significantly decreased biological process terms related to blood vessel morphology and development in *Dmp1*[Cre]-*DTA*[ki/wt] mice (Fig. 2g). When researching angiogenesis-related genes from a reported study[17,18], we found that *Vegfc*[19], *Slit3*[20], *Notch3*[21], and *Notch4*[22] overlapped with the differentially expressed downregulated genes in

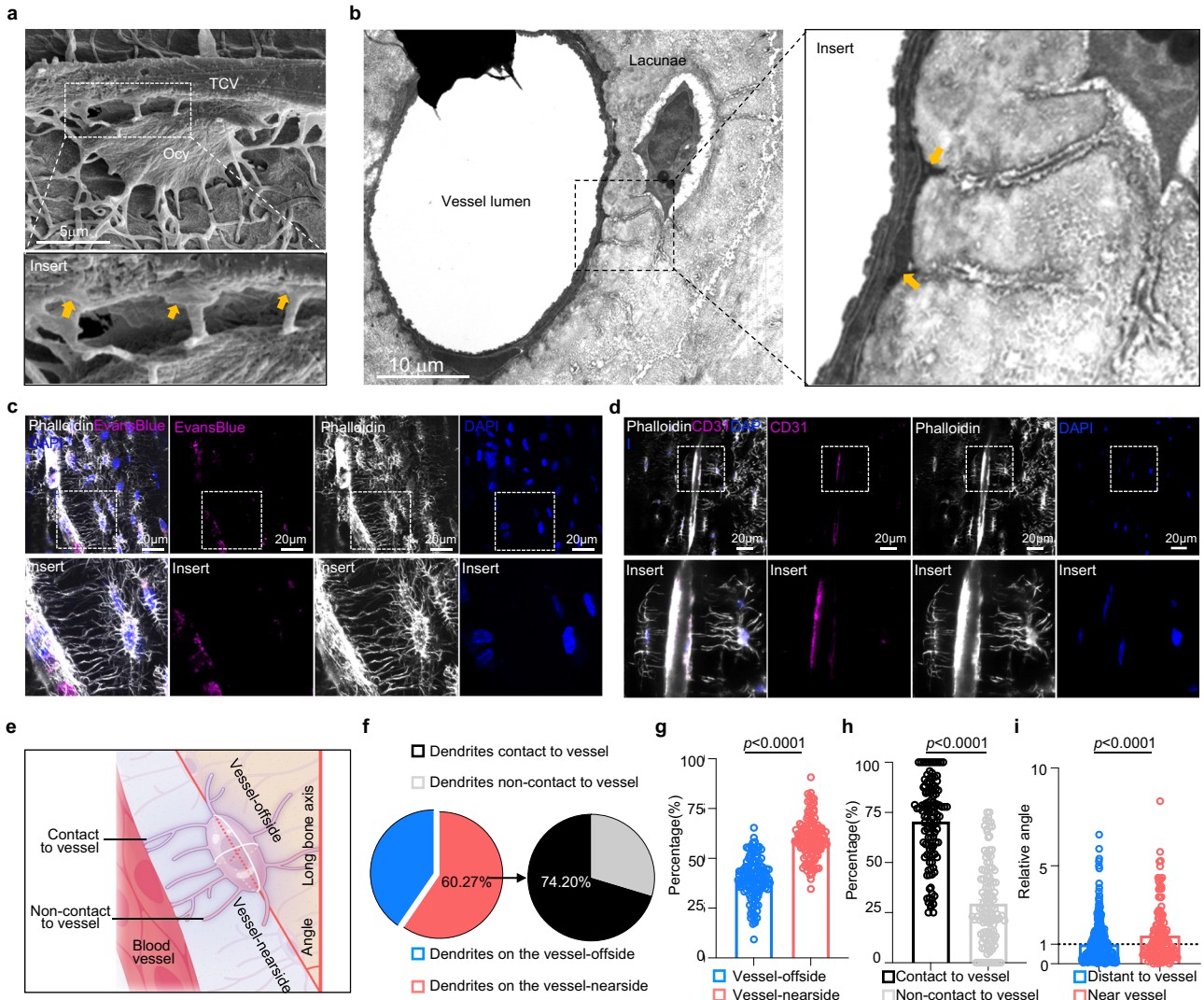

**Fig. 1 | Osteocytes connect to endothelial cells via dendrites. a** Representative SEM images from 2-month-old male WT mouse femur cortical bone revealed abundant dendritic (yellow arrow) connections from osteocytes with inflated endfeet abutting TCVs. Scale bars, 5 μm. **b** Representative TEM images from 2-month-old male WT mice showing the dendrite connections (yellow arrow) between lacunae-chambered osteocytes and endothelial cells in the vessel canals. Scale bars, 10 μm. **c** Representative confocal images of femur cortical bone from three 4-week-old male WT mice injected with Evans Blue intraperitoneally showed multiple dendrites extending from osteocytes to blood vessels. **d** Representative confocal images of femur cortical bone from 4-week-old male WT mice showed the dendrite connection between osteocytes and endothelial cells. Scale bars, 20 μm. **e** Schematic diagram of osteocytes beside TCVs for further analysis and quantification. This figure was created by P.L. and cartonized by Ms. Lina Cao. **f** Pie diagram of the dendrites of osteocytes beside TCV of **d** ($n = 119$ osteocytes examined over 5 independent mice). **g** Percentage of the dendrites on the vessel-offside or vessel-nearside among all the dendrites of every osteocyte beside the blood vessel. ($n = 119$ osteocytes examined over 5 independent mice). **h** Quantitative percentage of dendrites in contact with vessels or in noncontact with vessels among all the vessel-nearside dendrites as described in **g**. **i** The relative angle between the long bone axis and the long axis of the osteocyte soma distant from TCVs (no dendrite contacts with TCVs) or near TCVs, as shown in **d** (142 osteocytes adjacent to TCVs and 368 osteocytes distant to TCVs were examined from independent 5 mice). Data were presented as the means ± SEMs; Significance was calculated using unpaired t test with two-tailed $P$ value **g**, **h**, **i**. Source data are provided as a Source Data file.

our cortical bone bulk RNA-seq database of $Dmp1^{Cre}$-$DTA^{ki/wt}$ mice (Fig. 2h, Supplementary Table 1a). Furthermore, qPCR analysis of cortical bone validated that these four genes were significantly decreased in $Dmp1^{Cre}$-$DTA^{ki/wt}$ mouse femurs (Fig. 2i). To investigate the source of four genes, we performed qPCR analysis on the MLO-Y4 osteocytes cell line and bEnd.3 endothelial cell line. The results showed the endothelial cells and osteocytes express $Slit3$, $Notch3$ and $Vegfc$, while the expression of $Notch4$ is relatively low in osteocytes (Supplementary Fig. 1k). Next, we wondered whether osteocyte regulating endothelial cells angiogenesis is linked to these overlapped genes. We transfected MLO-Y4 cells with small interfering RNA (siRNA) of $Vegfc$, $Slit3$, $Notch3$, $Notch4$ to knock down expression of target genes individually. The efficiency of knocking down expression of target genes

were validated by qPCR (Supplementary Fig. 1l). Then, we tested the role of MLO-Y4 cells conditioned media on bEnd.3 tube formation and wound healing assay (Supplementary Fig. 1m, o). The result showed that the conditional media obtained from the MLO-Y4 cell lines with siRNA of $Vegfc$, $Slit3$, $Notch3$, $Notch4$ do not significantly alter the tube formation and migration ability of bEnd.3 cells (Supplementary Fig. 1n, p), suggesting the weak link between $Vegfc$, $Slit3$, $Notch3$, $Notch4$ and the direct regulation of osteocytes on angiogenesis of endothelial cells.

## TCV endothelial cells acquire mitochondria from osteocytes

Intercellular mitochondrial transfer has been frequently realized as an important event for tissue development and revitalization[23]. Our

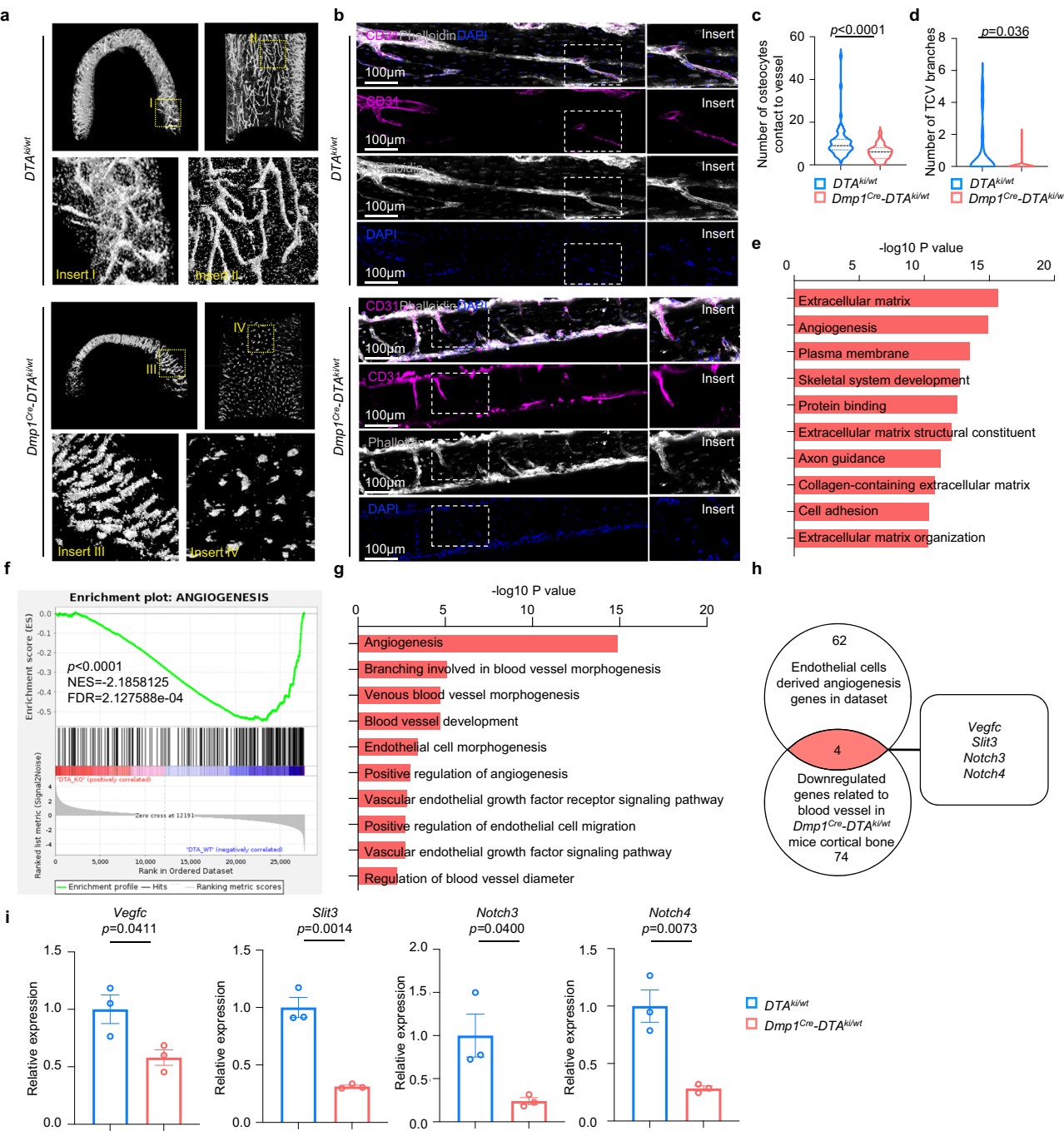

**Fig. 2 | Osteocytes regulate TCVs and angiogenesis. a** Representative images of high-resolution μCT (1 μm resolution) on 6-week-old male mouse femurs presented the complicated canal network in *DTA^ki/wt* control mouse cortical bone and crude canal system in *Dmp1^Cre-DTA^ki/wt* mouse cortical bone. **b** Representative confocal images of CD31-labeled blood vessels and phalloidin-labeled osteocytes from 6-week-old male *DTA^ki/wt* and *Dmp1^Cre-DTA^ki/wt* mouse femur cortical bone. Scale bars, 100 μm. **c, d** Quantitative analysis of the number of osteocytes in contact with blood vessels **c** and the number of TCV branches **d** from 6-week-old male *DTA^ki/wt* and *Dmp1^Cre-DTA^ki/wt* mouse femur cortical bone as shown in **b** (Data were quantified from 5 biologically independent samples). **e** Top 10 downregulated GO terms in 4-week-old *Dmp1^Cre-DTA^ki/wt* mouse femur cortical bone compared with *DTA^ki/wt*

control mice. **f** GSEA of angiogenesis (*P* < 0.0001, NES = −2.1858) in femur cortical bone of 4-week-old *Dmp1^Cre-DTA^ki/wt* mice compared with *DTA^ki/wt* control mice. **g** Selected significantly downregulated blood vessel-related GO terms in *Dmp1^Cre-DTA^ki/wt* mice. **h** Venn diagram of overlapping genes from reported endothelial cell-derived angiogenesis genes and downregulated genes related to blood vessels in *Dmp1^Cre-DTA^ki/wt* mice. **i** RT-qPCR analysis of the overlapping genes in **h** in cortical bone of 4-week-*old DTA^ki/wt* and *Dmp1^Cre-DTA^ki/wt* mouse femurs (*n* = 3 biologically independent samples). Data were presented as the means ± SEMs; Significance was calculated using unpaired t test with two-tailed *P* value **c, d, i** or hypergeometric test **e, g** or permutation test **f**. Source data are provided as a Source Data file.

previous study showed that mitochondria are dynamically transferred between osteocytes within their dendritic network[11]. Considering the pivotal role of mitochondria in endothelial angiogenesis[24] and the close association between osteocyte dendrites and TCV endothelial cells (Fig. 1a−d), we hypothesized that endothelial cells in cortical bone

require mitochondria from surrounding osteocytes to promote TCV vascularization. We first visualized the mitochondrial transfer activity of osteocytes in vitro by transfecting the fluorescent protein Dendra2 into the mitochondria of the osteocyte-like cell line MLO-Y4 (MLO-Y4^Mito-Dendra2). A 2D co-culture system of MLO-Y4^Mito-Dendra2 cells and the

mouse endothelial cell line bEnd.3 labeled with CellMask was then established, with the two types of cells number ratio at 1:1 (Fig. 3a). After co-culturing for 24 hours, confocal imaging results proved the acquisition of Dendra2-labeled mitochondria in bEnd.3 endothelial cells, indicating that the osteocyte-like MLO-Y4 cells transfer mitochondria to endothelial cells (Fig. 3b).

To further explore whether the percentage of endothelial cells that received mitochondria would increase with prolonged co-culture time and higher number ratio of donor/recipient cells, bEnd.3 cells were cocultured with MLO-Y4$^{Mito-Dendra2}$ for 24 h, 48 h, and 72 h and 7 days in vitro with the number ratio of MLO-Y4$^{Mito-Dendra2}$/bEnd.3 at 1:1 and 3:1, respectively. The flow cytometry analysis of the co-culture

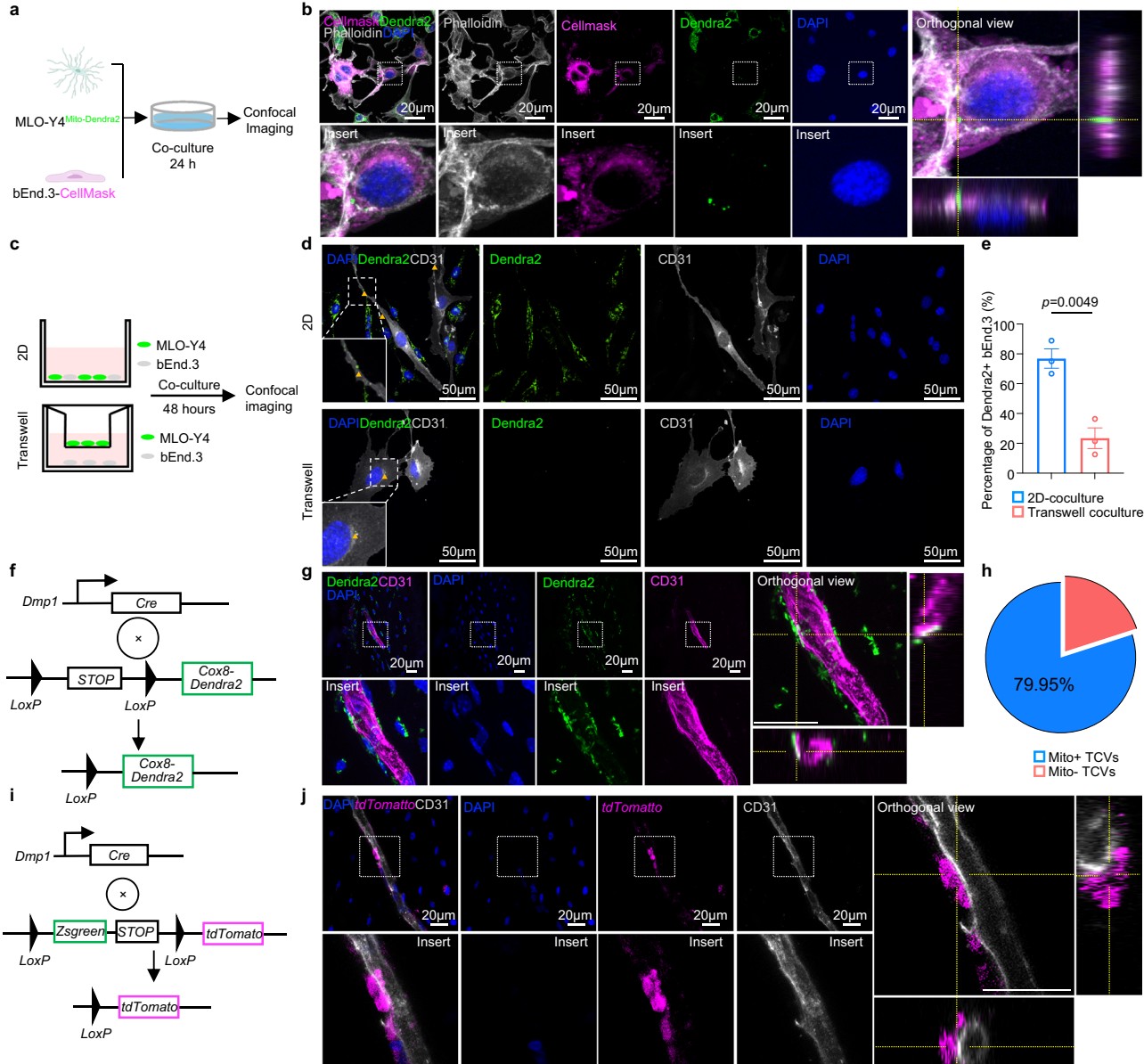

**Fig. 3 | Mitochondrial transfer from osteocytes to endothelial cells. a** Flowchart of 2D coculture of MLO-Y4$^{Mito-Dendra2}$ and CellMask-labeled bEnd.3 endothelial cells. This figure was created by P.L. and D.L.L. and cartonized by Mr. Zihao Li. **b** Representative confocal images of CellMask-labeled bEnd.3 endothelial cells with orthogonal view show the acquisition of mitochondria derived from MLO-Y4$^{Mito-Dendra2}$ after coculture. Scale bars, 20 μm. **c** Schematic diagram for 2d-coculture and transwell coculture system of bEnd.3 cells and MLO-Y4 cells with number ratio at 3:1 (MLO-Y4: bEnd.3) for 48 hours. **d, e** Representative confocal images **d** and quantitative result **e** of bEnd.3 cells acquired with Mito-Dendra2 fluorescence as shown in **c**, scale bars, 50 μm, yellow arrows represent transferred mitochondria (*n* = 3 biologically independent samples). **f** Schematic diagram for the generation of the *Dmp1*$^{Cre}$-*Cox8*$^{Dendra2}$ mouse line. When crossed to the *Dmp1*$^{Cre}$ mouse line, the termination cassette (STOP symbol) of *Cox8*$^{Dendra2}$ mice was removed to generate the *Dmp1*$^{Cre}$-*Cox8*$^{Dendra2}$ mouse line with osteocyte-specific labeling of mitochondria by Dendra2 green fluorescence. **g** Confocal images of

femur cortical bone from a 4-week-old male *Dmp1*$^{Cre}$-*Cox8*$^{Dendra2}$ mouse show the translocation of osteocyte-derived mitochondria (Dendra2) in CD31-labeled endothelial cells. Scale bars, 20 μm. **h** Quantitative result of TCVs that acquired with Mito-Dendra2 fluorescence in femur cortical bone from 4-6-week-old *Dmp1*$^{Cre}$-*Cox8a*$^{Dendra2}$ (*n* = 4 biologically independent samples). **i** Schematic diagram for the generation of *Dmp1*$^{Cre}$-*mGmT* mice. When crossed to the *Dmp1*$^{Cre}$ mouse line, the *Loxp*-flanked *Zsgreen* cassette will be removed to produce the *Dmp1*$^{Cre}$-*mGmT* mouse strain, in which the *Dmp1*$^+$ cells will be labeled by tdTomato fluorescent protein while *Dmp1* cells will express Zsgreen fluorescence protein. **j** Confocal images of femur cortical bone collected from a 4-week-old male *Dmp1*$^{Cre}$-*mGmT* mouse revealed the adjacent but noncolocalized relationship between CD31-labeled endothelial cells and tdTomato-labeled *Dmp1*$^+$ cells. Scale bars, 20 μm. Data were presented as the means ± SEMs. Significance was calculated using unpaired t test with two-tailed *P* value **e**. Source data are provided as a Source Data file.

system revealed that the fraction of endothelial cells that received the mitochondrial increased with extended duration of co-culture time and higher number ratio of donor/recipient cells (Supplementary Fig. 2a–d). The peak rate of 10% mitochondria-acquired endothelial cells occurs at 3:1 and 72 hours of co-culture (Supplementary Fig. 2b, c). These data supported transfer of mitochondria is time and donor-recipient number ratio dependent. Moreover, we investigated the routine of MLO-Y4 osteocytes transferring mitochondria to bEnd.3 endothelial cells. We used transwell coculture system to separate bEnd.3 cells and MLO-Y4 cells, and measured the transfer of mitochondria without dendrites contact (Fig. 3c). The confocal image analysis revealed that mitochondria transfer may also occur through non-contact transferring in the local microenvironment (Fig. 3d, e).

Next, we evaluated the proportion and characteristics of mitochondria by confocal on bEnd.3 cells following a 48-h coculture with MLO-Y4$^{Mito-Dendra2}$ cells (Supplementary Fig. 3a). We showed proportion of transferred mitochondria is a relatively smaller fraction of the total mitochondrial population (host mitochondria) within the recipient cell, approximately 3.16% at 48 h coculture (Supplementary Fig. 3b). Characterization of the transferred mitochondria showed that they exhibited a smaller length compared to the host mitochondria, while their width remained similar (Supplementary Fig. 3c). Specifically, the average width of the transferred mitochondria, measuring 1.23 μm, resembled that of the host mitochondria (1.04 μm). Conversely, the average length of the transferred mitochondria, at 1.46 μm, was smaller than that of the host mitochondria (2.09 μm). The sphericity of the transferred mitochondria, the average ratio of length to width was also calculated, yielding a value of 1.22. This value is notably smaller than the host mitochondria's ratio of 2.09, thereby suggesting that the transferred mitochondria possess a more spherical morphology (Supplementary Fig. 3c).

To address whether endothelial cells acquire mitochondria from osteocytes in vivo, the $Cox8^{Dendra2}$ mouse lineage was employed to establish $Dmp1^{Cre}$-$Cox8^{Dendra2}$ mice, in which the mitochondria of osteocytes were specifically labeled by the green fluorescent protein Dendra2 (Fig. 3f). Confocal imaging of $Cox8^{Dendra2}$ mice and $Dmp1^{Cre}$-$Cox8^{Dendra2}$ cortical bone demonstrated the specificity of the signal (Supplementary Fig. 3d) and revealed the co-localization of Dendra2-labeled osteocyte mitochondria in CD31-marked endothelial cells (Fig. 3g). Quantification of cortical bones revealed that 79.95% ± 17.27% (Mean ± SD) of TCV fractions contain mitochondria from osteocytes (Fig. 3h). In addition, to examine whether there is nonspecific expression of $Dmp1$ in endothelial cells, $Dmp1^{Cre}$-$mGmT$ mice were employed, in which cells expressing $Dmp1$ were specifically labeled by tdTomato fluorescence (Fig. 3i). Confocal imaging of $Dmp1^{Cre}$-$mGmT$ mouse femurs revealed that $Dmp1$-positive cells contacted but did not colocalize with endothelial cells in cortical bone (Fig. 3j). These results demonstrated that osteocytes constantly transfer mitochondria to endothelial cells of TCV.

To examine specificity of mitochondria transferred from osteocytes to endothelial cells, we investigate the presence of osteocyte mitochondria in liver tissue as liver-bone axis has been shown to play a curial role in both of organs[25,26]. Confocal microscopy (Supplementary Fig. 3d) shows that no Dendra2 signal captured in liver blood vessels. Moreover, as megakaryocytes and platelets contain abundant of mitochondria and the latter circulating in blood vessel, there may be mitochondrial exchange from megakaryocytes to endothelial cells in TCVs. We thus employed megakaryocytes specific Cre promoter, $PF4$[27] to examine TCV of cortical bone in $PF4^{Cre}$-$Cox8^{Dendra2}$ mice. The results shows that there is no Dendra2 signals in TCVs and osteocytes, suggesting that mitochondria transfer from osteocytes to endothelial cells is a specific event in cortical bone (Supplementary Fig. 3d).

## Mitochondrial transfer from osteocytes to endothelial cells regulates TCV vascularization

The process of mitochondrial motility involves a large number of proteins, in which MIRO1, a critical protein of the mitochondrial transport machinery, acts as a necessary element for mitochondrial movement[28,29]. We thus knocked down the expression of the MIRO1-encoding gene $Rhot1$ in MLO-Y4$^{Mito-Dendra2}$ cells ($Rhot1^{KD}$-MLO-Y4$^{Mito-Dendra2}$) (Supplementary Fig. 4a, b) to examine the role of MIRO1 in mitochondrial transfer. Confocal imaging of CD31-labeled bEnd.3 cells cocultured with the negative control MLO-Y4$^{Mito-Dendra2}$ (NC-MLO-Y4$^{Mito-Dendra2}$) cells or $Rhot1^{KD}$-MLO-Y4$^{Mito-Dendra2}$ cells showed a tendency toward decreased mitochondrial transfer from $Rhot1^{KD}$-MLO-Y4$^{Mito-Dendra2}$ cells to endothelial cells (Fig. 4a). This tendency was demonstrated by flow cytometry analysis on 2D coculture system for 24 hours. The results showed the decreased efficiency of mitochondrial acquisition in bEnd.3 endothelial cells after coculture with $Rhot^{KD}$-MLO-Y4$^{Mito-Dendra2}$ cells compared to those cocultured with NC-MLO-Y4$^{Mito-Dendra2}$ cells (Fig. 4b, c, Supplementary Fig. 4c).

Next, to examine the role of MIRO1 in mediating mitochondrial transfer in vivo, the $Dmp1^{Cre}$-$Cox8^{Dendra2}$-$Rhot1^{fl/fl}$ mouse lineage was established, in which $Rhot1$ was specifically deleted in Mito$^{Dendra2}$-labeled osteocytes (Fig. 4d). Confocal imaging and quantification of mouse femur cortical bone revealed a significantly decreased transferred mitochondria number in each TCV of $Dmp1^{Cre}$-$Cox8^{Dendra2}$-$Rhot1^{fl/fl}$ mice compared to those of $Dmp1^{Cre}$-$Cox8^{Dendra2}$ control mice (Fig. 4e, f). Moreover, the ratio of transferred mitochondria area to TCVs area was decreased in $Dmp1^{Cre}$-$Cox8^{Dendra2}$-$Rhot1^{fl/fl}$ mice compared with $Dmp1^{Cre}$-$Cox8^{Dendra2}$ mice (Supplementary Fig. 4d). These results implied the critical effect of MIRO1 in mediating mitochondrial transfer from osteocytes to endothelial cells.

To investigate the role of mitochondrial transfer from osteocytes to endothelial cells in TCV vascularization, we employed $Dmp1^{Cre}$-$Rhot1^{fl/fl}$ mice (Fig. 4g). Confocal imaging of CD31-labeled endothelial cells in mouse femur cortical bone showed a concession of TCVs in $Dmp1^{Cre}$-$Rhot1^{fl/fl}$ mice compared to $Rhot1^{fl/fl}$ control mice, as evidenced by the decreased blood vessel branches (Fig. 4h, i). Similarly, the high-resolution μCT results of canals in mouse femur cortical bone indicated that the TCV morphology of $Dmp1^{Cre}$-$Rhot1^{fl/fl}$ mice was immature with noncontinuous straight trackers that lacked interlacing canals compared to $Rhot1^{fl/fl}$ control mice (Fig. 4j). The high-resolution μCT results of perpendicular canals in cortical bone revealed a comparable feature of TCV morphological change between $DTA^{ki/wt}$ and $Dmp1^{Cre}$-$DTA^{ki/wt}$ mice.

While these data supported that osteocyte mitochondria plays a role in TCVs formation, it is not clear if the alteration of TCVs is directly or indirectly due to the depletion of $Rhot1$ in osteocytes. To answer this question, we firstly investigated the effect of $Rhot1$ depletion of osteocytes on osteoblasts, osteoclasts and bone marrow. Histological staining and flow cytometry analysis were performed on 6-week-old control $Rhot1^{fl/fl}$ mice and $Dmp1^{Cre}$-$Rhot1^{fl/fl}$. ALP staining showed that number of osteoblasts was unchanged in $Dmp1^{Cre}$-$Rhot1^{fl/fl}$ compared with $Rhot1^{fl/fl}$ (Supplementary Fig. 5a, b). TRAP staining showed that knocking out $Rhot1$ in osteocytes did not change number of osteoclasts (Supplementary Fig. 5c, d). Flow cytometry analysis of bone marrow cells from in $Rhot1^{fl/fl}$ and $Dmp1^{Cre}$-$Rhot1^{fl/fl}$, showed that no significant changes were observed in both lymphoid cells (B cells, CD4$^+$ T cells, CD8$^+$ T cells, neutrophils, monocytes, other myeloid cells) (Supplementary Fig. 6a, b), and hematopoietic cells (long term-hematopoietic stem cell (LT-HSC), short term-hematopoietic stem cell (ST-HSC), total granulocyte-macrophage progenitor (GMP), common myeloid progenitor (CMP), megakaryocyte-erythrocyte progenitor (MEP), multipotent progenitors (MPP), common lymphoid progenitor (CLP)) (Supplementary Fig. 6c, d).

We then investigate the alteration of osteocytes in 6-week-old $Dmp1^{Cre}$-$Rhot1^{fl/fl}$ mice. TUNEL assay of femur cortical bone showed

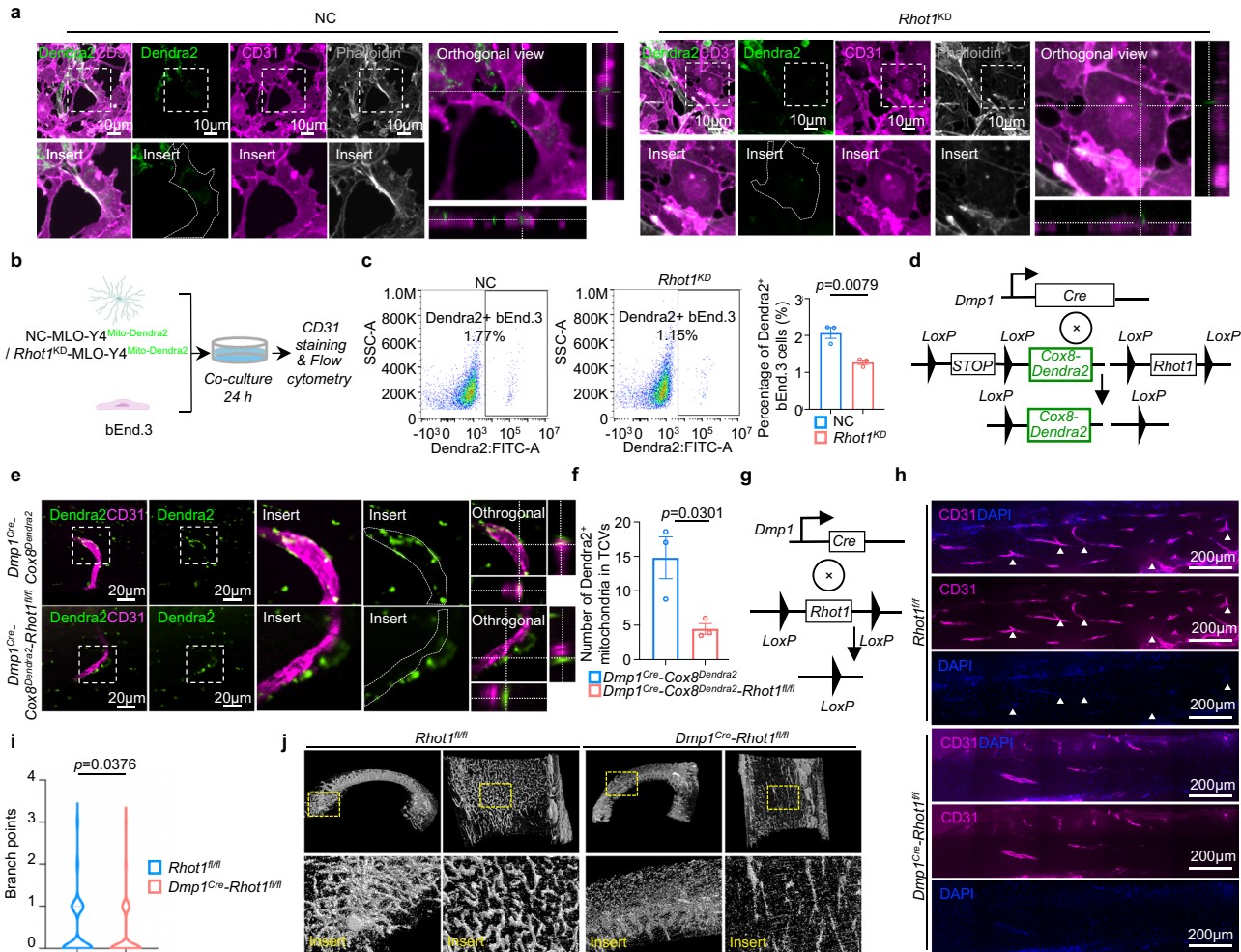

**Fig. 4 | Impaired mitochondrial transfer by deleting *Rhot1* in osteocytes leads to TCV regression. a** Confocal images show the decreased transfer of mitochondria from *Rhot1*^KD^-MLO-Y4^Mito-Dendra2^ cells to CD31-stained bEnd.3 cells after 24 h of coculture. Scale bars, 10 μm. **b** Workflow for the flow cytometry analysis of the mitochondrial acquisition efficiency in bEnd.3 cells after coculturing with NC-MLO-Y4^Mito-Dendra2^ or *Rhot1*^KD^-MLO-Y4 cells. This figure was created by P.L. and D.L.L. and cartonized by Mr. Zihao Li. **c** Representative dot-plots and quantitative result of the percentage of bEnd.3 cells acquired with Mito-Dendra2 fluorescence in entire bEnd.3 cells population after coculturing with NC-MLO-Y4^Mito-Dendra2^ or *Rhot1*^KD^-MLO-Y4 cells (*n* = 3 biologically independent samples). **d** Schematic diagram for the generation of the *Dmp1*^Cre^-*Cox8*^Dendra2^-*Rhot1*^fl/fl^ mouse line. **e** Representative confocal images of Dendra2-labeled mitochondria and CD31-labeled endothelial cells from the femur cortical bone of 6-week-old male *Dmp1*^Cre^-*Cox8*^Dendra2^ control mice and *Dmp1*^Cre^-*Cox8*^Dendra2^-*Rhot1*^fl/fl^ mice. Scale bars, 20 μm. **f** Quantitative assessment of the number of Dendra2-labeled mitochondria

in each TCV as shown in **e**. (Data were quantified from three mice, and four different views per mouse were captured for the quantification). **g** Strategy diagram of the generation of the *Dmp1*^Cre^-*Rhot1*^fl/fl^ mouse line for the specific deletion of the *Rhot1* gene in *Dmp1*-expressing cells by crossing the Loxp-flanked *Rhot1* allele with the *Dmp1*^Cre^ mouse line. **h** Representative confocal images of CD31-labeled blood vessels from femur cortical bone of 6-week-old male *Rhot1*^fl/fl^ control mice or *Dmp1*^Cre^-*Rhot1*^fl/fl^ mice. Scale bars, 200 μm. **i** Quantitative assessment of the number of TCV branches (white arrowhead) shown in **h** (Data were quantified from three mice, and upper and lower cortical bone of each mouse were captured for the quantification). **j** Representative images of high-resolution μCT (1 μm resolution) on 6-week-old male mouse femurs presenting the regressed canal network in *Dmp1*^Cre^-*Rhot1*^fl/fl^ mice compared to *Rhot1*^fl/fl^ control mice. Data were presented as the means ± SEMs. Significance was calculated using unpaired t test with two-tailed *P* value **c**, **f**, **i**. Source data are provided as a Source Data file.

*Rhot1* knocking out did not induce the apoptosis of osteocytes (Supplementary Fig. 5e, f). Consistently, no significant changes of lacunae in cortical bone were observed in the SEMs analysis (Supplementary Fig. 5g, h). The morphology of osteocytes was not altered as evidenced by the unaffected cell length and width, and ratio length/width (Supplementary Fig. 5i, j). Consistently, the number and length of osteocyte dendrites in *Dmp1*^Cre^-*Rhot1*^fl/fl^ mice were found to be unchanged (Supplementary Fig. 5k). Moreover, the bone mass remains unaffected after knocking out *Rhot1* in osteocytes, as supported by unobvious changes in trabecular bone volume (BV), BV/TV, trabecular separation (Tb.Sp), bone mineral density (BMD), cortical bone area (Ct.Ar), cortical thickness (Ct. Th), tissue mineral density (TMD) (Supplementary Fig 5l, m). Together, these data showed the depletion of *Rhot1* in cortical bone do not cause a significant change in bone cell components and

bone mass and supported that MIRO1-mediated mitochondrial transfer from osteocytes to endothelial cells in cortical bone is critical for maintaining the homeostasis of TCV vascularization.

## Endothelial cells acquire osteocyte mitochondria to alleviate oxidative stress and promote angiogenesis

To examine the role of osteocyte-derived mitochondria in endothelial functions, we next isolated mitochondria from MLO-Y4 cells and employed the mitochondrial transplantation assay on bEnd.3 cells[30,31]. Confocal imaging validated the viability of extracted mitochondria from MLO-Y4^Mito-Dendra2^ cells, as evidenced by the efficient staining of the mitochondrial membrane potential-dependent dye MitoTracker Red CMXRos (MTR) (Supplementary Fig. 7a). Next, after co-incubation with bEnd.3 endothelial cells for 24 h, confocal imaging revealed that

mitochondria isolated from MLO-Y4[Mito-Dendra2] cells were capable of being internalized into bEnd.3 endothelial cells (Fig. 5a). To examine the respiratory changes in endothelial cells after acquiring osteocyte mitochondria, we used an Agilent Seahorse XF96 analyzer to monitor the cellular oxygen consumption rates of bEnd.3 endothelial cells (Fig. 5b, c). The results revealed that the endothelial cells transplanted with osteocyte-derived mitochondria (Mito) exhibited higher levels of cellular oxidative phosphorylation, as evidenced by significant increases in the rates of basal respiration, ATP production, maximal respiration, and spare respiratory capacity (Fig. 5c). We also found bEnd.3 pretreated antimycin A and rotenone (A/R), mitochondrial complexes III and I inhibitors, experienced higher mitochondria acquisition rate, suggesting the rescuing role of osteocyte mitochondria on endothelial cells (Supplementary Fig. 7c, d).

The aberrant accumulation of ROS, which usually emerge as the byproduct of mitochondrial oxidative stress, is a primary cause of endothelial dysfunction and thus results in vascular damage[32]. To verify whether mitochondria acquired from osteocytes alleviate ROS stress in endothelial cells and restore their functional disability, antimycin A and rotenone were combinedly utilized to inhibit mitochondrial complexes III and I. Flow cytometry analysis after gradient A/R treatment revealed that the ROS level in bEnd.3 cells was significantly elevated after 2/2 μM A/R treatment (Supplementary Fig. 8a–d). The mitochondrial transplantation assay next proved that the acquisition of MLO-Y4-originated mitochondria remarkably alleviated ROS (DCFH-DA) stress in 2/2 μM A/R-treated bEnd.3 endothelial cells and healthy bEnd.3 (Fig. 5d, e, Supplementary 8 Fig. e–g). These results suggested that osteocyte-derived mitochondria play a role in replenishing energy production and reducing oxidative stress in endothelial cells.

To test the effect of osteocyte mitochondria on endothelial functions, we conducted a mitochondrial transplantation assay on endothelial cells and assessed the alteration of endothelial capability on cell proliferation, tube formation and cell migration. In detail, the proliferation, tube formation and migration ability of bEnd.3 cells were shown to be significantly disabled after different concentrations of A/R treatment, with evidence of decreased CCK8 OD value (Supplementary Fig. 8h–j), branch point number after tube formation assay (Supplementary Fig. 8l–n), and percentage of migrated area on the wound healing assay (Supplementary Fig. 8p–r), respectively. Intriguingly, by transplanting osteocyte mitochondria into A/R-damaged bEnd.3 and healthy bEnd.3 endothelial cells, the CCK8 OD value was significantly elevated in response to different masses of transplanted mitochondria (Fig. 5f-g, Supplementary Fig. 8k). The mass of isolated mitochondria was determined by the ratio of mitochondrial donor cells (MLO-Y4) to mitochondrial recipient cells (bEnd.3). We next examined the osteocyte mitochondria on endothelial tube formation ability and showed that acquisition of osteocyte mitochondria restored the capability of tube formation in A/R-damaged bEnd.3 and healthy bEnd.3 cells (Fig. 5h-i, Supplementary Fig. 8o), which implied the promising role of osteocyte mitochondria in endothelial angiogenesis. As cell migration is also an indicator of endothelial angiogenesis, we next performed cell scratch and wound healing assays to investigate the effect of acquired mitochondria on endothelial cell migration[33]. The results showed that mitochondrial transplantation on healthy and 4/4 μM A/R-damaged endothelial cells effectively recovered the disabled migration ability of bEnd.3 cells under the 30:1 ratio of mitochondrial donor cells (MLO-Y4) to recipient cells (bEnd.3) (Fig. 5j-k, Supplementary Fig. 8s).

Angiogenesis is a key component of bone repair, which relies on new blood vessels to transport the nutrients and inflammatory cells as well as precursor cells[34]. To examine the therapeutic effect of osteocyte-derived mitochondria on bone healing, we performed bone defect surgery on WT mice and isolated mitochondria from MLO-Y4 followed by in situ injected to the defected area of right femur in mice twice in one week (Fig. 5l). Then, the right femurs were collected and performed micro-CT scanning to analyze bone healing and performed

immunofluorescence staining of CD31 to analyze TCVs formation. Confocal imaging analyses of callus TCV network showed higher speed of angiogenesis in bone defect area after the mice received osteocyte mitochondria (Fig. 5m-n). Consistently, the micro-CT analysis revealed improvement of bone healing quality in mitochondria injection group compared with the vehicle group (Fig. 5o-p). Our results validated both a positive role of osteocyte mitochondria in TCVs formation and intensive relationship between angiogenesis and osteogenesis.

Taken together, these results demonstrated that endothelial cells acquire mitochondria from osteocytes to alleviate their oxidative stress, improve energy production, restore angiogenic capability and promotes TCVs formation necessary for bone defect healing.

## Induction of D-sphingosine by mitochondrial transfer activates endothelial function

To investigate the responsible contributors of the pro-angiogenesis effect after acquiring mitochondria, we examined the metabolic changes in endothelial cells after mitochondrial transplantation assay by performing untargeted metabolomic analysis of bEnd.3 endothelial cells that acquired or did not acquire osteocyte mitochondria. The results showed that among 427 (pos370, neg57) identified metabolites, 45 metabolites were differentially more abundant, whereas 2 metabolites were less abundant, in bEnd.3 cells acquiring mitochondria compared with control bEnd.3 cells (Fig. 6a, Supplementary Fig. 9a). Next, we filtered all the differentially detected metabolites that have been reported to be involved in angiogenesis and are biosynthesized or degraded in mitochondria (Fig. 6b), and revealed that D-sphingosine was the only overlapping candidate, which can be catalyzed by sphingosine kinase1 (SPHK1) or sphingosine kinase 2 (SPHK2) to produce sphingosine-1-phosphate (S1P), a frequently reported metabolite that play an important role in regulating endothelial proliferation and angiogenesis[35]. These results suggest that osteocyte mitochondria may regulate angiogenesis through the upregulated sphingolipid pathway.

Next, to test the effect of D-sphingosine on endothelial function, we performed a series of in vitro experiments with D-sphingosine (Sph) supplementation in bEnd.3 endothelial cells. The effect of sphingosine is dose dependent[36,37]. Specifically, high dose of sphingosine induces apoptosis, while low dose of sphingosine promotes cell growth[36,37]. Thus, we investigated approximal concentration of D-sphingosine before further experiment. CCK8 analysis were performed on different concentrations of D-sphingosine and bEnd.3 cell density and the results suggested D-sphingosine works positively lower than 50 nM (Supplementary Fig. 9b-c) with cell density more than 12500 cell/cm² (Supplementary Fig. 9d-e). Then, we investigated the effect of 12.5 nM and 50 nM sphingosine on A/R-damaged bEnd.3 cells. In agreement with previous mitochondrial transplantation data (Fig. 5c-j), D-sphingosine showed a rescuing effect similar to that of osteocyte mitochondria. We first tested its effect of alleviating ROS stress by supplying D-sphingosine in A/R-damaged bEnd.3 cells, and the results showed a significant reduction in ROS stress after supplementation with D-sphingosine in damaged bEnd.3 cells (Fig. 6c-d, Supplementary Fig. 9f). Furthermore, D-sphingosine treatment was also observed to be able to significantly recover endothelial cell proliferation of A/R-damaged bEnd.3 cells (Fig. 6e-f). Likewise, D-sphingosine treatment on A/R-induced endothelial cells were also demonstrated to be capable of restoring endothelial tube formation (Fig. 6g-h) and migration (Fig. 6i-j). We also investigated the effect of D-sphingosine on mitochondria transfer. Flow cytometry analysis showed that exogenous D-sphingosine supplementation decreased the mitochondria transfer process (Supplementary Fig. 9g-i). Moreover, the transfer rate decreased with a higher concentration of D-sphingosine (Supplementary Fig. 9i).

Further, SPHK1 has been reported to be involved in cell growth, proliferation, anit-apoptosis, and inflammatory response[38], while

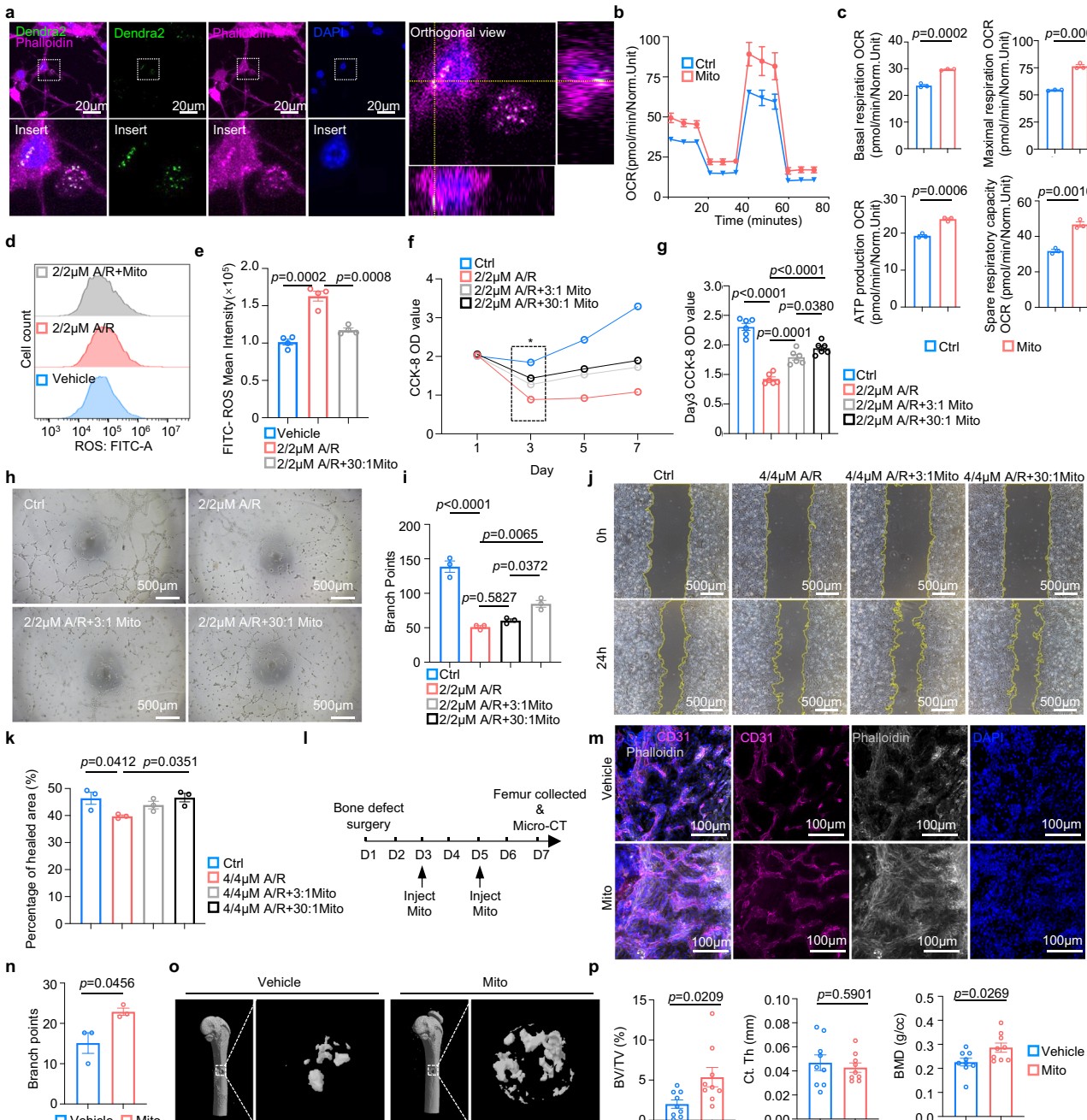

**Fig. 5 | Mitochondria transferred from osteocytes restore endothelial dysfunction. a** Confocal images of the acquisition of MLO-Y4-derived mitochondria (Mito) in bEnd.3 endothelial cells. **b**, **c** Extracellular flux analysis **b** quantitative results **c** and of OXPHOS activity in vehicle-treated healthy bEnd.3 cells, healthy bEnd.3 cells transplanted with MLO-Y4 cells mitochondria, 2 μM A/R-damaged bEnd.3 cells, and 2 μM A/R-damaged bEnd.3 cells transplanted with MLO-Y4 cells mitochondria ($n = 3$ biologically independent samples). **d**, **e** Representative histogram plot **d** and quantitative result **e** of DCFH-DA fluorescence (ROS) mean intensity of vehicle-treated bEnd.3 cells, 2 μM A/R-damaged bEnd.3 cells and 2 μM A/R-damaged bEnd.3 cells transplanted with MLO-Y4 cells mitochondria ($n = 4$ biologically independent samples). **f**, **g** CCK-8 cell proliferation assay **f** on vehicle-treated healthy bEnd.3 cells, 2 μM A/R-damaged bEnd.3 cells, and 2 μM A/R-damaged bEnd.3 cells transplanted with MLO-Y4 cells mitochondria at a ratio of 3:1 or 30:1 over 7 days and statistical result **g** of the OD value on day 3 ($n = 6$ biologically independent samples). **h**, **i** Representative images **h** and quantitative result **i** of the

tube formation assay on vehicle-treated healthy bEnd.3 cells, 2 μM A/R-damaged bEnd.3 cells, and 2 μM A/R-damaged bEnd.3 cells transplanted with MLO-Y4 cells mitochondria at a ratio of 3:1 or 30:1 ($n = 3$ biologically independent samples). **j**, **k** Representative images **j** and quantitative result **k** of the wound healing assay on vehicle-treated bEnd.3 cells, 2 μM A/R-damaged bEnd.3 cells, and 2 μM A/R-damaged bEnd.3 cells transplanted with MLO-Y4 cells mitochondria at a ratio of 3:1 or 30:1 ($n = 3$ biologically independent samples). **l** Workflow for bone defect surgery and mitochondrial injection. **m**, **n** Representative confocal images **m** of femur callus in bone defected area and quantitative results **n** ($n = 3$ biologically independent samples). **o**, **p** Representative micro-CT images **o** of femur defects 1 week after surgery and histomorphometric analysis **p** of the regenerated bone ($n = 9$ biologically independent samples). Data were presented as the means ± SEMs; Significance was calculated based on unpaired t test with two-tailed *P* value **c**, **e**, **g**, **n**, **p** or one-way ANOVA followed by Tukey's post hoc test **i**, **k**. Source data are provided as a Source Data file.

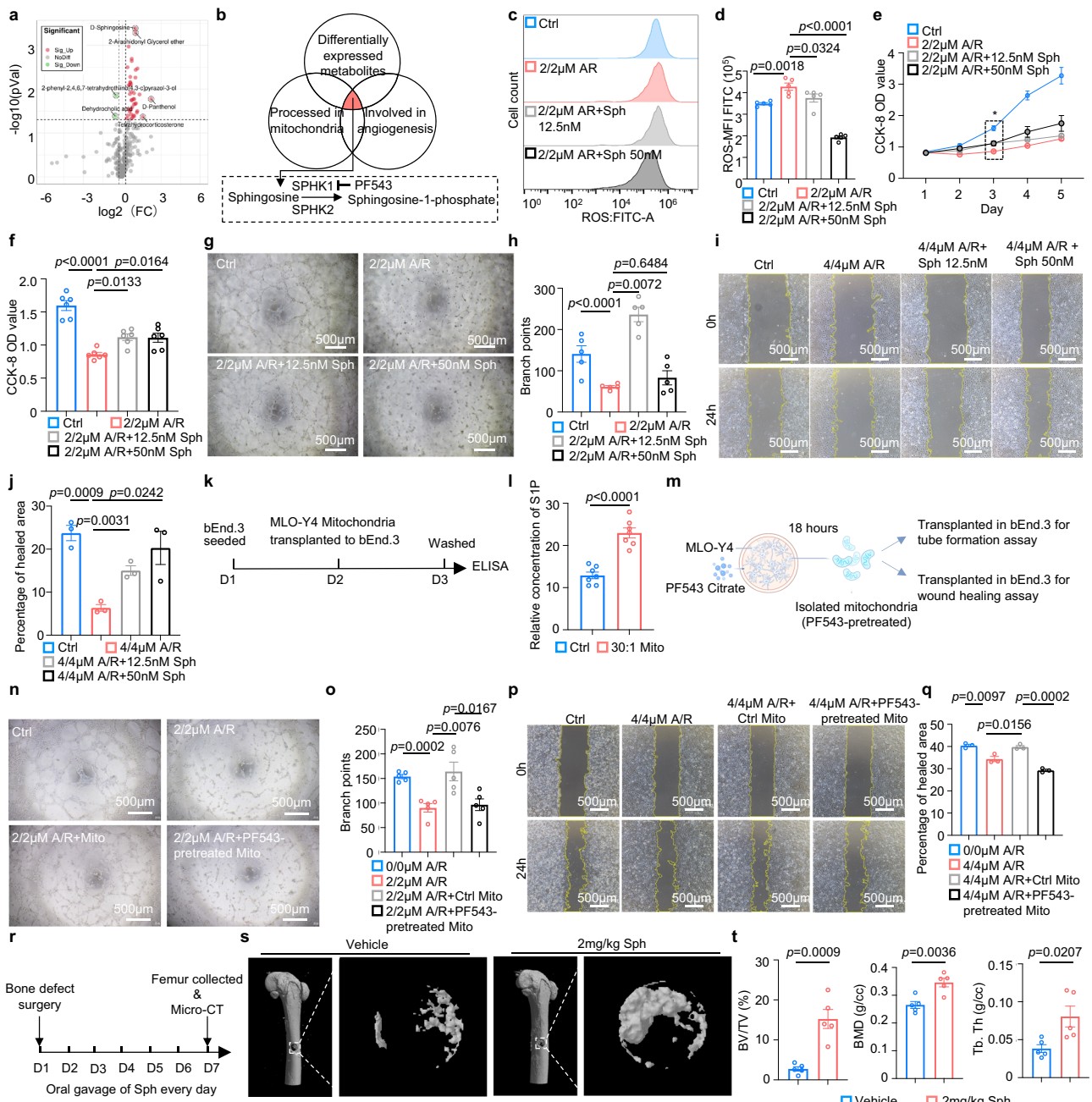

**Fig. 6 | Osteocyte-derived mitochondria regulate endothelial function by inducing D-sphingosine. a** Volcano plot of untargeted metabolomic analysis of the differentially detected (P < 0.05, FC > 1.2 or FC < 0.833, VIP > 1) metabolites in bEnd.3 cells transplanted with MLO-Y4 cells mitochondria. **b** Schematic of screening potential metabolites related to mitochondria that promote angiogenesis. **c, d** Representative histogram plot **c** and quantitative result **d** of ROS level of D-sphingosine treated healthy and damaged bEnd.3 cells. (n = 5 biologically independent samples). **e, f** The CCK-8 cell proliferation assay **e** of D-sphingosine treated healthy and damaged bEnd.3 cells throughout 5 days and statistical result **f** of the OD value on day 3 (n = 6 biologically independent samples). **g, h** Representative images **g** of the tube formation assay and quantitative result **h** of D-sphingosine treated healthy and damaged bEnd.3 cells. (n = 5 biologically independent samples). **i, j** Representative images **i** and quantitative result **j** of the wound healing assay of D-sphingosine treated healthy and damaged bEnd.3 cells. (n = 3 biologically independent samples). **k, l** Workflow **k** and the quantitative result **l** of sphingosine-1-phosphate (S1P) concentration in bEnd.3 cells, both with and without

the acquisition of MLO-Y4 cells mitochondria. (n = 7 biologically independent samples). **m** Workflow for tube formation and wound healing assays using bEnd.3 endothelial cells transplanted with mitochondria from osteocytes pre-treated with PF543 citrate. This figure was created by P.L. and cartonized by Ms. Lina Cao. **n, o** Tube formation assay on differentially treated bEnd.3 cells transplanted with mitochondria from PF-543 citrate-induced MLO-Y4 cells **n** and statistical result **o**. (n = 5 biologically independent samples). **p, q** Wound healing assay **p** and quantitative result **q** on healthy, damaged and damaged bEnd.3 cells transplanted with mitochondria from normal or PF-543 citrate-induced MLO-Y4 cells (n = 3 biologically independent samples). **r** Workflow for femoral defect surgery and D-sphingosine treatment. **s** Representative micro-CT images of femur defects 1 week after surgery. **t** Histomorphometric analysis of the regenerated bone. (n = 5 biologically independent samples). Data were presented as the means ± SEMs. Significance was calculated using unpaired t test with two-tailed P value **a, j, n, q, s** or one way ANOVA followed by Turkey's post hoc test **d, f, h**. Source data are provided as a Source Data file.

SPHK2 was described as a growth inhibitor[39,40]. Considering the sub-cellular localization of SPHK1 was reported to be strongly associated with mitochondrial outer membrane[41], we thus speculate that the extrinsic osteocyte mitochondria transferred into endothelial cells might promote angiogenesis by activating the synthesis of D-sphingosine and enhancing the catalytic effect of SPHK1 to induce S1P production in endothelial cells. We next tested the alteration of S1P in endothelial cells after acquiring osteocyte mitochondria by conducting ELISA assay on bEnd.3 cells transplanted with or without osteocyte mitochondria (Fig. 6k). The results revealed that supplementation of osteocyte mitochondria significantly improved S1P level in endothelial cells (Fig. 6l). To investigate whether osteocyte mitochondria rescuing endothelial cells functions through the effect of SPHK1, we performed endothelial cell tube formation and scratching assay with the supplementation of mitochondria isolated from MLO-Y4 cells pre-treated with SPHK1 inhibitor, PF-543 Citrate (PF543) (Fig. 6m). The results showed that the rescuing effect of osteocyte mitochondria was abolished after mitochondrial donor osteocytes were pretreated with PF543, as evidenced by endothelial tube formation assay (Fig. 6n-o) and migration assay (Fig. 6p-q). In summary, these data indicated that osteocyte mitochondria promote endothelial cells angiogenesis at least in part mediated via sphingosine pathway.

To examine the therapeutic effect of sphingosine in vivo, we examine the role of sphingosine on bone healing (Fig. 6r). Femoral bone defects were created on the inner side of the right distal femur cortical bone of the mice, and the mice were then treated with 0 mg/kg or 2 mg/kg D-sphingosine by oral gavage once a day for 7 consecutive days. Data on the 3D-reconstructed canal systems showed an increased canal component in D-sphingosine treatment group (Supplementary Fig. 10a), suggesting a higher ability of TCVs formation. To further investigate the TCVs alteration in these two groups, histological sections and immunofluorescence staining were utilized on the cortical bone defect area to compare TCVs architecture (Supplementary Fig. 10b-c). The results showed that numbers of blood vessel branch in

bone defect area were significantly higher in mice treated with D-sphingosine than those in mice treated with vehicle (Supplementary Fig. 10c). Micro-CT analysis of the bone defect area revealed that bone regeneration was enhanced in mice after receiving D-sphingosine treatment compared with control mice (Fig. 6s). Specifically, bone volume/tissue volume (BV/TV), bone thickness (Tb. Th), and bone mineral density (BMD) of newly generated bone in the bone defect area were significantly increased after D-sphingosine treatment (Fig. 6t). In summary, these results supported that sphingolipid pathway at least in part involves in the osteocyte mediated endothelial cells angiogenesis in cortical bone.

## Discussion

TCVs provide effective communications between extra and intra bone/marrow microenvironment via circulation. Here we show the osteocytes, the key cellular components of cortical bone, play a vital role in regulation of the TCVs homeostasis. Osteocytes transfer mitochondria to the endothelial cells of TCV. Partial ablation of osteocytes causes TCV regression and reduction of angiogenic genes. The mitochondria transfer from osteocyte to endothelial cells mediated by MIRO1 rescues endothelial stress resulting in restoration of the proliferation, tube formation, migration ability and acceleration of TCVs formation. Metabolomic analysis further showed mitochondrial transfer induces the biosynthesis and SPHK1-dependant catalysis of D-sphingosine. Administration of D-sphingosine promotes TCVs and bone formation in mouse cortical bone defect model (Fig. 7).

TCV is a perpendicular network within cortical bone. Using histological sections and µCT scans, it has been shown previously that over 86% of cortical bone canals were occupied by blood vessels[16]. In our study, we have used 1 µm resolution micro-CT to examine the perpendicular canal network surrogated to TCVs. The characteristics of canal acquired by µCT scanning were consistent with the TCV network morphology shown by CD31 immunofluorescence stanning, suggesting the potential application of high resolution µCT in

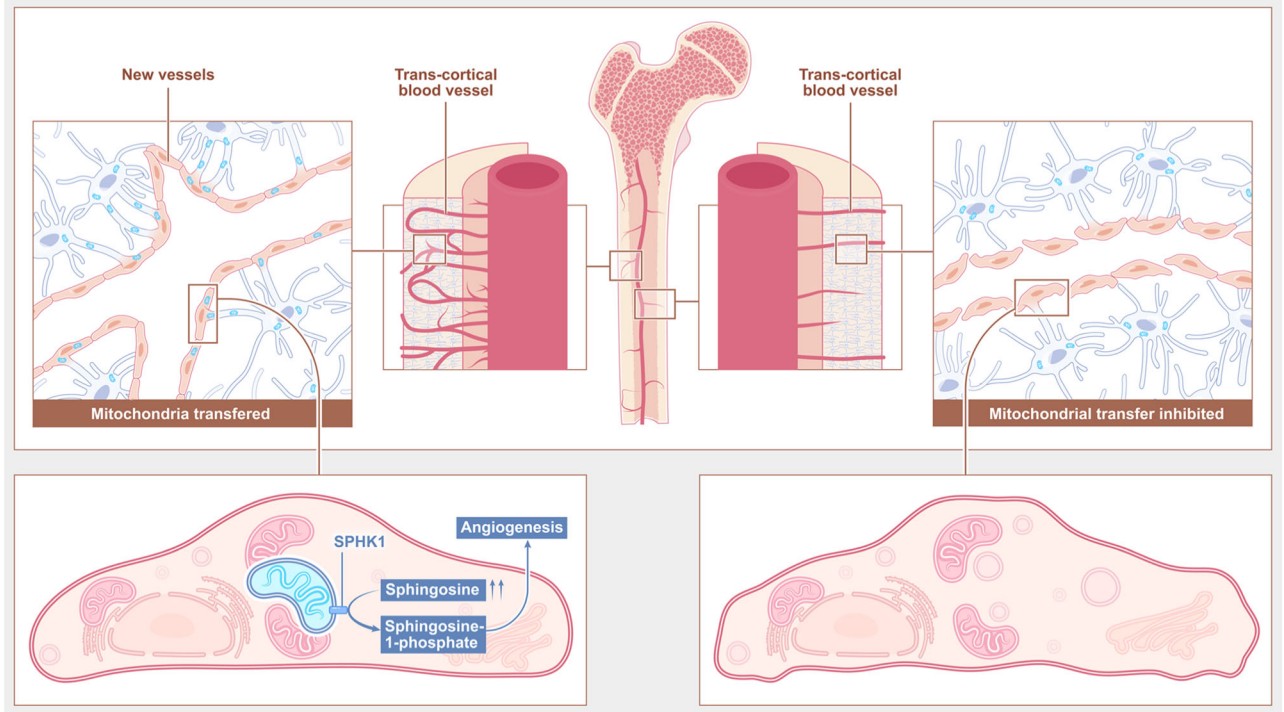

**Fig. 7 | Schematic diagram of mitochondria transferring from osteocytes to transcortical endothelial cells for maintaining TCVs vascularization.** Mitochondria transferred from osteocytes promote angiogenesis of TCVs via activating the production of endothelial D-sphingosine, which is further catalyzed into sphingosine-1-phosphate by the SPHK1 of osteocyte-derived mitochondria. This figure was created by P.L. and D.L.L. and cartonized by Ms. Lina Cao.

exploring the TCVs related work[42]. In cortical bone, osteocytes have multiple long, neuron-like dendrites to connect to each other and form a spatial network[43]. These dendrites serve as mechanotransducers that transmit mechanical signals to the cell body to regulate osteoclasts and osteoblasts and act as channels for the exchange of material, such as mitochondria, to maintain the metabolic balance of the whole system by rescuing stressed osteocytes[11,43]. Here we show for the first time that osteocytes in cortical bone stretch their dendrites with endfeet-like structures to endothelial cells and directly communicate with the network of TCVs. We further show that osteocytes are indispensable for endothelial cell angiogenesis and blood vessel development in cortical bone.

Spontaneous intercellular transfer of mitochondria occurs under physiological conditions and contributes to tissue development and remodeling. Various routes for mitochondrial transfer have been reported, including tunneling nanotubes, microvesicles, and extrusion and internalization of free mitochondria[23]. Our study suggested osteocytes deliver mitochondria to their dendritic endfeet in a MIRO-1-dependent manner and then transfer mitochondria to endothelial cells. However, the mechanisms on how mitochondria are released and entered to endothelial cells are not clear.

Vascularization in bone is an energy-consuming process, in which mitochondria play a crucial role[24]. ATP produced by oxidative phosphorylation (OXPHOS) in mitochondria not only acts as a signal to activate angiogenesis but also provides essential energy for the proliferation and migration of endothelial cells to support vascular development[44]. In addition, the levels of reactive oxygen species (ROS), detrimental byproducts of aerobic metabolism[45], are mainly regulated by mitochondria to maintain their dynamic balance[46]. Excessive ROS accumulation in mitochondria will secondarily lead to cell stress, including abnormal cell signaling, inflammation[47], cellular senescence[48], apoptosis[49] and necrosis[32], which eventually weaken angiogenesis[50]. As the primary layer of the circulatory system, endothelial cells are susceptible to multiple circulating toxins and pathogens that induce oxidative stress. Thus, endothelial cells are vulnerable to ROS accumulation as well as mitochondrial stress load. Our data are consistent with previous studies showing that the osteocyte-derived mitochondria alleviated the ROS stress of damaged endothelial cells to help restore the angiogenesis ability of endothelial cells[51]. Furthermore, the osteocyte-derived mitochondria act as critical mediators for vascular formation within cortical bone.

It is conceivable that the osteocyte-derived mitochondria regulate the endothelial cells not only by replacing the dysfunctional mitochondria, but also acting as a trigger to activate signal pathways. Although our data showed endothelial cells that received osteocytes-derived mitochondria had a higher level of D-sphingosine, it is not clear why is sphingosine increased in ECs via mitochondria transfer. While we speculated that transferred mitochondria may activate the production of sphingosine in ECs, further investigation on the mechanism of upregulation of sphingosine in osteocyte mitochondria transfer to endothelial cells is required. In addition, SPHK1 are widely expressed in all types of cells[52]. When SPHK1 was inhibited in donor cells, the rescuing effects of transferred mitochondria were significantly impaired, even though the recipient endothelial cells itself also expressed SPHK1, indicating the unreplaceable role of SPHK1 expressed in osteocyte mitochondria after transferring to endothelial cells. However, the underlying mechanism of the uniqueness of mitochondria remains unclear. Our data elucidated the specific function of osteocyte-derived mitochondria in regulating the sphingosine pathway of endothelial cells in TCVs.

Our study has limitations. The use of *Dmp1^Cre^-DTA^ki/wt^* mice as they may not be exclusive to osteocyte. Nevertheless, our comprehensive approach of in vitro assessment and the combined use of both *Dmp1^Cre^-Rhot1^fl/fl^* mice and *PF4^Cre^-Cox8^Dendra2^* mice enable us to demonstrate the specificity of osteocyte mitochondrial transfer to endothelial cells. Our

second limitation is that "lower efficiency" of mitochondria transfer in transwell coculture system could potentially be attributed to the dilution of released mitochondria in vitro. In addition, the use of isolated mitochondria for cocultured with endothelial cells may not mimic the reality in vivo even this is a common method used in mitochondrial transfer[30,53–57].

In conclusion, our study reveal that endothelial cells acquire mitochondria from endfeet of osteocytes to maintain the homeostasis of TCV network. TCV vascularization is supported by osteocyte-derived mitochondria transfer to endothelial cells. Increased D-sphingosine as result of mitochondria transfer displays similar proangiogenic effect as to osteocyte mitochondrial transfer. Administration of D-sphingosine accelerates TCVs formation and bone healing in mouse cortical bone defect model. Our results provide new insights into osteocyte-TCV interactions and inspire the potential application of mitochondrial therapy for bone-related diseases.

## Methods

### Subjects

All procedures regarding animal maintenance and experiments were carried out in accordance with the Institutional Animal Care and Use Committee (IACUC) and approved by the Animal Care and Use Committee of Shanghai Sixth People's Hospital Affiliated to Shanghai Jiao Tong University School of Medicine. C57BL/6 J (WT) (stock number: N000013) and mGmT [B6; JGpt-H11^em1Cln(CAG-LoxP-ZsGreen-Stop-LoxP-tdTomato)^/Gpt, stock number: T006163] mouse strains were purchased from Gem-Pharmatech Co. Ltd. Ltd. (Jiangsu, China). The *Dmp1^cre^* transgenic mice were provided by J. Q. (Jerry) Feng at Texas A&M College of Dentistry, USA[58]. *Cox8^Dendra2^* [B6; 129SGt(ROSA)26Sor^tm1(CAG-COX8/Dendra2)Dcc/J^], stock number: 0183885] and *Rhot1^fl/fl^* [B6(Cg)-Rhot1^tm2.1Jmsu^/J, stock number: 031126] mouse strains were purchased from the Jackson Laboratory. An osteocyte ablation mouse model was established by crossing *Dmp1^Cre^* mice with *Rosa26^em1Cin(SA-IRES-Loxp-ZsGreen-stop-Loxp-DTA)^* homozygotes to obtain *Dmp1^Cre^-Rosa26^em1Cin(SA-IRES-Loxp-ZsGreen-stop-Loxp-DTA)^* heterozygotes (DTA^ki/wt^)[14]. The mice were group-housed with at most 5 per cage in a stable environment (23–25 °C ambient temperature and 50% humidity). They were maintained under a 12-h light–dark cycle (lights on from 7:00 to 19:00) with free access to water and food ad libitum. Mice were fed with normal diet with 11.1% energy from fat, 22.9% from protein and 66% from carbohydrates. This study was performed on male mice to exclude the potential influence of estrogen on bone homeostasis.

### Cell culture

The mouse endothelial cell line bEnd.3 was obtained from ATCC, Catalog number CRL-2299. MLO-Y4 was a gift from Dr Lynda F Bonewald. bEnd.3 cells were maintained in high-glucose Dulbecco's modified Eagle's medium (DMEM) containing 10% fetal bovine serum (FBS) (Gibco Cat. 10099141) and 1% penicillin/streptomycin (P/S). MLO-Y4 cells were maintained in alpha–minimal essential medium (alphaMEM) containing 10% FBS and 1% P/S. For 2D coculture, bEnd.3 cells were seeded on coverslips in 24-well plates. After 24 hours, MLO-Y4 cells were seeded in the wells that cultured bEnd.3 cells. The cells were maintained at 37 °C in a 5% CO2 humidified incubator.

### Cell transfections

To generate the MLO-Y4^Mito-Dendra2^ cell line, Mito-Dendra2 was amplified from the plasmid (https://www.addgene.org/55796/) (the detailed sequences of Mito-Dendra2 are listed in Supplementary Table 2a) and cloned into the modified pHAGE vector containing Flag using restriction enzymes Not I and BamH I. The insertions were confirmed by DNA sequencing. The double oligonucleotides were annealed at 95 °C for 20 min and cloned into the pHAGE vector. 293 T cells cultured at 70% confluency were then cotransfected with shRNA and the lentivirus packaging plasmid (VSVG and Δ8.9). The lentiviral particles were harvested from the medium after 48 hours of cotransfection. The cell

medium was passed through a 0.22-μm filter, and the lentivirus was collected into Eppendorf tubes and kept at −20 °C for short-term storage or −80 °C for long-term storage. MLO-Y4-Cox8$^{Dendra2}$ cells were obtained by transfection with lentivirus for 48 hours followed by conditional selection with 10 mg/ml puromycin (Gibco, Cat. A1113803).

To generate the *Rhot1$^{KD}$*-MLO-Y4$^{Mito-Dendra2}$ cell line, shRNA oligonucleotides were synthesized by Tsingke Biotechnology Co., Ltd. (the detailed shRNA sequences are listed in Supplementary Table 2b). The double oligonucleotides were annealed at 95 °C for 20 minutes and cloned into the pLKO.1-hygro lentivirus vector. 293 T cells were at 70% confluency and cotransfected with shRNA and the lentivirus packaging plasmid (psPAX2 and pMD2.G). An appropriate empty vector was created for shRNA constructs. The lentiviral particles were prepared as previously described. MLO-Y4$^{Mito-Dendra2}$ cells were transduced with lentivirus for 48 h and selected by 500 μg/ml hygromycin (Thermo Fisher Scientific, Cat. 10687010). Stable cell lines were cultured in 500 μg/ml hygromycin.

To generate *Vegfc$^{KD}$*-MLO-Y4, *Slit3$^{KD}$*-MLO-Y4, *Notch3$^{KD}$*-MLO-Y4, *Notch4$^{KD}$*-MLO-Y4 cells, cells were plated on 12 well plate, and incubated in 1% lipofectamine (Invitrogen, Cat. L3000015) and 50 nM siRNA (Genomeditech, Shanhai, China) in opti-MEM for 6 hours (the detailed siRNA sequences are listed in Supplementary Table 2c). Then the media were refreshed with alphaMEM containing 10% FBS and 1% P/S and cells were incubated for 42 hours before further experiments.

## Scanning electron microscopy (SEM) and transmission electron microscopy (TEM)

SEM and TEM were performed according to previous studies[12]. For TEM images, the nondecalcified thin bone sections were cut and stained with uranyl acetate and lead citrate and then photographed using a Philips CM12 in TEM mode.

For resin-casted osteocyte-dendrite SEM, the surface of MMA-embedded bones was repolished and acid-etched with 37% phosphoric acid for 2-10 seconds, followed by two 20-minute washes with 5% sodium hypochlorite. Then, the blocks were washed, dried, and coated with gold/palladium for SEM imaging using Zeiss 1555VP-FESEM.

## Evans Blue assay

Mouse was injected with 2% Evans Blue (Sigma Aldrich, Cat. E2129) in saline solution intraperitoneally at a concentration of 10 μl/g of body weight for twelve hours. Bone tissues from sacrificed mouse were then collected and fixed in 4% PFA for 24 hours. After washed by PBS for 3 times, the mouse femur was sectioned with a cryostat microtome (Leica) until the embedded osteocytes in the central region of the cortical bone were exposed. The connection between TCV and osteocyte dendrites was assessed by visualizing Evans Blue-filled blood vessels using the Olympus FV3000 confocal microscope.

## CellMask Staining

bEnd.3 cells were seeded on coverslips in 24-well plates and cultured in DMEM containing 10% FBS and 1% P/S for 8 hours. Next, coverslips were washed 3 times with PBS, and the medium was refreshed with DMEM containing 1× Cell Mask Deep Red Actin Tracking Stain (Invitrogen, Cat. H32713) for 30 minutes. After washing, the medium was refreshed with DMEM containing 10% FBS and 1% P/S, and the cells were cultured overnight. Next, coverslips were washed 3 times with PBS before coculture with MLO-Y4$^{Mito-Dendra2}$ cells.

## Immunofluorescence staining and confocal imaging

For bone immunofluorescence staining, mouse femurs were collected by cleaning off soft tissues and periosteum under a microscope. The collected femurs were fixed in 4% PFA at 4 °C for 24 hours. Then the femurs were embedded in O.C.T. compound (Sakura Finetek Cat. 4583) for coronal sections of the femurs with a cryostat microtome (Leica) to expose the embedded osteocytes in the central region of the

cortical bone. After washing the O.C.T. compound by PBS, femurs were permeabilized with 0.1% Triton X-100 for 1 hour. Blockage of femurs was next performed with 3% bovine serum albumin (BSA) (Sigma-Aldrich Cat. A7030)/PBS for 1 hour. For bone sample injected with Evans Blue, Rhodamine Phalloidin (1:200, Invitrogen, Cat. R415) was stained for 24 hours at room temperature to label osteocyte dendrites. For bone samples without Evans Blue injection, the primary anti-CD31 antibody (1:200, R&D systems, Cat. AF3628) was incubated for 48 hours at 4 °C. After washing with PBS for 3 times, femurs were incubated with donkey anti-goat IgG (H + L) cross-adsorbed secondary antibody Alexa Fluor 647 (1:200, Thermo Fisher Scientific, Cat. A21447) mixed with Rhodamine Phalloidin (1:200, Invitrogen, Cat. R415) for 24 hours at room temperature. After washed by PBS for 3 times, bone samples were counterstained with DAPI for 15 minutes at room temperature. Next, PBS-washed femurs were mounted onto 35mm glass-bottom petri dishes (MatTek, Cat. P35G-0.170-14-C) with ProLong Diamond antifade medium (Invitrogen, Cat. P36971). Confocal images were instantly acquired on Olympus FV3000, Olympus SpinSR, or Zeiss 710 confocal microscope.

For immunofluorescence staining of cell samples, cells were fixed with 4% PFA for 15 minutes followed by 20 minutes permeabilization with 0.1% Triton X-100. Next, cells were washed with PBS for 3 times and then blocked with 3% BSA/PBS for 1 hour. Next, CellMask-labeled bEnd.3 cells that were co-cultured with MLO-Y4$^{Mito-Dendra2}$ or normal bEnd.3 cells transplanted with isolated mitochondria from MLO-Y4$^{Mito-Dendra2}$ cells were stained with Alexa Fluor Plus 647 Phalloidin (1:200, Invitrogen, Cat. A30107) or Rhodamine phalloidin (1:200, Invitrogen, Cat. R415), respectively, for 20 minutes followed by staining of DAPI for 15 minutes. For bEnd.3 cell samples co-cultured with NC-MLO-Y4$^{Mito-Dendra2}$ or *Rhot1$^{KD}$*-MLO-Y4$^{Mito-Dendra2}$ cells, the primary anti-CD31 antibody (1:100, R&D Systems, Cat. AF3628) were incubated overnight at 4 °C. After washing with PBS for 3 times, the cells were incubated with donkey anti-goat IgG (H + L) cross-adsorbed secondary antibody Alexa Fluor 568 (1:200, Thermo Fisher Scientific, Cat. A11057) for 2 hours at room temperature followed by 20 minutes incubation of Alexa Fluor Plus 647 Phalloidin (1:200, Invitrogen, Cat. A30107) and 15 min incubation of DAPI. For staining of SPHK1 on MLO-Y4$^{Mito-Dendra2}$ cells, the primary anti-SPHK1 antibody (1:100, Affinity, Cat. DF6005) were incubated overnight at 4 °C. After washing with PBS for 3 times, the cells were incubated with donkey anti-rabbit IgG (H + L) cross-adsorbed secondary antibody Alexa Fluor 647 (1:200, Thermo Fisher Scientific, Cat. A31573) for 2 hours at room temperature followed by 30 minutes incubation of Alexa Fluor Plus 647 Phalloidin (1:200, Invitrogen, Cat. A30107) and 15 minutes incubation of DAPI. Cells samples were then mounted with ProLong Diamond antifade medium and dried overnight before confocal imaging using Leica SP8 (Leica Application Suite X 3.5.5.19976), ZEISS LSM 710 (Zeiss ZEN pro), Olympus SpinSR/fv3000 (Olympus cellSens Dimension 3.2), or Nikon A1 confocal microscopes. For immunostaining of mitochondria isolated from MLO-Y4$^{Mito-Dendra2}$, the isolated mitochondrial pellet was resuspended and incubated in 100 μl DMEM containing 1% P/S and 500 nM Mitotracker Red CMXRos (MTR) (Invitrogen, Cat. M7512) for 30 minutes at 37 °C. Next, 20 μl of the suspended media was dipped on glass slide and covered by a glass coverslip. The isolated mitochondria were immediately examined using Leica SP8 confocal microscope.

## Measurement of osteocyte dendrites near the TCV

Confocal images of the femur cortical bone were acquired at the mid-cortical region. Then the images were analyzed by using ImageJ. To qualify osteocytes near vessels, the number of osteocytes that contacted each TCV was counted. To qualify the angle of osteocytes, osteocytes were divided into near-vessel and nonnear-vessel types according to whether they were in contact with TCVs. Using the angle measure tool of ImageJ, the angle between the cortical long axis and the osteocyte long axis was measured 3 times. The absolute value of

the angle was recorded, and the average value was calculated to represent the angle of an osteocyte. For qualifying osteocyte dendrites, only osteocytes that had dendrite contact with TCVs were included. The number of dendrites located in the vessel near-side or vessel off-side was manually counted according to the relative position to the osteocyte long axis. In addition, the number of dendrites that were in contact with TCVs or in noncontact with TCVs in the vessel near side was also counted.

## Micro quantitative computed tomography (μCT) analysis
The mouse femurs were collected and fixed in 4% PFA for 24 h before processing for μCT scanning. Scanning was performed with the Sky-Scan1276 μCT instrument at 1μm resolution, and original data were then reconstructed for further analysis. The third trochanter of the mouse femur was marked as the reference for selecting the region of interest (ROI), which was set with an offset of 40 slides below the reference followed by a height of 2001 slides in CTAn software. The canals within the cortical bone of the ROI were analyzed in CTAn software (CTAn 1.16.1.0 + ) by phase contrast processing, which enables the visualization of canals within cortical bone[59,60]. The presented images of three-dimensional reconstructions were acquired by stacking the two-dimensional images from the indicated regions with CTVox software (CTvox 3.3.1).

## Gene ontology (GO) analysis and GSEA
Datasets of bulk RNA-Seq on femur cortical bone of $Dmp1^{Cre}$-$DTA^{ki/wt}$ mice and the control $DTA^{ki/wt}$ mice was utilized based on the reported study from our group (GSE202356, http://www.ncbi.nlm.nih.gov/geo/)[14]. GO analysis was performed to facilitate elucidating the biological implications of marker genes and differentially expressed genes. Differential gene analysis was performed by DESeq2 software and then subjected to enrichment analysis of GO functions. GSEA analyses were performed using normalized expression to identify the most significantly pathways following KEGG gene setsGSEA was performed using GSEA software (version 4.1.0; Broad Institute, MIT). Genes were ranked according to their expression; gene sets were searched from website (https://www.gsea-msigdb.org).

## Antimycin A and rotenone treatment
bEnd.3 cells were seeded into plates and cultured with DMEM containing 10% FBS and 1% P/S for 24 hours. Next, the medium was treated with DMEM containing relative concentrations of antimycin A (Sigma Aldrich, Cat. a8674) and rotenone (Sigma Aldrich, Cat. r887) for 3 hours. After washing gently with PBS, the medium was replaced by DMEM containing 10% FBS and 1% P/S, and culturing was continued for further assays.

## Mitochondrial transfer rate assessment by Flow cytometry
For assessing mitochondrial transfer rate in the coculture system, $5 \times 10^4$ bEnd.3 cells and $5 \times 10^4$ or $15 \times 10^4$ MLO-Y4/MLO-Y4$^{Mito-Dendra2}$ cells were seeded in 6 well plated. The media were refreshed every 24 hours. After targeted time (24 hours, 48 hours, 72 hours or 7 days), cells were dispatched and stained with Zombie Violet™ Fixable Viability Kit (Biolegend, Cat. 423113), TruStain fcX™ PLUS (Biolegend, Cat. 156603), APC anti-mouse CD31 Antibody (Biolegend, Cat. 102510) sequentially. Cell pellets were then resuspended in 200 μl PBS containing 2% FBS and immediately analyzed with a CytoFLEX Flow Cytometer (Beckman Coulter, California, United States, CytExpert 2.3.1.22). bEnd.3 cocultured with MLO-Y4 (without Dendra2 fluorescence) were utilized for gating the Dendra2+ bEnd.3 cells. Data were analyzed by FlowJo software version 10.4.

## Histological section and immunochemistry staining
Femurs or tibias were collected from mice and soft tissue was removed. Bone samples were fixed in the 4% PFA for 24 hours and decalcification was performed with 10% EDTA. 4μm-thick paraffin-embedded sections were obtained for further staining. ALP staining was performed by incubating sections with Rabbit polyclonal antibody to Alkaline Phosphatase (1:100, Affinity, Cat. DF6225) overnight at 4 ˚C. After washing with PBS for 3 times, samples were incubated with HRP conjugated Goat Anti-Rabbit IgG (H + L) (Servicebio, Cat. GB23303) at room temperature for 50 minutes. The slices were photographed under a microscope (Nikon E100, Japan). TRAP staining was performed by using tartrate-resistant acid phosphatase (TRAP) kit (Servicebio, Cat. G1050) according to the manufacturer's instructions. Imaging was performed on a microscope (Nikon Eclipse E100, Japan). TUNEL staining was performed by using Fluorescein (FITC) TUNEL Cell Apoptosis Detection Kit (Servicebio, Cat. G1501). All kit instructions were followed for paraffin-embedded tissues with a 22-minute proteinase K treatment and fixation with 4% paraformaldehyde. Imaging was performed on a microscope (Nikon, Eclipse C1, Japan).

## ROS measurement
Cellular ROS levels were determined using an ROS probe, 2',7'-dichlorofluorescin diacetate (DCFH-DA) (Solarbio, Cat. D6470). bEnd.3 cells seeded in 12-well plates were treated with DMEM containing 10 μM DCFH-DA for 20 minutes at 37 °C in a 5% CO2 humidified incubator. Next, the cells were washed with PBS one time and centrifuged at $300 \times g$ for 5 minutes. Cell pellets were then resuspended in 200 μl PBS containing 2% FBS and immediately analyzed with a CytoFLEX Flow Cytometer (Beckman Coulter, California, United States).

## RNA isolation and real-time quantitative PCR
For isolating RNA from cells, an RNA purification kit for cells (EZBioscience, Cat. B0004DP) was utilized according to the manufacturer's protocol. For isolating RNA from cortical bone, femurs and tibia were cleaned of soft tissue and bone marrow and crushed in a tissue grinder machine (Servicebio). RNA was isolated from the bone powder using the RNA purification kit for tissue (EZbioscience, Cat. EZB-RN001-plus) according to the manufacturer's protocol. An additional DNase1 digestion step was performed to ensure that the samples were not contaminated with genomic DNA. RNA concentration was assessed with Nanodrop spectrophotometers (Thermo Fisher Scientific, QuantStudio™ 7 Flex Real-Time PCR System, QuantStudio Real-Time PCR 1.3).

For reverse transcription, 1000 ng RNA was reverse transcribed using 4×Reverse Transcription Master Mix (EZbioscience, Cat. EZB-RT2GQ). qPCR was performed using 2×SYBR Green Color qPCR Mix (EZbioscience, Cat. A0001-R1) following the manufacturer's recommendation. Samples were tested on a Quant StudioTM 7 Flex Real-Time PCR System (Thermo Fisher Scientific). The results were calculated using the ΔΔCT method and are presented as the x-fold increase relative to GAPDH mRNA levels. Primers were synthesized by BioSune company and are listed in (Supplementary Table 1b).

## Mitochondrial isolation and transplantation
Mitochondria were extracted from MLO-Y4 cells by a density gradient ultracentrifugation with the help of mitochondrial isolation kit (Thermo Fisher Scientific, Cat. 89874) according to the manufacturer's protocol. For mitochondrial transplantation, the isolated mitochondria were suspended in DMEM containing 10% FBS and 1% P/S, added to bEnd.3 medium and cocultured for at least 24 hours.

## Measurement of OXPHOS activity of bEnd.3
bEnd.3 cells were seeded on 6 well plate at a density of 100,000 per well. After 12 hours, mitochondria from MLO-Y4 cells well isolated and co-cultured with bEnd.3 cells for 24 hours. Then bEnd.3 cells were washed with PBS for 3 times and trypsinized and seeded on a Sea-Horse® 96well XF-96 plate at a density of about 8,000 per well in XF base medium supplemented with 1 g/L glucose, 1 mM sodium

pyruvate, and 2 mM glutamine and were then placed into a SeaHorse XF Extracellular 24 Flux Analyzer, in order to measure their oxygen consumption rate (OCR). Mitochondrial respiration inhibitors -1.0 μM oligomycin, 1.0 μM carbonyl cyanid-4 phenylhydrazone (FCCP), 0.5 μM antimycin A and rotenonewere used to treat the cells in the system, and OCR was measured before and after treatment with the inhibitors, for determination of basal respiration, ATP production, maximal respiration, and spare respiratory capacity. All results were normalized to the number of cells per well, counted immediately after detection.

### Cell proliferation assay

The proliferation of bEnd.3 cells under various conditions was evaluated by CCK-8 assay (Meilunbio, Cat. MA0218). Cells were seeded on 96-well plate at 4000 cells per well for D-Sphingosine proliferation experiment, and 8000 cells per well for other experiments. After 24 h, 10% CCK-8 solution was added to each well for 3 h, and the absorbance value was measured at 450 nm by using a Multiscan Spectrum (Bio-Rad, MPM 6.0). Next, the supernatant was discarded, and each well was washed twice with PBS. The healthy cell group, damaged cell group (2 μM antimycin A and 2 μM rotenone) and damaged cell group that received mitochondrial transplantation were established. The CCK-8 assay was performed on days 3, 5, and 7.

### Cell scratching and wound-healing assay

bEnd.3 cells were seeded in 12-well plates and cultured in DMEM containing 10% FBS and 1% P/S until at least 90% confluence. Wounds were made with a sterile pipette tip, and then the dislodged cells were washed away with PBS. The medium was refreshed with DMEM containing 1% P/S. Wounds were photographed by microscopy (Nikon, Nikon Nis Elements 5.11) at 0 h and after 24 h. The rates of wound healing were measured by Fiji ImageJ (2.0.0-rc-69/1.52n). The percentage of healed area was calculated using the following formula: (1 - (Area of scratch at 24 hours / Area of scratch at 0 hours)) * 100%.

### Tube formation

bEnd.3 cells were seeded in 6-well plates and cultured for 24 h. Matrigel (Corning, Cat. 354234) was added to a precooled 96-well plate and incubated at 37 °C for 30 min to allow gel formation. bEnd.3 cells (approximately $1.2 \times 10^5$ cells/mL) were seeded in prepared 96-well plates and cultured for 12 h. Then, the cells were photographed by microscopy (Nikon), and tube formation in the images was quantified according to the branch points.

### Metabolic analysis (UHPLC–MS/MS analyses)

UHPLC–MS/MS analyses were performed using a Vanquish UHPLC system (Thermo Fisher Scientific) coupled with an Orbitrap Q ExactiveTM HF mass spectrometer (Thermo Fisher Scientific) in LCSW. Samples were injected onto a Hypesil Gold column (100×2.1 mm, 1.9μm) using a 12-min linear gradient at a flow rate of 0.2 mL/min. The eluents for the positive polarity mode were eluent A (0.1% FA in water) and eluent B (methanol). The eluents for the negative polarity mode were eluent A (5 mM ammonium acetate, pH 9.0) and eluent B (methanol). The solvent gradient was set as follows: 2% B, 1.5 min; 2-85% B, 3 min; 85-100% B, 10 min; 100-2% B, 10.1 min; 2% B, 12 min. A Q ExactiveTM HF-X mass spectrometer was operated in positive/negative polarity mode with a spray voltage of 3.5 kV, capillary temperature of 320 °C, sheath gas flow rate of 35 psi and aux gas flow rate of 10 L/min, S-lens RF level of 60, and Aux gas heater temperature of 350 °C.

The raw data files generated by UHPLC–MS/MS were processed using Compound Discoverer 3.1 (CD3.1, Thermo Fisher Scientific) to perform peak alignment, peak picking, and quantitation for each metabolite. The main parameters were set as follows: retention time tolerance, 0.2 minutes; actual mass tolerance, 5 ppm; signal intensity tolerance, 30%; signal/noise ratio, 3; and minimum intensity. After that,

peak intensities were normalized to the total spectral intensity. The normalized data were used to predict the molecular formula based on additive ions, molecular ion peaks and fragment ions. Then, peaks were matched with the mzCloud (https://www.mzcloud.org/), mzVault and MassList databases to obtain accurate qualitative and relative quantitative results. Statistical analyses were performed using the statistical software R (R version R3.4.3), Python (Python 2.7.6 version) and CentOS (CentOS release 6.6). When data were not normally distributed, normal transformations were attempted using the area normalization method.

These metabolites were annotated using the KEGG database (https://www.genome.jp/kegg/pathway.html), HMDB database (https://hmdb.ca/metabolites) and LIPIDMaps database (http://www.lipidmaps.org/). Principal component analysis (PCA) and partial least squares discriminant analysis (PLS-DA) were performed with metaX[61] (a flexible and comprehensive software for processing metabolomics data). We applied univariate analysis (t test) to calculate the statistical significance (P value). The metabolites with VIP > 1, P value < 0.05 and fold change (FC) ≥ 1.2 or FC ≤ 0.833 were considered to be differential metabolites. Volcano plots were used to filter metabolites of interest based on log2 (FoldChange) and -log10(p value) of metabolites by ggplot2 in R language. For clustering heatmaps, the data were normalized using z scores of the intensity areas of differential metabolites and were plotted by the Pheatmap package in R language. The correlation between differential metabolites was analyzed by cor () in R language (method = pearson). Statistically significant correlations between differential metabolites were calculated by cor.mtest () in R language. A P value < 0.05 was considered statistically significant, and correlation plots were plotted by the corrplot package in R language. The functions of these metabolites and metabolic pathways were studied using the KEGG database. The metabolic pathway enrichment of differential metabolites was performed. When the ratio was satisfied by x/n > y/N, metabolic pathways were considered enriched, and when the P value of metabolic pathways was <0.05, metabolic pathways were considered statistically significantly enriched.

### Femoral bone defect model and mitochondria injection

Femoral bone defect surgery was performed on 12-week-old male mice. Briefly, mice were anesthetized by inhalation of Isoflurane. A bone defect was created on the inner side of the right distal femur using a 0.6 mm-diameter round drill. The cortical bone was penetrated from the periosteum to the endosteum, followed by closure of the incision. Mitochondria injection was performed according to previous studies[30]. Briefly, mitochondria injections were administered on day 3 and day 5. For each injection, osteocyte mitochondria were isolated from 8 million MLO-Y4 cells for each mouse and resuspended in 50 μl PBS. The mitochondria were drawn using an insulin needle (1 ml, U-40, Braun, KDL0403008, China) and subperiosteally injected 1-2 mm above the defect area. Mice were sacrificed on day 7.

### Micro-CT analysis of bone defect area

Micro-CT analysis of femur bone defect was performed by using Skyscan 1176 (Bruker-MicroCT, Skycan 1.6.10.4). Scans were performed with a pixel size of 8.97 mm at 60 kV and 385 mA through a 0.5 mm aluminum filter. Cylindrical ROI was created to with 0.6 mm diameter at bone defect area for quantification. ROIs for analyses of femurs were obtained automatically using CTAn (Bruker-MicroCT). Histomorphometric analysis was performed using CTAn (Bruker-MicroCT, CTAn 1.16.1.0 +).

### Zoledronic acid treatment

4-week-old $Dmp1^{Cre}\text{-}DTA^{ki/wt}$ mice were treated with 100 μg/kg body mass zoledronic acid (MCE, Cat. HY-13777) in 100 μl PBS or pure 100 μl

PBS once per week for 2 weeks. All treatments were administered by intraperitoneal injection. After 2 weeks, mice were sacrificed for TRAP staining and immunofluorescence staining.

## Statistics and reproducibility

Data are presented as the mean ± standard error of the mean (SEM) except were stated otherwise. Graphs and statistics were analyzed by GraphPad Prism (version 9.1.1(223); GraphPad Software, San Diego, CA, USA). Microscopy images shown are representative of at least 3 independent experiments. Detailed data processing, sample size and statistical methods for each result were shown in the corresponding figure legends. A two tailed Student's t test was used for mean comparisons between two groups. Comparisons of multiple individual datasets were performed using oneway analysis of variance (ANOVA) with Dunnett's post hoc test or KruskalWallis test with Dunn's post hoc test in accordance with the normal or nonnormal distribution of data. $p < 0.05$ was considered as statistically significant.

## Reporting summary

Further information on research design is available in the Nature Portfolio Reporting Summary linked to this article.

## Data availability

Source data are provided with this paper, as indicated in the figure legends. Bulk RNA-seq datasets have been deposited in the Gene Expression Omnibus (GEO) database under accession codes GSE202356[14]. Metabolomics data have been deposited in the Metabolights[62] database under accession codes MTBLS7332. Source data are provided with this paper.

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

## Acknowledgements

This study was performed with the support of the National Natural Science Foundation of China (82002339, 81820108020, 12172224), Shanghai Frontiers Science Center of Degeneration and Regeneration in Skeletal System (BJ1-9000-22-4002), Shanghai Municipal Hospital Orthopedic Specialist Alliance, Shanghai Municipal Health Commission key priority discipline project; Shanghai Spinal Disease and Trauma Orthopedics Research Center (2022ZZ01014). We acknowledge Mr. Zihao Li and Ms. Lina Cao for their assistance in the cartonization of the schematic diagrams.

## Author contributions

J.J.G., M.H.Z., D.L.L. and C.Q.Z. provided the essential ideas and designed the experiments. P.L., L.C., H.Z., Z.M.C., J.Q.F., Y.Z., Y.Y.M., P.D., Y.D.P., S.Zhu., C.A.G., H.L., W.K.Z., J.Z., Jia.M, S.H. T, S.Zhang, M.Y., B.Q.W. and D.L.L. performed the research, Y.Z., W.G.Z., H.L., W.K.Z., P.D. provided suggestions on experiments. P.L., L.C., H.Z., D.L.L., Jio.M., G.Y.L. and Y.Q. analyzed the data. P.L., L.C. and H.Z. drafted the manuscript. J.J.G., D.L.L., C.Q.Z. and M.H.Z. revised the manuscript.

## Competing interests

The authors declare that they have no competing interests.
