## [Peer Review File · Nature Communications]

Osteocyte mitochondria regulate angiogenesis of transcortical vesselsREVIEWER COMMENTS

Reviewer #1 (Remarks to the Author):

In this study, the authors effectively quantified the contacts of osteocytes towards transcortical vessels (TCV), establishing intact osteocyte contact with TCV in the cortical bone of mice. They demonstrated, through the labelling of mitochondria, that the transfer of osteocyte mitochondria into TCV endothelial cells might occur *in vitro* and *in vivo*. The authors propose this as the main mechanism by which osteocytes alleviate endothelial dysfunction of TCV, substantiated by a series of experiments involving osteocyte-derived mitochondria on endothelial cells. Metabolomics then suggest that this mitochondria-derived compound improves TCV function.

This is an intriguing study, although I partially disagree with some of the exclusive functions attributed to mitochondrial transfer in the alleviation of TCV dysfunction. I believe additional experiments could help clarify this.

The effect of diphtheria toxin on the absolute number of osteocytes in bone was not shown in Fig. 2. Given the authors report on a partial ablation, this is of paramount importance. It could also be likely that osteocytes close to TCV would be more vulnerable to diphtheria exposure. The potential effects of osteoblast ablation, which could occur to a minor degree in cortical bone, should be assessed.

Does osteocyte ablation trigger increased bone remodeling by osteoclasts, and could their activity influence TCV branching? This could be functionally tested by inhibiting osteoclast activity with bisphosphonates while ablating DMP1 cre positive cells. The abundance of TRAP positive polynuclear cells should also be demonstrated.

In Fig. 3a, the authors elegantly demonstrate that Dendra2-labelled mitochondria appear in endothelial cells. The quantification is seen in Fig. 4C, showing that 2 percent of b.End3 cells received mitochondria. It would be highly interesting to understand the proportion of presumably transferred mitochondria compared to host mitochondria in these cells. Co-staining with mitotracker would be informative (partially done by the authors, but not quantified), as well as assessing branching, sphericity, and size of the mitochondria.

The authors demonstrate in Fig. 5b to c that endothelial cells, which received osteocyte-derived mitochondria, show a substantial increase in aspects of respiration as revealed by the Agilent Seahorse XF96 analyzer. This is impressive. However, co-cultures reveal that only a tiny fraction (2%) of endothelial cells receive mitochondria, suggesting that this may not have significant effects on TCVs, implying that other factors may play a role. Consequently, the authors report from their RNASeq that angiogenesis regulated genes are decreased in diphtheria exposed Dmp1Cre DTAwf/flox mice. Are these genes expressed in osteocytes? Do they act on TCV independently of mitochondrial transfer?

The authors should knock down some of these factors in MLOY4 cells and test their ability to influence endothelial cells to prove this point.

Just as for the *in vitro* experiment, the transfer of mitochondria from osteocytes should also be quantitatively shown *in vivo*.

It remains unclear whether the impairment of mitochondrial transfer and effects on TCV affects bone architecture. Therefore, the cortical bone phenotype of Dmp1cre Rhot1fl/fl mice should be assessed using ultra micro CT and electron microscopy.

The potential effect of D-Sphingosine is intriguing, but the resulting sphingosine 1 phosphate is also heavily involved in the activity of bone cells and in maintaining vascular integrity via S1P receptors. What is the TCV architecture after D-sphingosine treatment?

Reviewer #2 (Remarks to the Author):

In this manuscript, Liao, Chen, Zhou and colleagues aim to demonstrate that osteocyte-derived mitochondria play a critical role in regulating endothelial cell homeostasis and transcortical blood vessel maintenance in bone. This is a compelling hypothesis and model that is likely to be of high interest to many readers studying skeletal biology, the bone marrow microenvironment, and endothelial cells. The approach involves in vitro co-culture systems and mouse models with labeled mitochondria from Dmp1-expressing osteolineage cells (mature osteoblasts, osteocytes). Unfortunately, the critical in vivo evidence presented is limited (e.g. images of transplanted mitochondria in a single cell without quantification) and missing key controls (described below). Given the broad claims made, additional analysis is needed to improve the rigor of the data presented and the impact of the findings. Of the comments listed below, major points 3, 4, and 6 are most critical.

Major points:

1. Figure 2: The DTA model of osteocyte ablation is quite extreme, as many of the same authors presented in eLife in 2022. Are differences in osteoblasts, osteoclasts, bone marrow, and bone mass also present in the DTA mice here? Attributing the changes in the TCVs to loss of osteocytes alone (as on line 126, line 135) is likely an oversimplification in a model where several cell and tissue types are ultimately affected.
2. Figure 2: Do the authors mean to imply that angiogenic genes Vegfc, Slit3, Notch3, and Notch4 are expressed in osteocytes in the cortical bone tissue? Or are these genes differentially expressed only in the endothelial cells within homogenized cortical bone lysate? Can the authors perform in situ hybridization or a complementary in vitro model to validate that these genes are expressed in osteocytes?
3. Figure 3: The experiments presented in Figure 3, particularly 3c-d, are key to the paper's claims and require substantial further analysis to rigorously support that osteocyte-derived mitochondria transfer to endothelial cells. Please present quantification of the percent of endothelial cells containing Dendra2-labeled mitochondria across multiple biological replicates. How many Dendra2-labeled mitochondria does each endothelial cell contain? Images of Cre-negative control animals should be presented to demonstrate the specificity of the signal. The critical role of osteocytes would also be better supported if the authors labeled mitochondria with a different Cre promoter (e.g. in hematopoietic stem cells) to describe the cell-type specificity of mitochondria transfer from osteocytes to endothelial cells. Finally, the authors should investigate whether osteocytes only transfer mitochondria to endothelial cells within bone TCVs. Do other endothelial cells in the mouse outside bone also show labeling in this model?
4. Figure 4: What is the general osteocyte and bone phenotype in Dmp1-Cre ; Rhot1fl/fl mice? Are there effects on osteocyte morphology or apoptosis? Are changes in cortical or trabecular bone mass or remodeling noted? Can differences in TCV vascularization potentially be attributed to detrimental effects of Rhot1 deletion on osteoblasts, osteoclasts, bone mass, or bone marrow? Without data along these lines, it is difficult to understand the significance of the TCV phenotype that is reported.
5. Line 258-260: "Taken together, these results demonstrated that endothelial cells acquire mitochondria from osteocytes to alleviate their oxidative stress, improve energy production, and restore angiogenic capability." This wording suggests that endothelial cells preferentially acquire osteocyte mitochondria when under oxidative stress. To support this claim, the authors would need to 1) perform all mitochondrial transplant experiments in Figure 5 without A/R damage to demonstrate any uptake/effect of osteocyte mitochondria on endothelial cells not experiencing oxidative stress, 2) perform mitochondrial transplant experiments using mitochondria from a different cell type (e.g. from healthy endothelial cells) to determine whether these effects are specific to osteocytes, and 3) quantify the proportion of endothelial cells that contain transplanted mitochondria in each condition to determine whether acquisition is preferentially occurring in any condition.

6. Figure 6R-T: The relationship between TCV homeostasis and fracture healing is unclear. Do the authors propose that endothelial cells during fracture healing rely on mitochondria transferred from osteocytes? If so, the authors should provide histologic evidence that this is occurring. Seven days is rather short to evaluate bone mineralization after skeletal injury. It is strange that the authors present data related to fracture callus mineralization here without looking at the effects of Sph treatment on callus angiogenesis. What is the blood vessel phenotype in the Sph-treated mice within the fracture site? Substantial further work is needed to understand how this study fits with the rest of the manuscript.

Minor points:

1. Figure 1C-I: How many independent mice and how many osteocytes per mouse are represented in these measurements?

2. Line 105-106: "In detail, 60.27% of the total dendrites were distributed on the vessel nearside, among which 74.2% were in contact with endothelial cells". These numbers do not match the data presented in Figure 1F (59.6% vessel nearside / 70.35% contact) and Figure 1H (approximately 25% contact to vessel). If panel H is showing detail from panel F, it may be helpful to color it black and grey.

3. Line 185-186: Does "digesting the cocultured cells" mean that cells were fixed and permeabilized for flow cytometry experiments? Please explain the sample preparation for flow cytometry (Figure 4C) in the methods. If cells were not fixed, a live-dead stain would be beneficial. Further, a control condition in which bEnd3 cells are co-cultured with Mloy4 cells without Dendra2-labeled mitochondria is important for gating and quantifying which endothelial cells are Dendra2-positive.

4. Figure 4F: Please describe the quantification strategy. Is the number of mitochondria measured per cell or within a different unit? What does one point on the graph represent? Please ensure multiple biological replicates are measured and presented.

5. Line 251-252 and Line 268 both refer to endothelial cells 'acquiring' mitochondria in experiments where endothelial cells have been co-cultured with mitochondria. Judging by Figure 4C, this would be approximately 2% of cells causing a bulk phenotype. Again, it is important to show what percentage of cells took up the transplanted mitochondria.

6. Figure 6: Can the authors comment on why Sph appears to become more effective with higher concentration for some outcomes (Figure 6D,J), but also saturate at 12.5 nM (Figure 6F), and reverse its effect with high concentration (Figure 6H). This inconsistency would suggest that Sph is not the only factor that matters here.

Reviewer #3 (Remarks to the Author):

Liao et al present a fascinating manuscript detailing a novel mechanism whereby osteocytes regulate angiogenesis and transcortical vessel (TCV) formation by transferring mitochondria that contain sphingosine. The authors provide a fascinating description of the TCV network between osteocytes and endothelial cells (ECs) and demonstrate that depletion of osteocytes leads to the formation of sparse, disorganized TCV network. In co-culture systems using mitochondria reporter osteocytes, they show that osteocytes transfer mitochondria to bEnd.3 cells in vitro. This process occurs in vivo as well and was impaired in the absence of MIRO1 in osteocytes, identifying that this protein (which is known to shuttle mitochondrial along tunnelling nanotubes) mediates mitochondria transfer from osteocytes to ECs. Deletion of MIRO1 in osteocytes was also associated with TCV regression in vivo. Adding purified mitochondria from osteocytes onto bEnd.3 cells rescued A/R-induced defects in cell counts, branch

point formation, and migration. The mitochondria-treated cells have more sphingosine (Sph) and sphingosine-1-phosphate (S1P), so the authors treated bEnd.3 cells stressed with A/R with or without Sph and observed that Sph also rescued cell counts, branch point formation, and migration. Mitochondria-mediated rescue of A/R-stressed bEnd.3 cells was blocked by PF543, suggesting that this process is dependent on the formation of S1P. This manuscript is rigorous and uses the ideal strategies to track mitochondria transfer in vitro and in vivo. Conceptually, this manuscript increases our understanding of how mitochondria transfer regulates angiogenic responses through production of sphingolipids and reveals a previously uncharacterized mechanism of mitochondria transfer along TCVs. While I am enthusiastic about this manuscript, I do have several major concerns:

Major:

1. My biggest conceptual concern is that the mechanism of mitochondria transfer from osteocytes to ECs is along TCVs in a MIRO1-dependent manner, a lot like tunnelling nanotubes (TNTs). With TNTs, the mitochondria are deposited directly into the cytoplasm. However the in vitro studies in Fig 5-6 utilize exogenous mitochondria purified from osteocytes and then simply add them to the ECs in culture. Uptake of free mitochondria is a very different transfer mechanism than delivery via a tubular structure such as the TCV. I am having a hard time understanding this. Are the TCVs making contact with the ECs and then ejecting free mitochondria at the EC surface for capture? Or are they depositing the mitochondria directly into the EC's cytoplasm?
2. What is the gating strategy for Fig 3c?
3. Statistics are needed for Fig 5b.
4. Does exogenous Sph treatment affect mitochondria transfer from osteocytes to ECs?
5. Fig 6m-q: another group is required, specifically 4/4 A/R + PF543 (no mito)
6. Does administration of mitochondria improve bone healing in the surgical model?
7. The authors have not really distinguished whether D-sphingosine or S1P are responsible for the phenotypes observed in ECs. Their PF543 data suggest that perhaps S1P might be the responsible factor. The authors should clarify this experimentally, or they should temper the conclusions and text to more accurately state that mitochondria transfer delivers sphingolipids that promote EC responses, rather than making such specific claims about D-sphingolipid.

Minor:

1. Fig 6h: why does 12.5 nM Sph restore branch points after A/R but 50nM does not?
2. Fig 5d and Fig 6c are not very convincing and underlies the data in Fig 5e and 6d, respectively. I am not sure what value the ROS data add here given that the functional readouts of cell counts and branch points are more prominent and relevant.
3. The discussion would be improved if the authors could speculate about why is sphingosine delivered to ECs via mitochondria transfer.

REVIEWER COMMENTS

**Reviewer #1 (Remarks to the Author):**

In this study, the authors effectively quantified the contacts of osteocytes towards transcortical
vessels (TCV), establishing intact osteocyte contact with TCV in the cortical bone of mice. They
demonstrated, through the labelling of mitochondria, that the transfer of osteocyte mitochondria
into TCV endothelial cells might occur in vitro and in vivo. The authors propose this as the main
mechanism by which osteocytes alleviate endothelial dysfunction of TCV, substantiated by a series
of experiments involving osteocyte-derived mitochondria on endothelial cells. Metabolomics then
suggest that this mitochondria-derived compound improves TCV function. This is an intriguing
study, although I partially disagree with some of the exclusive functions attributed to
mitochondrial transfer in the alleviation of TCV dysfunction. I believe additional experiments
could help clarify this.

**Response to Reviewer 1:**

*Thank you for your constructive comments. We have made revision as according to the suggestion.*
*We have addressed each point using subtitle as indicated below.*

**Reviewer #1 comments 1:** The effect of diphtheria toxin on the absolute number of osteocytes in
bone was not shown in Fig. 2. Given the authors report on a partial ablation, this is of paramount
importance. It could also be likely that osteocytes close to TCV would be more vulnerable to
diphtheria exposure. The potential effects of osteoblast ablation, which could occur to a minor
degree in cortical bone, should be assessed.

**Response to Reviewer #1 comments 1:**

*1) The effect of diphtheria toxin on the absolute number of osteocytes in bone was not shown in*
*Fig. 2 - We have conducted the immunofluorescence staining on 4-week-old $DTA^{ki/wt}$ and $Dmp1^{Cre}$ -*
*$DTA^{ki/wt}$ mice (Supplementary Fig. 1a). The quantified results revealed the absolute number of*
*osteocytes was decreased in $Dmp1^{Cre}$ - $DTA^{ki/wt}$ (Supplementary Fig. 1b).*

*2) The potential effects of osteoblast ablation, which could occur to a minor degree in cortical*
*bone, should be assessed - Yes, in our previous study¹, we found osteoblasts were secondarily*
*decreased as a results of osteocytes ablation in trabecular bone of $Dmp1^{Cre}$ - $DTA^{ki/wt}$ mice.*
*According to your suggestion, we investigated the osteoblast alteration in cortical bone by*
*performing ALP immunohistochemistry staining on 4-week-old $Dmp1^{Cre}$ - $DTA^{ki/wt}$ mice and control*
*$DTA^{ki/wt}$ mice (Supplementary Fig. 1c). In line with the results obtained from trabecular bone in*
*our previous study¹, the ALP staining indicated a decrease in osteoblasts number in the cortical*
*bone of $Dmp1^{Cre}$ - $DTA^{ki/wt}$ mice (Supplementary Fig. 1d). However, in the $Dmp1^{Cre}$ - $Rhot1^{fl/fl}$ mice,*
*which TCVs formation was impaired by inhibition of osteocyte mitochondrial transfer, osteoblasts*
*number were unaffected supported by the unchanged ALP staining results in cortical bone*
*(Supplementary Fig. 5a-b), indicating the crucial role of osteocyte mitochondria in the regulation*
*of the TCVs network.*

*We have included the information in the revised manuscript and discussion. See line 114-117, line*
*143-146 and line 512-515 in the main text.*

*(Line 114-117) “Partial ablation of osteocytes was evidenced by the immunofluorescence staining*
*of cortical bone. Quantification analysis showed that the absolute number of osteocytes were*
*significantly decreased in $Dmp1^{Cre}$ - $DTA^{ki/wt}$ mice in comparison with control $DTA^{ki/wt}$ mice.*
*(Supplementary Fig. 1a-b).”*

(Line 143-146) “Using tartrate-resistant acid phosphatase (TRAP) and alkaline phosphatase
 (ALP) staining on $Dmp1^{Cre}-DTA^{ki/wt}$ mice cortical bone, we showed that ablation of osteocytes in
 cortical bone causes a significant increase of osteoclast number (Supplementary Fig. 1c-d) and
 decrease of osteoblast number (Supplementary Fig. 1e-f).”
 (Line 512-515) “A limitation of our study is the use of $Dmp1^{Cre}-DTA^{ki/wt}$ mice as they may not be
 exclusive to osteocyte. Nevertheless, our comprehensive approach of in vitro assessment and the
 use of $Dmp1^{Cre}-Rhot1^{fl/fl}$ mice and $PF4^{Cre}-Cox8^{Dendra2}$ mice enable to demonstrate the specificity
 of osteocyte mitochondrial transfer in maintaining TCV networks.”

 **Supplementary Fig. 1. (a-b) Representative confocal images of femur cortical bone (a) and**
 **quantitative result of osteocyte number (b) from 4-week-old control $DTA^{ki/wt}$ mice and $Dmp1^{Cre}-$**
 **$DTA^{ki/wt}$ mice. $n=3$, scale bars, 100µm. (c-d) Representative ALP staining images of femur cortical**
 **bone (c) and quantitative result of ALP positive cell number (d) from 4-week-old control $DTA^{ki/wt}$**
 **mice and $Dmp1^{Cre}-DTA^{ki/wt}$ mice. $n=3$, scale bars, 100µm, yellow arrows represent ALP positive**
 **cells.**

**Supplementary Fig. 5. (a-b) Representative ALP staining images of femur cortical bone (a) and**
 **quantitative result of ALP positive cell number (b) from 6-week-old control $Rhot1^{fl/fl}$ and $Dmp1^{Cre}-$**
 **$Rhot1^{fl/fl}$ mice. $n=3$, scale bars, 100µm, yellow arrows represent ALP positive cells.**

 **Reviewer #1 comments 2: Does osteocyte ablation trigger increased bone remodeling by**
 **osteoclasts, and could their activity influence TCV branching? This could be functionally tested**
 **by inhibiting osteoclast activity with bisphosphonates while ablating DMP1 cre positive cells. The**
 **abundance of TRAP positive polynuclear cells should also be demonstrated.**

**Response to Reviewer #1 comments 2:**

**1) Does osteocyte ablation trigger increased bone remodeling by osteoclasts? - Yes, our previous**
 **study showed that the osteocytes ablation upregulates osteoclasts lineage and function in**
 **trabecular bone and bone marrow¹. As per your suggestion, we investigated the osteoclasts**
 **alteration in the cortical bone of $Dmp1^{Cre}-DTA^{ki/wt}$. The TRAP staining results revealed the number**
 **of cortical bone osteoclasts also increased in the $Dmp1^{Cre}-DTA^{ki/wt}$ mice in comparison with the**
 **control group (Supplementary Fig. 1e-f).**

2) Could the activity of osteoclasts influence TCV branching? - Not in the context of osteocyte-
 ablation. It has been reported that osteoclasts promote TCVs formation in TNF- α overexpressed
 mice². However, it is not clear whether osteoclasts can also induce TCV formation in mice with
 osteocyte-ablation. Thus, we administered zoledronic acid, a specific type of bisphosphonate, to
 4-week-old $Dmp1^{Cre}DTA^{ki/wt}$ mice to suppress osteoclast activity. After 2 weeks, the lower limb
 was collected for osteoclast and TCVs assessment. The TRAP staining revealed the zoledronic
 acid injection decreased the mice osteoclasts number in cortical bone in comparison with the PBS
 injection mice (Supplementary Fig. 1g-h). Further, the CD31 immunofluorescence staining
 revealed neither the TCVs number nor the TCVs branches points have a significant change
 (Supplementary Fig. 1i-j). These results indicated the absence of osteocytes was a stronger adverse
 condition for the TCVs formation compared to the increase of osteoclasts.
 We have included the data in the revised manuscript. See line 139-153 in the main text.
 (Line 139-153) "As it has been reported that osteoclastic activity plays a role in mediating TCVs
 formation³, to answer if osteocyte ablation induced TCV regression is mediated by the induction
 of osteoclast activities in $Dmp1^{Cre}-DTA^{ki/wt}$ mice. We performed zoledronic acid injection to inhibit
 the activities of osteoclasts in 4-week-old $Dmp1^{Cre}-DTA^{ki/wt}$ mice. Lower limbs were collected for
 osteoclast staining and TCVs assessment. Using tartrate-resistant acid phosphatase (TRAP) and
 alkaline phosphatase (ALP) staining on $Dmp1^{Cre}-DTA^{ki/wt}$ mice cortical bone, we showed that
 ablation of osteocytes in cortical bone causes a significant increase of osteoclast number
 (Supplementary Fig. 1c-d) and decrease of osteoblast number (Supplementary Fig. 1e-f).
 Administration of zoledronic acid for 2 weeks revealed decrease of osteoclast number in cortical
 bone in comparison with the PBS injection mice (Supplementary Fig. 1g-h). The CD31
 immunofluorescence staining showed that zoledronic acid does not alter the TCVs number nor the
 TCVs branches points as compared to the control (Supplementary Fig. 1i-j). These results
 indicated that the decreased TCVs when osteocytes were ablated is not due to the induction of
 osteoclast activities. Together, these results showed that osteocytes regulate the vascularization of
 TCVs directly in the manner that is independent of osteoclasts."

 *Supplementary Fig. 1. (e-f) Representative TRAP staining images of femur cortical bone (e) and*
 *quantitative result of TRAP positive cell number (f) from 4-week-old control $DTA^{ki/wt}$ mice and*
 *$Dmp1^{Cre}-DTA^{ki/wt}$ mice. n=3, scale bars, 100 μ m, yellow arrows represent TRAP positive cells. (g-*

*h) Representative TRAP staining images of femur cortical bone (g) and quantification of TRAP*
*positive cell number (h) from PBS or zoledronic acid (Zole) treated 4-week-old Dmp1^{Cre}-*
*DTA^{ki/wt} mice. n=3, scale bars, 100µm, yellow arrows represent TRAP positive cells. (i-j)*
*Representative confocal images of femur cortical bone (i) and quantitative result (j) of TCVs from*
*PBS or zoledronic acid treated 4-week-old Dmp1^{Cre}-DTA^{ki/wt} mice. n=3, scale bars, 100µm. (k)*
*RT-qPCR analysis of overlapped genes in bEnd.3 cells and MLO-Y4 cells.*

**Reviewer #1 comments 3:** In Fig. 3a, the authors elegantly demonstrate that Dendra2-labelled
mitochondria appear in endothelial cells. The quantification is seen in Fig. 4C, showing that 2
116 percent of b.End3 cells received mitochondria. It would be highly interesting to understand the
117 proportion of presumably transferred mitochondria compared to host mitochondria in these cells.
Co-staining with mitotracker would be informative (partially done by the authors, but not
quantified), as well as assessing branching, sphericity, and size of the mitochondria.

**Response to Reviewer #1 comments 3:**

*Thank you for the comment. We have further evaluated the proportion and characteristics of*
*mitochondria by confocal on bEnd.3 cells following a 48-hour coculture with MLO-Y4^{Mito-Dendra2}*
*cells (Supplementary Fig. 3a).*

*1) We showed proportion of transferred mitochondria is a relatively smaller fraction of the total*
*mitochondrial population within the recipient cell, approximately 3.16% at 48 hours co-culture*
*(Supplementary Fig. 3b).*

*2) Characterization of the transferred mitochondria showed that they exhibited a smaller length*
*compared to the host mitochondria, while their width remained similar (Supplementary Fig. 3c).*
*Specifically, the average width of the transferred mitochondria, measuring 1.23±0.09µm,*
*resembled that of the host mitochondria (1.04±0.03µm). Conversely, the average length of the*
*transferred mitochondria, at 1.46±0.10µm, was smaller than that of the host mitochondria*
*(2.09±0.04µm).*

*3) The sphericity of the transferred mitochondria, the ratio of length to width was also calculated,*
*yielding a value of 1.22±0.06. This value is notably smaller than the host mitochondria's ratio of*
*2.09±0.06, thereby suggesting that the transferred mitochondria possess a more spherical*
*morphology (Supplementary Fig. 3c).*

*We have included the data in the revised manuscript. See line 211-223 in the main text.*

*(Line 211-223) “Next, we evaluated the proportion and characteristics of mitochondria by*
*confocal on bEnd.3 cells following a 48-hour coculture with MLO-Y4^{Mito-Dendra2} cells*
*(Supplementary Fig. 3a). We showed proportion of transferred mitochondria is a relatively*
*smaller fraction of the total mitochondrial population (host mitochondria) within the recipient cell,*
*approximately 3.16% at 48 hours co-culture (Supplementary Fig. 3b). Characterization of the*
*transferred mitochondria showed that they exhibited a smaller length compared to the host*
*mitochondria, while their width remained similar (Supplementary Fig. 3c). Specifically, the*
*average width of the transferred mitochondria, measuring 1.23µm, resembled that of the host*
*mitochondria (1.04µm). Conversely, the average length of the transferred mitochondria, at*
*1.46µm, was smaller than that of the host mitochondria (2.09µm). The sphericity of the transferred*
*mitochondria, the average ratio of length to width was also calculated, yielding a value of 1.22.*
*This value is notably smaller than the host mitochondria's ratio of 2.09, thereby suggesting that*
*the transferred mitochondria possess a more spherical morphology (Supplementary Fig. 3c).”*

 *Supplementary Fig. 3. (a) Representative confocal images of bEnd.3 cells cocultured with MLO-*
 *Y4^{Mito-Dendra2} with number ratio at 3:1 (MLO-Y4: bEnd.3) for 48 hours. Mitochondria were stained*
 *with Mitotracker Red CMXRos (MTR), scale bars, 20μm, white arrow represents MLO-Y4-derived*
 *mitochondria (transferred mitochondria). (b) Quantitative result of number ratio of bEnd.3*
 *mitochondria (host mito) and MLO-Y4-derived mitochondria (transferred mitochondria) in*
 *bEnd.3 cells as shown in (a). n=9. (c) Quantitative results of morphology of host mitochondria*
 *and transferred mitochondria as shown in (a) n=9.*

**Reviewer #1 comments 4:** The authors demonstrate in Fig. 5b to c that endothelial cells, which
 received osteocyte-derived mitochondria, show a substantial increase in aspects of respiration as
 revealed by the Agilent Seahorse XF96 analyzer. This is impressive. However, co-cultures reveal
 that only a tiny fraction (2%) of endothelial cells receive mitochondria, suggesting that this may
 not have significant effects on TCVs, implying that other factors may play a role. Consequently,
 the authors report from their RNASeq that angiogenesis regulated genes are decreased in
 diphtheria exposed Dmp1Cre DTAwt/flox mice. Are these genes expressed in osteocytes? Do they
 act on TCV independently of mitochondrial transfer? The authors should knock down some of
 these factors in MLOY4 cells and test their ability to influence endothelial cells to prove this point.

**Response to Reviewer #1 comments 4:**

*1) 2% fraction of endothelial cells does not have impact on TCV - Transfer of mitochondria is time*
 *dependent. The structural change of TCV in vivo would be the result of accumulative effect from*
 *a small proportion of mitochondria transfer over time. The 2% fraction of endothelial cells*
 *receiving mitochondria is an in vitro snap-shot recorded at 24 hours co-culture. The study was for*
 *the purpose of elucidating the impact of miro1 knockdown on mitochondrial transfer. Considering*
 *the relatively short duration and low ratio of donor/recipient cells of the co-culture system (24*
 *hours and 1:1, respectively), it is not surprised that we observe a low percent of mitochondrial*
 *transfer between the two cells in our in vitro experiment. To further explore whether the percentage*
 *of endothelial cells that received mitochondria would increase with prolonged co-culture time and*
 *higher number ratio of donor/recipient cells, bEnd.3 cells were cocultured with MLO-Y4^{Mito-Dendra2}*
 *for 24 hour, 48 hours and 72 hours and 7 days in vitro with the number ratio of MLO-Y4^{Mito-Dendra2}*
 */bEnd.3 at 1:1 and 3:1, respectively. The flow cytometer analysis of the co-culture system revealed*
 *that the fraction of endothelial cells that received the mitochondria increased with extended*
 *duration of co-culture and higher ratio of donor/recipient cells (Supplementary Fig. 2a-d). The*
 *peak rate of 10% mitochondria-acquired endothelial cells occurs at 3:1 and 72 hours of co-culture*
 *(Supplementary Fig. 2c). Given that osteocytes and endothelial cells have extended lifespans in*
 *vivo^{4,5}, and osteocytes is the major cells resident in cortical bone⁶, these two cell types can coexist*
 *for a long time in vivo. Thus, there may be higher ratio of transferred osteocyte mitochondria in*
 *endothelial cells of TCVs in vivo. In support of this, we identified 48 TCVs from 4 Dmp1^{Cre-}*
 *Cox8^{Dendra2} mice, and we found 79.95% of TCVs received osteocyte mitochondria (Fig. 3h). We*
 *have updated the data in the revised manuscript. See line 194-203 and line 230-231 in the main*
 *text.*

(Line 194-203) “To further explore whether the percentage of endothelial cells that received
 mitochondria would increase with prolonged co-culture time and higher number ratio of
 donor/recipient cells, bEnd.3 cells were cocultured with MLO-Y4^{Mito-Dendra2} for 24 hours, 48 hours
 and 72 hours and 7 days in vitro with the number ratio of MLO-Y4^{Mito-Dendra2} /bEnd.3 at 1:1 and
 3:1, respectively. The flow cytometry analysis of the co-culture system revealed that the fraction
 of endothelial cells that received the mitochondrial increased with extended duration of co-culture
 time and higher number ratio of donor/recipient cells (Supplementary Fig. 2a-d). The peak rate of
 10% mitochondria-acquired endothelial cells occurs at 3:1 and 72 hours of co-culture
 (Supplementary Fig. 2b-c). These data supported transfer of mitochondria is time and donor-
 recipient number ratio dependent.”

(Line 230-231) “Quantification of cortical bones revealed that 79.95% of TCV fractions contain
 mitochondria from osteocytes (Fig. 3h).”

 **Supplementary Fig. 2.** (a) Gate strategy to identify Dendra2⁺ bEnd.3 cells cocultured with MLO-
 Y4^{Mito-Dendra2} with number ratio at 1:1 and 3:1 (MLO-Y4: bEnd.3) for 24 hours, 48 hours and 72
 206 hours and 7 days. (b) Representative dot-plots and quantitative result of the percentage of bEnd.3
 cells acquired with Mito-Dendra2 fluorescence in entire bEnd.3 cells population after coculturing
 with MLO-Y4^{Mito-Dendra2} for 24 hours with number ratio at 1:1 and for 7 days with number ratio at

3:1 (MLO-Y4: bEnd.3). n=6. (c-d) Line plot (c) and quantitative result (d) of bEnd.3 cells acquired
with Mito-Dendra2 fluorescence in entire bEnd.3 cells population cocultured with MLO-Y4^{Mito-}
211 ^{Dendra2} with number ratio at 1:1 and 3:1 (MLO-Y4: bEnd.3) for 24 hours, 48 hours and 72 hours
and 7 days.

Fig. 3. (h) Quantitative result of TCVs that acquired with Mito-Dendra2 fluorescence in femur
cortical bone from 4-6-week-old *Dmp1^{Cre}-Cox8a^{Dendra2}*. n=4.

2) Are angiogenesis gene (*Slit3*, *Notch3*, *Notch4*, *Vegfc*) expressed in osteocytes? - We performed
RT-qPCR of bEnd.3 and MLO-Y4. The results revealed that the endothelial cells and osteocytes
express similar level of *Slit3*, *Notch3* and *Vegfc*, while the expression of *Notch4* is minimum in
osteocytes (Supplementary Fig. 1k). We have updated the data in the revised manuscript. See line
166-169 in the main text.

(Line 166-169) “To investigate the source of four genes, we performed qPCR analysis on the
MLO-Y4 osteocytes cell line and bEnd.3 endothelial cell line. The results showed the endothelial
cells and osteocytes express similar level of *Slit3*, *Notch3* and *Vegfc*, while the expression of
*Notch4* is relatively low in osteocytes (Supplementary Fig. 1k).”

Supplementary Fig. 1 (k) RT-qPCR analysis of overlapped genes in bEnd.3 cells and MLO-Y4 cells.

3) Do they act on TCV independently of mitochondrial transfer? The authors should knock down
some of these factors in MLOY4 cells and test their ability to influence endothelial cells to prove
this point. - We transfected MLO-Y4 cells with small interfering RNA (siRNA) of *Vegfc*, *Slit3*,
*Notch3*, *Notch4* to knock down expression of target genes individually. The efficiency of knocking
down expression of target genes were validated by RT-qPCR (Supplementary Fig.1l). To
investigate if there is a second mechanism, other than mitochondria transfer in osteocyte regulated
TCV, we tested the role of MLO-Y4 cells conditioned media on bEnd.3 by coculture for 24 hours
and performed tube formation assay and wound healing assay on bEnd.3 cells (Supplementary
Fig. 1m, o). The result showed that the conditional media from transfected MLO-Y4 cells did not
change the tube formation and migration ability of bEnd.3 cells (Supplementary Fig.1n, p), which
contrasts with the role of osteocyte mitochondria transfer (Fig. 5). We thus concluded that
osteocyte *Vegfc*, *Slit3*, *Notch3*, *Notch3* may not play a strong role in TCV formation as osteocytes
mitochondria. We have updated the data in the revised manuscript. See line 169-179 in the main
text.

(Line 169-179) “Next, we wondered whether osteocyte regulating endothelial cells angiogenesis
is linked to these overlapped genes. We transfected MLO-Y4 cells with small interfering RNA
(siRNA) of *Vegfc*, *Slit3*, *Notch3*, *Notch4* to knock down expression of target genes individually.
The efficiency of knocking down expression of target genes were validated by qPCR
(Supplementary Fig.1l). Then, we tested the role of MLO-Y4 cells conditioned media on bEnd.3
tube formation and wound healing assay (Supplementary Fig. 1m, o). The result showed that the
conditional media obtained from the MLO-Y4 cell lines with siRNA of *Vegfc*, *Slit3*, *Notch3*, *Notch4*
do not significantly alter the tube formation and migration ability of bEnd.3 cells (Supplementary

Fig.1n, p), suggesting the weak link between *Vegfc*, *Slit3*, *Notch3*, *Notch4* and the direct regulation of osteocytes on angiogenesis of endothelial cells.”

Supplementary Fig.1 (l) RT-qPCR analysis of targeted gene expression level in *Vegfc*^{KD}-MLO-Y4, *Slit3*^{KD}-MLO-Y4, *Notch3*^{KD}-MLO-Y4, *Notch4*^{KD}-MLO-Y4. n=4, NC, negative control. (m-n) Representative images of tube formation assay (m) and quantitative results (n) of number of branches point of bEnd.3 cells treated with NC-MLO-Y4, *Vegfc*^{KD}-MLO-Y4, *Slit3*^{KD}-MLO-Y4, *Notch3*^{KD}-MLO-Y4, *Notch4*^{KD}-MLO-Y4 conditional media (CM), respectively. n=3. scale bars, 500µm. (o-p) Representative images of wound healing assay (o) and quantitative result (p) of percentage of healing area of bEnd.3 cells treated with NC-MLO-Y4, *Vegfc*^{KD}-MLO-Y4, *Slit3*^{KD}-MLO-Y4, *Notch3*^{KD}-MLO-Y4, *Notch4*^{KD}-MLO-Y4 conditional media (CM), respectively. n=3, scale bars, 500µm.

Reviewer #1 comments 5: Just as for the in vitro experiment, the transfer of mitochondria from osteocytes should also be quantitatively shown in vivo.

Response to Reviewer #1 comments 5:

Yes, this is a good suggestion. We had quantified the transfer of mitochondria in vivo in our first submission manuscript (Fig. 4e-f). We quantified 12 confocal images of femur cortical bone, encompassing 26 TCVs of 3 mice, from both *Rhot1*^{fl/fl} mice and *Dmp1*^{Cre}-*Rhot1*^{fl/fl} mice, with each group consisting of 3 mice (Fig. 4e). Next, the number of osteocyte-derived mitochondria in each TCV were measured and compared (Fig. 4f). The results showed the number of transferred mitochondria in each TCVs was decreased in *Dmp1*^{Cre}-*Cox8*^{Dendra2}-*Rhot1*^{fl/fl} mice (Mean±SEM,

4.481±0.5185) compared with the $Cox8^{Dendra2}-Rhot1^{fl/fl}$ (Mean±SEM, 13.15±2.503), supporting the
MIRO1 facilitates the intercellular movement of mitochondria. As suggested, we further quantified
the area of osteocyte-transferred mitochondria in TCVs in vivo. The results showed that the ratio
of transferred mitochondria area to TCVs area was decreased in $Dmp1^{Cre}-Cox8^{Dendra2}-Rhot1^{fl/fl}$
mice compared with $Dmp1^{Cre}-Cox8^{Dendra2}$ mice (Supplementary Fig. 4d). The results were
consistent with the data presented in Fig. 4f. Our findings provide evidence that the transfer of
mitochondria from osteocytes to endothelial cells is common in vivo, and that this process is
mediated by MIRO1.

We have integrated the data in the revised manuscript. See line 262-270 in the main text.

(Line 262-270) “Next, to examine the role of MIRO1 in mediating mitochondrial transfer in vivo,
the $Dmp1^{Cre}-Cox8^{Dendra2}-Rhot1^{fl/fl}$ mouse lineage was established, in which *Rhot1* was specifically
deleted in *Mito*^{Dendra2}-labeled osteocytes (Fig. 4d). Confocal imaging and quantification of mouse
femur cortical bone revealed a significantly decreased transferred mitochondria number in each
TCV of $Dmp1^{Cre}-Cox8^{Dendra2}-Rhot1^{fl/fl}$ mice compared to those of $Dmp1^{Cre}-Cox8^{Dendra2}$ control mice
(Fig. 4e-f). Moreover, the ratio of transferred mitochondria area to TCVs area was decreased in
$Dmp1^{Cre}-Cox8^{Dendra2}-Rhot1^{fl/fl}$ mice compared with $Dmp1^{Cre}-Cox8^{Dendra2}$ mice (Supplementary Fig.
4d). These results implied the critical effect of MIRO1 in mediating mitochondrial transfer from
osteocytes to endothelial cells.”

Supplementary Fig. 4. (d) Quantitative result of area of Mito-Dendra2 fluorescence to area of
CD31 in femur cortical bone from 6-week-old $Dmp1^{Cre}-Cox8^{Dendra2}$ and $Dmp1^{Cre}-Cox8^{Dendra2}-$
$Rhot1^{fl/fl}$. n=3

**Reviewer #1 comments 6:** It remains unclear whether the impairment of mitochondrial transfer
and effects on TCV affects bone architecture. Therefore, the cortical bone phenotype of $Dmp1^{Cre}$
$Rhot1^{fl/fl}$ mice should be assessed using ultra micro-CT and electron microscopy.

**Response to Reviewer #1 comments 6:**

To evacuate the bone phenotype of $Dmp1^{Cre}-Rhot1^{fl/fl}$ mice, we performed the micro-CT analysis
on the femur from control 6-week-old $Rhot1^{fl/fl}$ and $Dmp1^{Cre}-Rhot1^{fl/fl}$ mice and calculated classical
bone mass indexes including bone volume (BV), BV/TV, trabecular separation (Tb.Sp), bone
mineral density (BMD), cortical bone area (Ct.Ar), cortical thickness (Ct.Th), tissue mineral
density (TMD) (Supplementary Fig. 5k-l). Our analysis revealed no significant difference in bone
phenotype between the two groups (Supplementary Fig. 5k-l). To access the bone architecture, we
conducted SEM on control 3 $Rhot1^{fl/fl}$ and 3 $Dmp1^{Cre}-Rhot1^{fl/fl}$ samples (Supplementary Fig. 5g).
The result showed that the knocking out of *Rhot1* in osteocytes did not alter the number of lacunae
in cortical bone (Supplementary Fig. 5h). Together these results suggested that the *Rhot1* depletion
did not significantly affect the cortical bone architecture.

We have updated the data in the revised manuscript. See line 297-298 and 300-306 in the main
text.

(Line 297-298) “Consistently, no significant changes of lacunae in cortical bone were observed
in the SEMs analysis (Supplementary Fig. 5g-h).”

(Line 300-306) “Moreover, the bone mass remains unaffected after knocking out Rhot1 in
osteocytes, as supported by obvious changes in trabecular bone volume (BV), BV/TV, trabecular
separation (Tb.Sp), bone mineral density (BMD), cortical bone area (Ct.Ar), cortical thickness
(Ct.Th), tissue mineral density (TMD) (Supplementary Fig 5k-l). Together, these data showed the
depletion of Rhot1 in cortical bone do not cause a significant change in bone cell components and
bone mass and supported that MIRO1-mediated mitochondrial transfer from osteocytes to
endothelial cells in cortical bone is critical for maintaining the homeostasis of TCV
vascularization.”

Supplementary Fig. 5 (g-h) Representative SEM images of femur cortical bone (g) and quantitative
result of lacunae number (h) from 6-week-old control Rhot1^{fl/fl} and Dmp1^{Cre}-Rhot1^{fl/fl} mice. n=3,
scale bars, 20µm. (k-l) Representative micro-CT images of femur (k) and quantitative result of
bone mass (l) from 6-week-old control Rhot1^{fl/fl} and Dmp1^{Cre}-Rhot1^{fl/fl} mice. n=4.

**Reviewer #1 comments 7:** The potential effect of D-Sphingosine is intriguing, but the resulting
sphingosine 1 phosphate is also heavily involved in the activity of bone cells and in maintaining
vascular integrity via S1P receptors. What is the TCV architecture after D-sphingosine treatment?

**Response to Reviewer #1 comments 7:**

We have performed a 1µm-resolution micro-CT scanning on the femur cortical bone defect healing
area from the mice administered with saline or D-sphingosine. Data on the 3D-reconstructed
canal systems showed an increased canal component in D-sphingosine treatment group,
suggesting a higher ability of TCVs formation (Supplementary Fig. 10a). To further investigate
the TCVs alteration in these two groups, histological sections and immunofluorescence staining
were utilized on the cortical bone defect area to compare TCVs architecture (Supplementary Fig.
10b). The results showed that numbers of blood vessel branch in bone defect area were
significantly higher in mice treated with D-sphingosine than those in mice treated with vehicle
(Supplementary Fig. 10c). In conclusion, D-sphingosine promotes the TCVs formation in the defect
area of cortical bone, and thus accelerates the bone formation. We have added the new data into
the manuscript line 436-442.

(Line 436-442) “Data on the 3D-reconstructed canal systems showed an increased canal
component in D-sphingosine treatment group (Supplementary Fig. 10a), suggesting a higher
ability of TCVs formation. To further investigate the TCVs alteration in these two groups,
histological sections and immunofluorescence staining were utilized on the cortical bone defect
area to compare TCVs architecture (Supplementary Fig. 10b-c). The results showed that numbers

of blood vessel branch in bone defect area were significantly higher in mice treated with D-
sphingosine than those in mice treated with vehicle (Supplementary Fig. 10c).”

*Supplementary Fig. 10. (a) Representative images of high-resolution μ CT (1 μ m resolution) on*
*femur callus presenting the naive canal structure in bone defect area from mice that treated with*
*D-sphingosine or saline. (b-c) Representative confocal images of callus (b) and quantitative result*
*of TCVs (c) in bone defect area from mice that treated with D-sphingosine or saline. n=4, scale*
*bars, 100 μ m. Data were presented as means \pm SEMs; Significance was calculated using unpaired*
*t-test with two tailed P value (c).*

**Reviewer #2 (Remarks to the Author):**

In this manuscript, Liao, Chen, Zhou and colleagues aim to demonstrate that osteocyte-derived
mitochondria play a critical role in regulating endothelial cell homeostasis and transcortical blood
vessel maintenance in bone. This is a compelling hypothesis and model that is likely to be of high
interest to many readers studying skeletal biology, the bone marrow microenvironment, and
endothelial cells. The approach involves in vitro co-culture systems and mouse models with
labeled mitochondria from Dmp1-expressing osteolineage cells (mature osteoblasts, osteocytes).
Unfortunately, the critical in vivo evidence presented is limited (e.g. images of transplanted
mitochondria in a single cell without quantification) and missing key controls (described below).
Given the broad claims made, additional analysis is needed to improve the rigor of the data
presented and the impact of the findings. Of the comments listed below, major points 3, 4, and 6
are most critical.

**Response to Reviewer #2:**

*We would like to thank you for your constructive comments, which contributed to the enhancement*
*of our manuscript. We have addressed each point using subtitle as indicated below.*

**Reviewer #2 major comments 1:** Figure 2: The DTA model of osteocyte ablation is quite extreme,
as many of the same authors presented in eLife in 2022. Are differences in osteoblasts, osteoclasts,
bone marrow, and bone mass also present in the DTA mice here? Attributing the changes in the
TCVs to loss of osteocytes alone (as on line 126, line 135) is likely an oversimplification in a
model where several cell and tissue types are ultimately affected.

**Response to Reviewer #2 major comments 1:**

*1) The DTA model of osteocyte ablation is quite extreme, as many of the same authors presented*
*in eLife in 2022. Are differences in osteoblasts, osteoclasts, bone marrow, and bone mass also*
*present in the DTA mice here? - Yes, the $Dmp1^{Cre}DTA^{ki/wt}$ mice used in both previous eLife study¹*
*and the current project were of the same age and gender, ensuring consistency in the experimental*
*conditions. In our previous study, we observed a disruption in the osteoblast lineage and an*

increase in osteoclast activity in trabecular bone, as well as the extinct changes in bone marrow
 and bone mass. Here, our current study specifically investigated the alteration of osteoclasts and
 osteoblast in cortical bone (Supplementary Fig. 1c-f). (For more details, please refer to the
 response to Reviewer #1 comments 1 and 2).

2) Attributing the changes in the TCVs to loss of osteocytes alone (as on line 126, line 135) is likely
 an oversimplification in a model where several cell and tissue types are ultimately affected - Yes,
 we acknowledge that the usage of $Dmp1^{Cre}$ -DTA^{ki/wt} mice may be an oversimplification model for
 detecting osteocytes' role in regulating TCVs. While we have excluded the role of osteoclast on
 TCVs in $Dmp1^{Cre}$ -DTA^{ki/wt} mice (See response to Reviewer #1 comments2), excluding all the
 altered cell lineage could be very challenging. To address the potential extreme effect of DTA model,
 we further analyzed the cortical bone osteoclasts, osteoblasts, and bone marrow components of
 the $Dmp1^{Cre}$ -Rhot1^{fl/fl} mice, which TCVs formation was impaired by inhibition of osteocyte
 mitochondrial transfer, no changes were observed in cortical bone osteoclasts, osteoblasts, and
 bone marrow components (Supplementary Fig. 5a-d, 6a-d). Collectively, these findings are
 sufficient to suggest the specific and essential role of osteocyte mitochondria in maintaining TCV
 networks. We have updated the Discussion (line 512-515) to include this information.

(Line 512-515) “A limitation of our study is the use of $Dmp1^{Cre}$ -DTA^{ki/wt} mice as they may not be
 exclusive to osteocyte. Nevertheless, our comprehensive approach of in vitro assessment and the
 use of $Dmp1^{Cre}$ -Rhot1^{fl/fl} mice and $PF4^{Cre}$ -Cox8^{Dendra2} mice enable to demonstrate the specificity
 of osteocyte mitochondrial transfer in maintaining TCV networks.”

S6a

S6b

S6c

S6d

Supplementary Fig. 1. (c) and quantitative result of ALP positive cell number (d) from 4-week-old

control $DTA^{ki/wt}$ mice and $Dmp1^{Cre}-DTA^{ki/wt}$ mice. $n=3$, scale bars, $100\mu m$, yellow arrows represent

ALP positive cells. (e-f) Representative TRAP staining images of femur cortical bone (e) and

quantitative result of TRAP positive cell number (f) from 4-week-old control $DTA^{ki/wt}$ mice and

$Dmp1^{Cre}-DTA^{ki/wt}$ mice. $n=3$, scale bars, $100\mu m$, yellow arrows represent TRAP positive cells.

Supplementary Fig. 5. (a-b) Representative ALP staining images of femur cortical bone (a) and

quantitative result of ALP positive cell number (b) from 6-week-old control $Rhot1^{fl/fl}$ and $Dmp1^{Cre}-$

$Rhot1^{fl/fl}$ mice. $n=3$, scale bars, $100\mu m$, yellow arrows represent ALP positive cells. (c-d)

Representative TRAP staining images of femur cortical bone (c) and quantitative result of TRAP

positive cell number (d) from 6-week-old control $Rhot1^{fl/fl}$ and $Dmp1^{Cre}-Rhot1^{fl/fl}$ mice. $n=3$, scale

bars, $100\mu m$, yellow arrows represent TRAP positive cells.

Supplementary Fig. 6. (a-b) Gate strategy to identify lymphoid cells (B cells, $CD4^+$ T cells, $CD8^+$

T cells, neutrophils, monocytes, other myeloid cells) (a) and quantitative result of cells percentages

(b) of bone marrow from from 6-week-old control $Rhot1^{fl/fl}$ and $Dmp1^{Cre}-Rhot1^{fl/fl}$ mice. $n=3$. (c-d)

Gate strategy to identify hematopoietic cells (LT-HSC, ST-HSC, Total GMP, CMP, MEP, MPP,

CLP) (c) and quantitative result of cells percentages (d) of bone marrow from from 6-week-old
control $Rhot1^{fl/fl}$ and $Dmp1^{Cre}-Rhot1^{fl/fl}$ mice.

**Reviewer #2 major comments 2:** Figure 2: Do the authors mean to imply that angiogenic genes
Vegfc, Slit3, Notch3, and Notch4 are expressed in osteocytes in the cortical bone tissue? Or are
these genes differentially expressed only in the endothelial cells within homogenized cortical bone
lysate? Can the authors perform in situ hybridization or a complementary in vitro model to validate
that these genes are expressed in osteocytes?

**Response to Reviewer #2 major comments 2:**

Angiogenic genes Vegfc, Slit3, Notch3, and Notch4 are expressed in osteocytes? - Yes. Please refer
to the response to Reviewer #1 comments 4. In summary, we conducted bulk RNA-seq analysis of
cortical bone in $Dmp1^{Cre}-DTA^{ki/wt}$ mice in our first submission manuscript. Briefly, we performed
a bulk RNA-seq of cortical bone in $Dmp1^{Cre}-DTA^{ki/wt}$ mice. The results showed angiogenic genes
Vegfc, Slit3, Notch3, and Notch4 were downregulated in cortical bone of $Dmp1^{Cre}-DTA^{ki/wt}$ mice
(Fig. 2h-i). As per your suggestion, we performed a complementary qPCR assay of MLO-Y4 and
bEnd.3 cell lines. The results revealed that both osteocytes and endothelial cells expressed Vegfc,
Slit3, and Notch3, while MLO-Y4 osteocytes showed significantly lower expression of Notch4
compared to bEnd.3 endothelial cells (Supplementary Fig. 1k). We have revised our Results (Line
166-169) to include this information in our updated manuscript.

(Line 166-169) “To investigate the source of four genes, we performed qPCR analysis on the
MLO-Y4 osteocytes cell line and bEnd.3 endothelial cell line. The results showed the endothelial
cells and osteocytes express similar level of Slit3, Notch3 and Vegfc, while the expression of
Notch4 is relatively low in osteocytes (Supplementary Fig. 1k).”

*Supplementary Fig. 1 (k) RT-qPCR analysis of overlapped genes in bEnd.3 cells and MLO-Y4 cells.*

**Reviewer #2 major comments 3:** Figure 3: The experiments presented in Figure 3, particularly
3c-d, are key to the paper’s claims and require substantial further analysis to rigorously support
that osteocyte-derived mitochondria transfer to endothelial cells. Please present quantification of
the percent of endothelial cells containing Dendra2-labeled mitochondria across multiple
biological replicates. How many Dendra2-labeled mitochondria does each endothelial cell contain?
Images of Cre-negative control animals should be presented to demonstrate the specificity of the
signal. The critical role of osteocytes would also be better supported if the authors labeled
mitochondria with a different Cre promoter (e.g. in hematopoietic stem cells) to describe the cell-
type specificity of mitochondria transfer from osteocytes to endothelial cells. Finally, the authors
should investigate whether osteocytes only transfer mitochondria to endothelial cells within bone
TCVs. Do other endothelial cells in the mouse outside bone also show labeling in this model?

**Response to Reviewer #2 major comments 3:**

1) Quantification of the percent of endothelial cells containing Dendra2-labeled mitochondria
across multiple biological replicates - The quantification of endothelial cells that received
mitochondria from osteocytes are an important data to our article. However, it is hard to
determine a single endothelial cell in vivo because of its continuous connection with their
neighboring endothelial cells. Alternatively, we determined the percentage of TCVs received
osteocyte mitochondria in 4 $Dmp1^{Cre}-Cox8a^{Dendra2}$ mice. The results showed that 79.95% of TCVs
received mitochondria from osteocytes (Fig. 3h). We have revised our Results (Line 230-231) to
include this information in our updated manuscript.

(Line 230-231) “Quantification of cortical bones revealed that 79.95% of TCV fractions contain
mitochondria from osteocytes (Fig. 3h).”

2) How many Dendra2-labeled mitochondria does each endothelial cell contain? - Due to the
same restriction of difficulty in distinguish the single endothelial cells, we did not quantify the
number of Dendra2-labeled mitochondria each endothelial cell contained. Alternatively, we
quantified the number of Dendra2-labeled mitochondria in each TCV. The average number of
transferred mitochondria in every single TCVs is 13.15 ± 2.503 in $Cox8^{Dendra2}-Rhot1^{fl/fl}$ and
4.481 ± 0.5185 in $Dmp1^{Cre}-Cox8^{Dendra2}-Rhot1^{fl/fl}$ (Fig. 4f). We also added another index, the area
ratio of transferred mitochondria to TCVs, for comparing the transfer ratio in two group. The
results also shown that the transferred mitochondria area is $6.809\% \pm 1.032$ of TCVs in $Cox8^{Dendra2}-$
$Rhot1^{fl/fl}$ significantly higher than 3.439 ± 0.4778 in $Dmp1^{Cre}-Cox8^{Dendra2}-Rhot1^{fl/fl}$ (Supplementary
Fig. 4d).

We have revised our Results (Line 262-270) to include this information in our updated manuscript.
(Line 262-270) “Next, to examine the role of MIRO1 in mediating mitochondrial transfer in vivo,
the $Dmp1^{Cre}-Cox8^{Dendra2}-Rhot1^{fl/fl}$ mouse lineage was established, in which Rhot1 was specifically
deleted in Mito^{Dendra2}-labeled osteocytes (Fig. 4d). Confocal imaging and quantification of mouse
femur cortical bone revealed a significantly decreased transferred mitochondria number in each
TCV of $Dmp1^{Cre}-Cox8^{Dendra2}-Rhot1^{fl/fl}$ mice compared to those of $Dmp1^{Cre}-Cox8^{Dendra2}$ control mice
(Fig. 4e-f). Moreover, the ratio of transferred mitochondria area to TCVs area was decreased in
$Dmp1^{Cre}-Cox8^{Dendra2}-Rhot1^{fl/fl}$ mice compared with $Dmp1^{Cre}-Cox8^{Dendra2}$ mice (Supplementary Fig.
4d). These results implied the critical effect of MIRO1 in mediating mitochondrial transfer from
osteocytes to endothelial cells.”

3) Images of Cre-negative control animals should be presented to demonstrate the specificity of
the signal - We have presented here confocal images from $Cox8^{Dendra2}$ and $Dmp1^{Cre}-Cox8^{Dendra2}$
mice respectively (Supplementary Fig. 3d). The results showed there is no Dendra2 signal in
$Cox8^{Dendra2}$ mice, while images from $Dmp1^{Cre}-Cox8^{Dendra2}$ mice showed abundant Dendra2
expression in osteocytes, indicating the specificity expression of this fluorescence protein in
$Dmp1^{+}$ cells. We have revised our Results (Line 227-229) to include this information in our
updated manuscript.

(Line 227-229) “Confocal imaging of $Cox8^{Dendra2}$ mice and $Dmp1^{Cre}-Cox8^{Dendra2}$ cortical bone
demonstrated the specificity of the signal (Supplementary Fig. 3d).”

4) The critical role of osteocytes would also be better supported if the authors labeled
mitochondria with a different Cre promoter (e.g. in hematopoietic stem cells) to describe the cell-
type specificity of mitochondria transfer from osteocytes to endothelial cells - Thank you for the
suggestion. Megakaryocytes and platelets contain abundant of mitochondria. They are ideal
components for investigating mitochondrial exchange with endothelial cells. To test the specificity
of mitochondrial transfer from osteocytes to endothelial cells, we thus employed megakaryocytes
specific Cre promoter, PF4⁷, to test the cell-type specificity of mitochondrial transfer. Analyses of

confocal images of cortical bone from $PF4^{Cre}\text{-Cox8}^{Dendra2}$ mice showed that there is no dendra2
 signals in TCVs and osteocytes in cortical bone, suggesting the specificity of mitochondria transfer
 from osteocytes to endothelial cells. (Supplementary Fig. 3d) We have revised our Results (Line
 242-248) to include this information in our updated manuscript.
 (Line 242-248) “Moreover, as megakaryocytes and platelets contain abundant of mitochondria
 and the latter circulating in blood vessel, there may be mitochondrial exchange from
 megakaryocytes to endothelial cells in TCVs. We thus employed megakaryocytes specific Cre
 promoter, $PF4^7$ to examine TCV of cortical bone in $PF4^{Cre}\text{-Cox8}^{Dendra2}$ mice. The results shows
 that there is no dendra2 signals in TCVs and osteocytes, suggesting that mitochondria transfer
 from osteocytes to endothelial cells is a specific event in cortical bone (Supplementary Fig. 3d).”
 5) Finally, the authors should investigate whether osteocytes only transfer mitochondria to
 endothelial cells within bone TCVs. Do other endothelial cells in the mouse outside bone also show
 labeling in this model? - As liver-bone axis plays a curial role in both of organs^{8,9}, we performed
 immunofluorescence staining in $Dmp1^{Cre}\text{-Cox8}^{Dendra2}$ mice liver to investigate whether osteocytes
 transfer mitochondria to endothelial cells outside bone. The confocal images (Supplementary Fig.
 3d) shows that no dendra2 signal captured in liver blood vessels, suggesting that osteocytes to
 endothelial cells are preferable pathway to transfer mitochondria to endothelial cells in bone than
 in other organs. The corresponding data were updated in our manuscript in line 239-242.
 (Line 239-242) “To examine specificity of mitochondria transferred from osteocytes to endothelial
 cells, we investigate the presence of osteocyte mitochondria in liver tissue as liver-bone axis has
 been shown to play a curial role in both of organs^{8,9}. Confocal microscopy (Supplementary Fig.
 3d) shows that no dendra2 signal captured in liver blood vessels.”

Fig. 3. (h) Quantitative result of TCVs that acquired with Mito-Dendra2 fluorescence in femur
 cortical bone from 4-6-week-old $Dmp1^{Cre}\text{-Cox8a}^{Dendra2}$, $n=4$.
 Supplementary Fig. 3. (d) Representative confocal images of femur cortical bone from 4-week-old
 $Cox8a^{Dendra2}$, $Dmp1^{Cre}\text{-Cox8a}^{Dendra2}$, $PF4^{Cre}\text{-Cox8a}^{Dendra2}$ and liver from $PF4^{Cre}\text{-Cox8a}^{Dendra2}$ mice.

*Supplementary Fig. 4. (d) Quantitative result of area of Mito-Dendra2 fluorescence to area of*
*CD31 in femur cortical bone from 6-week-old $Dmp1^{Cre}-Cox8a^{Dendra2}$ and $Dmp1^{Cre}-Cox8a^{Dendra2}-$*
*$Rhot1^{fl/fl}$. n=3*

**Reviewer #2 major comments 4:** Figure 4: What is the general osteocyte and bone phenotype in
$Dmp1^{Cre}; Rhot1^{fl/fl}$ mice? Are there effects on osteocyte morphology or apoptosis? Are changes
in cortical or trabecular bone mass or remodeling noted? Can differences in TCV vascularization
potentially be attributed to detrimental effects of Rhot1 deletion on osteoblasts, osteoclasts, bone
mass, or bone marrow? Without data along these lines, it is difficult to understand the significance
of the TCV phenotype that is reported.

**Response to Reviewer #2 major comments 4:**

1) What is the general osteocyte and bone phenotype in $Dmp1^{Cre}-Rhot1^{fl/fl}$ mice? - We do not
observe the cortical bone phenotype change in $Dmp1^{Cre}-Rhot1^{fl/fl}$ mice in the term of bone mass
(Supplementary Figure 5k-l).

2) Are there effects on osteocyte morphology or apoptosis? - To investigate the alteration of
osteocyte morphology in $Dmp1^{Cre}-Rhot1^{fl/fl}$, we performed confocal imaging analysis on 6-week-
old $Rhot1^{fl/fl}$ and $Dmp1^{Cre}-Rhot1^{fl/fl}$ mice. We quantified the osteocyte length, width, plus the ratio
of length to width (Supplementary Fig. 5i-j). The results revealed that Rhot1 depletion did not alter
the morphology of osteocytes (Supplementary Fig. 5j), supported by osteocyte length, width, as
well as the ratio of length to width were unchanged (Supplementary Fig. 5j). Furthermore, the
TUNEL staining of cortical bone from both groups indicated apoptosis of osteocytes are not
altered in the $Dmp1^{Cre}-Rhot1^{fl/fl}$ mice (Supplementary Fig. 5e-f).

3) Are changes in cortical or trabecular bone mass or remodeling noted? - We performed a normal
resolution micro-CT analysis on 6-week-old mice femur and showed that there were no notifiable
changes in cortical bone and trabecular bone phenotype (Supplementary Fig. 5k-l).

4) Can differences in TCV vascularization potentially be attributed to detrimental effects of Rhot1
deletion on osteoblasts, osteoclasts, bone mass, or bone marrow? - There is a misunderstanding
on this comment. As we deleted Rhot1 using $Dmp1^{Cre}$ that is specific for osteocytes, we did not
expect the change in mature osteoblasts, osteoclast population. To investigate if the deletion of
Rhot1 deletion in osteocytes results in alteration of osteoblasts, osteoclasts, and bone marrow, we
performed substantial experiment on 6-week-old $Dmp1^{Cre}-Rhot1^{fl/fl}$ mice and control $Rhot1^{fl/fl}$ mice.
ALP staining showed that number of osteoblasts were unchanged in $Dmp1^{Cre}-Rhot1^{fl/fl}$ compared
with $Rhot1^{fl/fl}$ (Supplementary Fig. 5a-b). TRAP staining showed that knocking out Rhot1 in
osteocytes does not change osteoclast population (Supplementary Fig. 5c-d). Flow cytometry
analysis of bone marrow cells from in $Rhot1^{fl/fl}$ and $Dmp1^{Cre}-Rhot1^{fl/fl}$, showed that no significant
changes were observed in both hematopoietic cells (long term-hematopoietic stem cell (LT-HSC),
short term-hematopoietic stem cell (ST-HSC), total granulocyte-macrophage progenitor (GMP),
common myeloid progenitor (CMP), megakaryocyte-erythrocyte progenitor (MEP), multipotent
progenitors (MPP), common lymphoid progenitor (CLP)) and lymphoid cells (B cells, $CD4^{+}$ T
cells, $CD8^{+}$ T cells, neutrophils, monocytes, other myeloid cells) (Supplementary Fig. 6a-d).

Taken together we showed that depletion of Rhot1 in osteocytes has little effect on general
phenotype of cortical bone and bone marrow cells, while impaired significantly on angiogenesis
of TCVs, supporting the specific role of osteocyte mitochondria in regulating TCVs. The
corresponding data were updated in our manuscript in line 281-306.

(Line 281-306) “While these data supported that osteocyte mitochondria plays a role in TCVs
formation, it is not clear if the alteration of TCVs is directly or indirectly due to the depletion of

*Rhot1* in osteocytes. To answer this question, we firstly investigated the effect of *Rhot1* depletion
of osteocytes on osteoblasts, osteoclasts and bone marrow. Histological staining and flow
cytometry analysis were performed on 6-week-old control *Rhot1^{fl/fl}* mice and *Dmp1^{Cre}-Rhot1^{fl/fl}*.
ALP staining showed that number of osteoblasts were unchanged in *Dmp1^{Cre}-Rhot1^{fl/fl}* compared
with *Rhot1^{fl/fl}* (Supplementary Fig. 5a-b). TRAP staining showed that knocking out *Rhot1* in
osteocytes did not change number of osteoclasts (Supplementary Fig. 5c-d). Flow cytometry
analysis of bone marrow cells from in *Rhot1^{fl/fl}* and *Dmp1^{Cre}-Rhot1^{fl/fl}*, showed that no significant
changes were observed in both lymphoid cells (B cells, CD4⁺ T cells, CD8⁺ T cells, neutrophils,
monocytes, other myeloid cells) (Supplementary Fig. 6a-b), and hematopoietic cells (long term-
hematopoietic stem cell (LT-HSC), short term-hematopoietic stem cell (ST-HSC), total
granulocyte-macrophage progenitor (GMP), common myeloid progenitor (CMP),
megakaryocyte-erythrocyte progenitor (MEP), multipotent progenitors (MPP), common lymphoid
progenitor (CLP)) (Supplementary Fig. 6c-d). We then investigate the alteration of osteocytes in
6-week-old *Dmp1^{Cre}-Rhot1^{fl/fl}* mice. TUNEL assay of femur cortical bone showed *Rhot1* knocking
out did not induce the apoptosis of osteocytes (Supplementary Fig. 5e-f). Consistently, no
significant changes of lacunae in cortical bone were observed in the SEMs analysis
(Supplementary Fig. 5g-h). The morphology of osteocytes was not altered as evidenced by the
unaffected cell length and width, and ratio length/width (Supplementary Fig. 5i-j). Moreover, the
bone mass remains unaffected after knocking out *Rhot1* in osteocytes, as supported by obvious
changes in trabecular bone volume (BV), BV/TV, trabecular separation (Tb.Sp), bone mineral
density (BMD), cortical bone area (Ct.Ar), cortical thickness (Ct. Th), tissue mineral density (TMD)
(Supplementary Fig 5k-l). Together, these data showed the depletion of *Rhot1* in cortical bone do
not cause a significant change in bone cell components and bone mass and supported that MIRO1-
mediated mitochondrial transfer from osteocytes to endothelial cells in cortical bone is critical for
maintaining the homeostasis of TCV vascularization.”

 *Supplementary Fig. 5. (a-b) Representative ALP staining images of femur cortical bone (a) and*
 *quantitative result of ALP positive cell number (b) from 6-week-old control $Rhot1^{fl/fl}$ and $Dmp1^{Cre}$ -*
 *$Rhot1^{fl/fl}$ mice. $n=3$, scale bars, 100 μ m, yellow arrows represent ALP positive cells. (c-d)*
 *Representative TRAP staining images of femur cortical bone (c) and quantitative result of TRAP*
 *positive cell number (d) from 6-week-old control $Rhot1^{fl/fl}$ and $Dmp1^{Cre}$ - $Rhot1^{fl/fl}$ mice. $n=3$, scale*
 *bars, 100 μ m, yellow arrows represent TRAP positive cells. (e-f) Representative TUNEL staining*
 *images of femur cortical bone (e) and quantitative result of TUNEL positive cell number (f) from*
 *6-week-old control $Rhot1^{fl/fl}$ and $Dmp1^{Cre}$ - $Rhot1^{fl/fl}$ mice. $n=3$, scale bars, 100 μ m, white arrows*
 *represent TUNEL positive cells. (g-h) Representative SEM images of femur cortical bone (g) and*
 *quantitative result of lacunae number (h) from 6-week-old control $Rhot1^{fl/fl}$ and $Dmp1^{Cre}$ - $Rhot1^{fl/fl}$*

mice. $n=3$, scale bars, $20\mu\text{m}$. (i-j) Representative confocal images of femur cortical bone (i) and
 quantitative result of osteocytes morphology (j) from 6-week-old control $Rhot1^{fl/fl}$ and $Dmp1^{Cre}$ -
 $Rhot1^{fl/fl}$ mice. $n=3$, scale bars, $100\mu\text{m}$. (k-l) Representative micro-CT images of femur (k) and
 quantitative result of bone mass (l) from 6-week-old control $Rhot1^{fl/fl}$ and $Dmp1^{Cre}$ - $Rhot1^{fl/fl}$ mice.
 $n=4$.

 *Supplementary Fig. 6. (a-b) Gate strategy to identify lymphoid cells (B cells, CD4⁺ T cells, CD8⁺*
 *T cells, neutrophils, monocytes, other myeloid cells) (a) and quantitative result of cells percentages*
 *(b) of bone marrow from from 6-week-old control $Rhot1^{fl/fl}$ and $Dmp1^{Cre}$ - $Rhot1^{fl/fl}$ mice. $n=3$. (c-d)*
 *Gate strategy to identify hematopoietic cells (LT-HSC, ST-HSC, Total GMP, CMP, MEP, MPP,*
 *CLP) (c) and quantitative result of cells percentages (d) of bone marrow from from 6-week-old*
 *control $Rhot1^{fl/fl}$ and $Dmp1^{Cre}$ - $Rhot1^{fl/fl}$ mice.*

 **Reviewer #2 major comments 5:** Line 258-260: “Taken together, these results demonstrated that
 endothelial cells acquire mitochondria from osteocytes to alleviate their oxidative stress, improve
 energy production, and restore angiogenic capability.” This wording suggests that endothelial cells

preferentially acquire osteocyte mitochondria when under oxidative stress. To support this claim,
the authors would need to 1) perform all mitochondrial transplant experiments in Figure 5 without
A/R damage to demonstrate any uptake/effect of osteocyte mitochondria on endothelial cells not
experiencing oxidative stress, 2) perform mitochondrial transplant experiments using
mitochondria from a different cell type (e.g. from healthy endothelial cells) to determine whether
these effects are specific to osteocytes, and 3) quantify the proportion of endothelial cells that
contain transplanted mitochondria in each condition to determine whether acquisition is
preferentially occurring in any condition.

**Response Reviewer #2 major comments 5:**

1) perform all mitochondrial transplant experiments in Figure 5 without A/R damage to
demonstrate any uptake/effect of osteocyte mitochondria on endothelial cells not experiencing
oxidative stress - To elucidate the effect of osteocyte mitochondria in healthy endothelial cells not
experiencing oxidative stress, we conducted mitochondrial transplantation in healthy bEnd.3 and
investigated its impact on ROS level, cellular proliferation, scratch assay, and tube formation
assay (Supplementary Fig. 8e-g, k, o, s). The results indicated that bEnd.3 cells received MLO-Y4
mitochondria demonstrate a general improvement, characterized by lower oxidative stress
(Supplementary Fig. 8e-g), higher proliferation (Supplementary Fig. 8k), increased migration
(Supplementary Fig. 8o), and enhanced tube formation speed (Supplementary Fig. 8s), in
comparison to the control bEnd.3 cells. These results suggest that osteocyte mitochondria serve
not only rescuing bEnd.3 cells in a damaged state but also promoting health intracellular status.
The corresponding data were integrated in Results in line 339-360.

(Line 339-360) “To test the effect of osteocyte mitochondria and exclude the involvement of
confounding factors secreted from osteocytes on endothelial functions, we conducted a
mitochondrial transplantation assay on endothelial cells and assessed the alteration of endothelial
capability on cell proliferation, tube formation and cell migration. In detail, the proliferation, tube
formation and migration ability of bEnd.3 cells were shown to be significantly disabled after
different concentrations of A/R treatment, with evidence of decreased CCK8 OD value
(Supplementary Fig. 8h-j), branch point number after tube formation assay (Supplementary Fig.
8l-n), and percentage of migrated area on the wound healing assay (Supplementary Fig. 8p-r),
respectively. Intriguingly, by transplanting osteocyte mitochondria into A/R-damaged bEnd.3 and
healthy bEnd.3 endothelial cells, the CCK8 OD value was significantly elevated in response to
different masses of transplanted mitochondria (Fig. 5f-g, Supplementary Fig. 8k). The group of
different masses of isolated mitochondria was determined by the ratio of mitochondrial donor cells
(MLO-Y4) to mitochondrial recipient cells (bEnd.3). We next examined the osteocyte mitochondria
on endothelial tube formation ability and showed that acquisition of osteocyte mitochondria
restored the capability of tube formation in A/R-damaged bEnd.3 and healthy bEnd.3 cells (Fig.
5h-i, Supplementary Fig. 8o), which implied the promising role of osteocyte mitochondria in
endothelial angiogenesis. As cell migration is also an indicator of endothelial angiogenesis, we
next performed cell scratch and wound healing assays to investigate the effect of acquired
mitochondria on endothelial cell migration¹⁰. The results showed that mitochondrial
transplantation on healthy and 4/4 μ M A/R-damaged endothelial cells effectively recovered the
disabled migration ability of bEnd.3 cells under the 30:1 ratio of mitochondrial donor cells (MLO-
Y4) to recipient cells (bEnd.3) (Fig. 5j-k, Supplementary Fig. 8s).”

2) perform mitochondrial transplant experiments using mitochondria from a different cell type
(e.g. from healthy endothelial cells) to determine whether these effects are specific to osteocytes -
To investigate whether endothelial cell mitochondria have effect on endothelial cells, we isolated

*bEnd.3* endothelial cell mitochondria and co-cultured them with *bEnd.3* cells (ratio: 30:1 and 3:1,
 respectively), followed by ROS detection (Response Fig. 1a-c). The quantification results for ROS
 levels indicated that, at the 30:1 ratio, endothelial cell-derived mitochondria were only effective
 in damaged *bEnd.3* cells but failed to reduce ROS levels in healthy *bEnd.3* cells (Response Fig.1c).
 While at the 3:1 ratio, endothelial cell-derived mitochondria were unable to mitigate oxidative
 stress in either healthy or damaged *bEnd.3* cells (Response Fig.1c). In comparison to the results
 of osteocyte mitochondria transplantation, the efficiency of endothelial cell mitochondria in
 dealing with ROS was relatively low.
 3) quantify the proportion of endothelial cells that contain transplanted mitochondria in each
 condition to determine whether acquisition is preferentially occurring in any condition - We
 quantified the transferred mitochondria in both healthy *bEnd.3* and 2/2 μ M A/R damaged *bEnd.3*.
 The confocal imaging revealed that damaged *bEnd.3* has a higher proportion of cells that
 containing osteocyte mitochondria (Supplementary Fig. 7c-d). We have added the data into the
 manuscript line 322-325.
 (Line 322-325) “We also found *bEnd.3* pretreated antimycin A and rotenone (A/R), mitochondrial
 complexes III and I inhibitors, experienced higher mitochondria acquisition rate, suggesting the
 rescuing role of osteocyte mitochondria on endothelial cells (Supplementary Fig. 7c-d).”

 Response Fig. 1 (a) Gate strategy to identify *bEnd.3* cells treated with or without *bEnd.3* cells
 mitochondria. (b-c) Representative histogram plot of ROS levels (b) and statistical result (c) of
 *bEnd.3* cells after being treated with *bEnd.3* cells mitochondria. n=4.

*Supplementary Fig. 7. (c) Representative confocal images of healthy or 2/2 μ M antimycin*
*A/rotenone damaged bEnd.3 cells after transplantation of MLO-Y4^{Mito-Dendra2} mitochondria for 48*
*hours. scale bars, 50 μ m, yellow arrows represent transferred mitochondria. (d) Quantitative*
*result of bEnd.3 cells acquired with Mito-Dendra2 fluorescence in entire bEnd.3 cells population*
*as shown in (c) n=3.*

*Supplementary Fig. 8. (e-g) The effect of MLO-Y4 cells mitochondria on ROS level on healthy*
*bEnd.3 cells. (e) Gate strategy. (f-g) Representative histogram plot of ROS levels (f) and statistical*
*result (g) of bEnd.3 cells treated with MLO-Y4 cells mitochondria. n=4. (k) Statistical result of*
*CCK-8 OD value of bEnd.3 cells treated with or without MLO-Y4 cells mitochondria. n=7. (o)*
*Statistical result of tube formation assay of bEnd.3 cells treated with or without MLO-Y4 cells*
*mitochondria. n=4 (s) Quantitative result of wound healing assay of bEnd.3 cells treated with or*
*without MLO-Y4 cells mitochondria. n=6.*

**Reviewer #2 major comments 6:** Figure 6R-T: The relationship between TCV homeostasis and
fracture healing is unclear. Do the authors propose that endothelial cells during fracture healing
rely on mitochondria transferred from osteocytes? If so, the authors should provide histologic
evidence that this is occurring. Seven days is rather short to evaluate bone mineralization after
skeletal injury. It is strange that the authors present data related to fracture callus mineralization
here without looking at the effects of Sph treatment on callus angiogenesis. What is the blood
vessel phenotype in the Sph-treated mice within the fracture site? Substantial further work is
needed to understand how this study fits with the rest of the manuscript.

**Response to Reviewer #2 major comments 6:**

*Thank you. We have conducted experiments to answer these questions.*

1) Histologic and micro-CT evidence - Blood vessel phenotype in the osteocyte mitochondria-
treated mice within the fracture site and bone healing - We isolated osteocyte mitochondria and
in situ injected to the defected area of femur in mice twice in one week. The femurs were collected
and performed micro-CT scanning to analyze bone healing and performed immunofluorescence
staining of CD31 to analyze TCVs formation. Confocal imaging analyses of callus TCVs network
showed higher speed of angiogenesis in bone defect area after the mice received osteocyte
mitochondria (Fig. 5l-m). Consistently, the micro-CT analysis revealed improvement of bone
healing quality in mitochondria injection group compared with the vehicle group (Fig. 5n-o). Our
results validated both a positive role of osteocyte mitochondria in TCVs formation and intensive
relationship between angiogenesis and osteogenesis.

2) Histologic evidence - Blood vessel phenotype in the Sph-treated mice within the fracture site -
Histological section and immunofluorescence staining were performed on defected femur cortical
bone from mice that received D-Sphingosine or saline. The results showed abundant transcortical
blood vessel in mice administered D-sphingosine as compared to the control group
(Supplementary Fig. 10a-c), suggesting that D-sphingosine pathway may be the potential
mechanism that osteocyte mitochondria promote TCVs formation.

*We have added the data into the manuscript line 362-372 and line 436-442.*

*(Line 362-372) “Angiogenesis is a key component of bone repair, which relies on new blood*
*vessels to transport the nutrients and inflammatory cells as well as precursor cells¹¹. To examine*
*the therapeutic effect of osteocyte-derived mitochondria on bone healing, we isolated*
*mitochondria from MLO-Y4 in situ injected to the defected area of right femur in mice twice in*
*one week (Fig. 5l). The right femurs were collected and performed micro-CT scanning to analyze*
*bone healing and performed immunofluorescence staining of CD31 to analyze TCVs formation.*

Confocal imaging analyses of callus TCVs network showed higher speed of angiogenesis in bone
 defect area after the mice received osteocyte mitochondria (Fig. 5m-n). Consistently, the micro-
 CT analysis revealed improvement of bone healing quality in mitochondria injection group
 compared with the vehicle group (Fig. 5o-p). Our results validated both a positive role of osteocyte
 mitochondria in TCVs formation and intensive relationship between angiogenesis and
 osteogenesis.”

(Line 436-442) “Data on the 3D-reconstructed canal systems showed an increased canal
 component in D-sphingosine treatment group (Supplementary Fig. 10a), suggesting a higher
 ability of TCVs formation. To further investigate the TCVs alteration in these two groups,
 histological sections and immunofluorescence staining were utilized on the cortical bone defect
 area to compare TCVs architecture (Supplementary Fig. 10b-c). The results showed that numbers
 of blood vessel branch in bone defect area were significantly higher in mice treated with D-
 sphingosine than those in mice treated with vehicle (Supplementary Fig. 10c).”

Fig. 5. (l) Workflow for bone defect surgery and mitochondrial injection. (m-n) Representative
 confocal images of femur callus in bone defected area (m) and quantitative result (n). n=3. (o-p)
 Representative micro-CT images of femur defects 1 week after surgery (o) and histomorphometric
 analysis of the regenerated bone. n=9.

Supplementary Fig. 10. (a) Representative images of high-resolution μ CT (1 μ m resolution)
 on femur callus presenting the naive canal structure in bone defect area from mice that treated with
 D-sphingosine or saline. (b-c) Representative confocal images of callus (b) and quantitative
 result of TCVs (c) in bone defect area from mice that treated with D-sphingosine or saline. n=4, scale
 bars, 100 μ m. Data were presented as means \pm SEMs; Significance was calculated using unpaired
 t-test with two tailed P value (c).

**Reviewer #2 minor comments 1:** Figure 1C-I: How many independent mice and how many
osteocytes per mouse are represented in these measurements?

**Response to Reviewer #2 minor comments 1:** *Figure 1c representative images of three 4-week-*
*old male mice. Figure 1d is a representative confocal image from five 4-week-old male C57BL/6J*
*mice. For the quantification of the osteocyte dendrites (Fig. 1f-h), there are totally 119 osteocytes*
*nearby TCVs from five 4-week-old male C57BL/6J mice. For quantification of osteocytes*
*deflection (Fig. 1i), there are 142 osteocytes adjacent to TCVs and 368 osteocytes distant to TCVs*
*from five 4-week-old male C57BL/6J mice. We have added the number of mice and osteocytes in*
*the Figure Legend (Fig. 1c-i) (Line 1043-1056).*

*(Line1043-1056, Fig. Legend 1c-i) “(c) Representative confocal images of femur cortical bone*
*from three 4-week-old male WT mice injected with Evans Blue intraperitoneally showed multiple*
*dendrites extending from osteocytes (phalloidin, gray) to blood vessels (Evans blue, magenta). (d)*
*Representative confocal images of femur cortical bone from 4-week-old male WT mice showed the*
*dendrite connection between osteocytes (phalloidin, gray) and endothelial cells (CD31, magenta).*
*Scale bars, 20µm. (e) Schematic diagram of osteocytes beside TCVs for further analysis and*
*quantification. (f) Pie diagram of the dendrites of osteocytes beside TCV of (d), 119 osteocytes*
*from 5 mice were quantified. (g) Percentage of the dendrites on the vessel-offside or vessel-*
*nearside among all the dendrites of every osteocyte beside the blood vessel. 119 osteocytes from*
*5 mice were quantified. (h) Quantitative percentage of dendrites in contact with vessels or in*
*noncontact with vessels among all the vessel-nearside dendrites as described in (g). (i) The relative*
*angle between the long bone axis and the long axis of the osteocyte soma distant from TCVs (no*
*dendrite contacts with TCVs) or near TCVs, as shown in (d), 142 osteocytes adjacent to TCVs and*
*368 osteocytes distant to TCVs from 5 mice were quantified.”*

**Reviewer #2 minor comments 2:** Line 105-106: “In detail, 60.27% of the total dendrites were
distributed on the vessel nearside, among which 74.2% were in contact with endothelial cells”.
These numbers do not match the data presented in Figure 1F (59.6% vessel nearside / 70.35%
contact) and Figure 1H (approximately 25% contact to vessel). If panel H is showing detail from
panel F, it may be helpful to color it black and grey.

**Response to Reviewer #2 minor comments 2:**

*We sincerely apologize for this oversight. After careful recalculation and verification, we have*
*identified the mistake in the percentages marked on the figures (Fig. 1f). We have now corrected*
*these inaccuracies to reflect the accurate data.*

*Further, we have taken your suggestion and have changed the color of Figure 1H to black and*
*grey to provide better clarity and understanding. We have updated our figures in Figure 1f and h.*

Fig. 1 f) Pie diagram of the dendrites of osteocytes beside TCV of (d), 119 osteocytes from 5 mice
were quantified. (g) Percentage of the dendrites on the vessel-offside or vessel-nearside among all
the dendrites of every osteocyte beside the blood vessel. 119 osteocytes from 5 mice were quantified.
(h) Quantitative percentage of dendrites in contact with vessels or in noncontact with vessels
among all the vessel-nearside dendrites as described in (g).

**Reviewer #2 minor comments 3:** Line 185-186: Does “digesting the cocultured cells” mean that
cells were fixed and permeabilized for flow cytometry experiments? Please explain the sample
preparation for flow cytometry (Fig. 4C) in the methods. If cells were not fixed, a live-dead stain
would be beneficial. Further, a control condition in which bEnd3 cells are co-cultured with Mloy4
cells without Dendra2-labeled mitochondria is important for gating and quantifying which
endothelial cells are Dendra2-positive.

**Response to Reviewer #2 minor comments 3:**

Sorry for the confusing of using the word of “digesting”. We mean “dispatching”. In our previous
experiments, we directly stained the cells with CD31 antibody without fixation and
permeabilization of cells for flow cytometry analysis as CD31 a membrane protein. Based on your
suggestion, we have performed additional flow cytometry experiments and included a live-dead
staining step using Zombie violent (Fig. 4c). The new results, which include live-dead staining of
mitochondria transfer rate after knocking down Rhot1, were consistent with our previous findings
and provide further evidence that Rhot1 plays a role in mediating mitochondrial transfer. We
believe that the addition of live-dead staining improves the accuracy and reliability of our flow
cytometry data. We have included a step-by-step description of the sample preparation in Method
(Line 698-706) to ensure better replicability of our experiments.

(Line 698-706) “For assessing mitochondrial transfer rate in the coculture system, 5×10^4 bEnd.3
cells and 5×10^4 or 15×10^4 MLO-Y4/MLO-Y4^{Mito-Dendra2} cells were seeded in 6 well plated. The
media were refreshed every 24 hours. After targeted time (24 hours, 48 hours, 72 hours or 7 days),
cells were dispatched and stained with Zombie Violet™ Fixable Viability Kit (Biolegend, Cat.
423113), TruStain fcX™ PLUS (Biolegend, Cat. 156603), APC anti-mouse CD31 Antibody
(Biolegend, Cat. 102510) sequentially. Cell pellets were then resuspended in 200µl PBS
containing 2% FBS and immediately analyzed with a CytoFLEX Flow Cytometer (Beckman
Coulter, California, United States). bEnd.3 co-cultured with MLO-Y4 (without Dendra2
fluorescence) were utilized for gating the Dendra2⁺ bEnd.3 cells. Data were analyzed by FlowJo
software version 10.4.”

Regarding your suggestion for a control condition, we apologize for not presenting it in our
previous manuscript. We would like to clarify that we had used bEnd3-WT MLO-Y4 cells for gating
and quantifying Dendra2⁺ endothelial cells in our previous study. We have revised the gating
strategy and included them in the supplementary material (Supplementary Fig. 2a, 4c, 9g).

4c

Fig. 4. (c) Representative dot-plots and quantitative result of the percentage of bEnd.3 cells
 acquired with Mito-Dendra2 fluorescence in entire bEnd.3 cells population after coculturing with
 NC-MLO-Y4^{Mito-Dendra2} or Rhot1^{KD}-MLO-Y4 cells. n=3.

**Supplementary Fig. 2. (a) Gate strategy to identify Dendra2⁺ bEnd.3 cells cocultured with MLO-**
 **Y4^{Mito-Dendra2} with number ratio at 1:1 and 3:1 (MLO-Y4: bEnd.3) for 24 hours, 48 hours and 72**
 **hours and 7 days.**

**Supplementary Fig. 4. (c) Gate strategy to identify bEnd.3 endothelial cells acquired fluorescent**
 **mitochondria from NC-MLO-Y4^{Mito-Dendra2} or Rhot1^{KD}-MLO-Y4^{Mito-Dendra2} after 24 hours of**
 **coculture.**

**Supplementary Fig. 9. (g) Gate strategy to identify bEnd.3 endothelial cells acquired fluorescent**
 **mitochondria in 2D coculture system supplemented with 0nM, 12.5nM and 50nM D-sphingosine.**

**Reviewer #2 minor comments 4:** Figure 4F: Please describe the quantification strategy. Is the
number of mitochondria measured per cell or within a different unit? What does one point on the
graph represent? Please ensure multiple biological replicates are measured and presented.

**Response to Reviewer #2 minor comments 4:**

*Three $Dmp1^{Cre}-Cox8^{Dendra2}-Rhot1^{fl/fl}$ and three control $Cox8^{Dendra2}-Rhot1^{fl/fl}$ mice were employed*
*for confocal imaging analysis. Each mouse was captured with 4 different fields of view on femur*
*cortical bone. Then we quantified the number of transferred mitochondria in each TCVs from 12*
*images per group. Therefore, each point means the number of mitochondria in one TCV, and we*
*obtained 3 biological replicates. We have revised figure to present biological replicates and added*
*quantification strategy in Figure Legend (Fig. 4f) (Line 1111-1114).*

*(Line 1111-1114, Figure Legend Fig. 4f) "(f) Quantitative assessment of the number of Dendra2-*
*labeled mitochondria in each TCV as shown in (e). Data were quantified from three mice per*
*group, and four different views per mouse were captured for the quantification."*

**Reviewer #2 minor comments 5:** Line 251-252 and Line 268 both refer to endothelial cells
'acquiring' mitochondria in experiments where endothelial cells have been co-cultured with
mitochondria. Judging by Figure 4C, this would be approximately 2% of cells causing a bulk
phenotype. Again, it is important to show what percentage of cells took up the transplanted
mitochondria.

**Response to Reviewer #2 minor comments 5:**

*1) Judging by Figure 4C, this would be approximately 2% of cells causing a bulk phenotype - The*
*primary aim of the experiment in Fig. 4c was to investigate the role of protein MIRO1 (gene: Rhot1)*
*in mitochondria transferring, rather than the transfer rate. We acknowledge that the results (Fig.*
*4c) showed a relatively low percentage of bEnd.3 endothelial cells acquiring MLO-Y4*
*mitochondria, but this does not imply the mitochondrial transfer from osteocytes to endothelial*
*cells is rare. As mentioned in the response to Reviewer #1 comment 4, the low percentage of*
*transfer was due to the short period of coculture and the small number ratio of donor-recipient*
*cells. When we elongated the cocultured time and increased the donor cell number, we observed*
*a significant increase in the transfer rate of mitochondria (Supplementary Fig. 2a-c). Furthermore,*
*we have investigated the mitochondria transfer rate in vivo and found that 79.95% of TCVs*
*acquired osteocyte mitochondria (Fig. 3h), indicating that the transfer of mitochondria from*
*osteocytes to endothelial cells is a common biological phenomenon in vitro and in vivo. We have*
*updated the data in the revised manuscript. See line 194-203 and line 230-231 in the main text.*

*(Line 194-203) "To further explore whether the percentage of endothelial cells that received*
*mitochondria would increase with prolonged co-culture time and higher number ratio of*
*donor/recipient cells, bEnd.3 cells were cocultured with MLO-Y4^{Mito-Dendra2} for 24 hour, 48 hours*
*and 72 hours and 7 days in vitro with the number ratio of MLO-Y4^{Mito-Dendra2} /bEnd.3 at 1:1 and*
*3:1, respectively. The flow cytometry analysis of the co-culture system revealed that the fraction*
*of endothelial cells that received the mitochondrial increased with extended duration of co-culture*
*time and higher number ratio of donor/recipient cells (Supplementary Fig. 2a-d). The peak rate of*
*10% mitochondria-acquired endothelial cells occurs at 3:1 and 72 hours of co-culture*
*(Supplementary Fig. 2b-c). These data supported transfer of mitochondria is time and donor-*
*recipient number ratio dependent."*

*(Line 230-231) "Quantification of cortical bones revealed that 79.95% of TCV fractions contain*
*mitochondria from osteocytes (Fig. 3h)."*

*2) Again, it is important to show what percentage of cells took up the transplanted mitochondria -*
 *To examine the acquisition of transplanted osteocyte mitochondria in endothelial cells, we isolated*
 *MLO-Y4 Mito-Dendra2 mitochondria and co-cultured them with healthy bEnd.3 cells or bEnd.3*
 *cells damaged by 2/2 μM A/R for 48 hours. Our analysis using confocal imaging revealed that*
 *after 48 hours of co-culture, 56.33±1.01% of healthy bEnd.3 cells took up the transplanted*
 *mitochondria. In contrast, a higher percentage of damaged bEnd.3 cells acquired the transplanted*
 *mitochondria (69.49±4.44%) (Supplementary Fig. 7c-d). We have included the data in the revised*
 *manuscript. See line 331-335 in the main text.*
 *(Line 322-325) “We also found bEnd.3 pretreated antimycin A and rotenone (A/R), mitochondrial*
 *complexes III and I inhibitors, experienced higher mitochondria acquisition rate, suggesting the*
 *rescuing role of osteocyte mitochondria on endothelial cells (Supplementary Fig. 7c-d).”*

*Fig. 3. (h) Quantitative result of TCVs that acquired with Mito-Dendra2 fluorescence in femur*
 *cortical bone from 4-6-week-old $Dmp1^{Cre-Cox8a^{Dendra2}}$. $n=4$.*
 *Supplementary Fig. 2. (a) Gate strategy to identify $Dendra2^+$ bEnd.3 cells cocultured with MLO-*
 *$Y4^{Mito-Dendra2}$ with number ratio at 1:1 and 3:1 (MLO-Y4: bEnd.3) for 24 hours, 48 hours and 72*
 *hours and 7 days. (b) Representative dot-plots and quantitative result of the percentage of bEnd.3*
 *cells acquired with Mito-Dendra2 fluorescence in entire bEnd.3 cells population after coculturing*
 *with MLO-Y4^{Mito-Dendra2} for 24 hours with number ratio at 1:1 and for 7 days with number ratio at*
 *3:1 (MLO-Y4: bEnd.3). $n=6$. (c-d) Line plot (c) and quantitative result (d) of bEnd.3 cells acquired*

with Mito-Dendra2 fluorescence in entire bEnd.3 cells population cocultured with MLO-Y4^{Mito-}
 Dendra2 with number ratio at 1:1 and 3:1 (MLO-Y4: bEnd.3) for 24 hours, 48 hours and 72 hours
 and 7 days.

Supplementary Fig. 7. (c) Representative confocal images of healthy or 2/2 μ M antimycin
 A/rotenone damaged bEnd.3 cells after transplantation of MLO-Y4^{Mito-Dendra2} mitochondria for 48
 923 hours. scale bars, 50 μ m, yellow arrows represent transferred mitochondria. (d) Quantitative
 result of bEnd.3 cells acquired with Mito-Dendra2 fluorescence in entire bEnd.3 cells population
 as shown in (c) n=3.

**Reviewer #2 minor comments 6:** Figure 6: Can the authors comment on why Sph appears to
 become more effective with higher concentration for some outcomes (Figure 6D,J), but also
 saturate at 12.5 nM (Figure 6F), and reverse its effect with high concentration (Figure 6H). This
 inconsistency would suggest that Sph is not the only factor that matters here.

**Response to Reviewer #2 minor comments 6:**

*That's a good question. In our preliminary experiments, we observed that the effects of D-*
 *sphingosine varied depending on the concentration. Specifically, low concentrations of D-*
 *sphingosine promoted cellular proliferation, while high concentrations induced cell death*
 *(Supplementary Fig. 9b-c). Thus, we used 12.5nM and 50nM for further study. These observations*
 *are consistent with previous studies on sphingosine^{12, 13}. The inconsistent effects of 50nM D-*
 *sphingosine probably caused by the different cell density in each experiment. Because we found*
 *that 50nM D-sphingosine promoted cell proliferation when cell density was high but induced cell*
 *death when the density was low (Supplementary Fig. 9d-e). Therefore, experiments with high cell*
 *seeding density (Scratch assay and ROS detection) required a higher dose of sphingosine to see a*
 *significant effect, while experiments with lower cell seeding density (Tube formation) were more*
 *sensitive to D-sphingosine and showed beneficial effects at lower doses, but detrimental effects at*
 *higher doses.*

We have revised Results in line 394-400 to include this information.

(Line 394-400) *“The effect of sphingosine is dose dependent^{12, 13}. Specifically, high dose of*
 *sphingosine induces apoptosis, while low dose of sphingosine promotes cell growth^{12, 13}. Thus, we*
 *investigated approximal concentration of D-sphingosine before further experiment. CCK8*
 *analysis were performed on different concentrations of D-sphingosine and bEnd.3 cell density and*
 *the results suggested D-sphingosine works positively lower than 50nM (Supplementary Fig. 9b-c)*
 *with cell density more than 12500 cell/cm² (Supplementary Fig. 9d-e).”*

*Supplementary Fig. 9. (b-c) Line plot (b) and quantitative result (c) of CCK-8 assay to investigate*
*the effect of different D-sphingosine concentration on bEnd.3 with seeding density 12500 cell/cm².*
*(d-e) Line plot (d) and quantitative result (e) of CCK-8 assay to investigate the reaction of bEnd.3*
*to 50nM D-sphingosine with cell seed density at 6250, 12500 and 18750 and 25000 cells/cm².*

**Reviewer #3 (Remarks to the Author):**

Liao et al present a fascinating manuscript detailing a novel mechanism whereby osteocytes
regulate angiogenesis and transcortical vessel (TCV) formation by transferring mitochondria that
contain sphingosine. The authors provide a fascinating description of the TCV network between
osteocytes and endothelial cells (ECs) and demonstrate that depletion of osteocytes leads to the
formation of spare, disorganized TCV network. In co-culture systems using mitochondria reporter
osteocytes, they show that osteocytes transfer mitochondria to bEnd.3 cells in vitro. This process
occurs in vivo as well and was impaired in the absence of MIRO1 in osteocytes, identifying that
this protein (which is known to shuttle mitochondrial along tunnelling nanotubes) mediates
mitochondria transfer from osteocytes to ECs. Deletion of MIRO1 in osteocytes was also
associated with TCV regression in vivo. Adding purified mitochondria from osteocytes onto
bEnd.3 cells rescued A/R-induced defects in cell counts, branch point formation, and migration.
The mitochondria-treated cells have more sphingosine (Sph) and spingosine-1-phosphate (S1P),
so the authors treated bEnd.3 cells stressed with A/R with or without Sph and observed that Sph
also rescued cell counts, branch point formation, and migration. Mitochondria-mediated rescue of
A/R-stressed bEnd.3 cells was blocked by PF543, suggesting that this process is dependent on the
formation of S1P. This manuscript is rigorous and uses the ideal strategies to track mitochondria
transfer in vitro and in vivo. Conceptually, this manuscript increases our understanding of how
mitochondria transfer regulates angiogenic responses through production of sphingolipids and
reveals a previously uncharacterized mechanism of mitochondria transfer along TCVs. While I am
enthusiastic about this manuscript, I do have several major concerns:

***Response to Reviewer #3:*** *We are grateful for your constructive comments which helped to*
*improve our manuscript. The following are our point-by-point responses:*

**Reviewer #3 major comments 1:** My biggest conceptual concern is that the mechanism of
mitochondria transfer from osteocytes to ECs is along TCVs in a MIRO1-dependent manner, a lot
like tunnelling nanotubes (TNTs). With TNTs, the mitochondria are deposited directly into the
cytoplasm. However, the in vitro studies in Fig 5-6 utilize exogenous mitochondria purified from
osteocytes and then simply add them to the ECs in culture. Uptake of free mitochondria is a very
different transfer mechanism than delivery via a tubular structure such as the TCV. I am having a
hard time understanding this. Are the TCVs making contact with the ECs and then ejecting free
mitochondria at the EC surface for capture? Or are they depositing the mitochondria directly into
the EC's cytoplasm?

***Response to Reviewer #3 major comments 1:***

*1) Are osteocytes transfer mitochondria to endothelial cells via TNTs? - Mitochondria can be*
*transferred through microvesicles¹⁴, tunneling nanotubes (TNTs)¹⁵, and extrusion/internalization*
*of free mitochondria¹⁶. In this study, we found dendritic structures between osteocytes and ECs,*
*suggesting the possibility of intercellular mitochondria transfer from osteocytes to ECs. However,*
*our main objective in this study was to investigate the effect of osteocyte mitochondria on ECs,*
*and therefore, we did not specifically investigate the path of mitochondria transfer. We have added*
*this in Discussion. See line 477-481 in the main text*

(Line 477-481) “Spontaneous intercellular transfer of mitochondria occurs under physiological
 conditions and contributes to tissue development and remodeling. Various routes for
 mitochondrial transfer have been reported, including tunneling nanotubes, microvesicles, and
 extrusion and internalization of free mitochondria¹⁶. Our study showed that osteocytes may utilize
 the special structure of extension of dendritic inflated endfeet abutting on the TCV for transferring
 of mitochondria.”

2) Why using mitochondrial transplantation? - Isolating mitochondria from donor cells and
 cocultured it with recipient cells is a simplify model in studying the role of mitochondria transfer.
 Comparing with cell-to-cell coculture, the model can exclude the influence of non-mitochondrial
 cellular components such as ions, exosomes and cytokines. Thus, this method has been widely used
 when study mitochondrial transferring¹⁷⁻²². Nevertheless, we acknowledge that investigating
 mitochondria transfer by transplantation may not mimic the reality, more optimal methods should
 be developed. We have added this as one of the limitations. See line 515-517 in the main text.

(Line 515-517) “Another limitation is that the use of isolated mitochondria for cocultured with
 endothelial cells may not mimic the reality in vivo even this is a common method used in
 mitochondrial transfer¹⁷⁻²².”

3) Are the TCVs making contact with the Ecs and then ejecting free mitochondria at the EC surface
 for capture? Or are they depositing the mitochondria directly into the EC’s cytoplasm? -

Investigating the pathway that osteocyte transferred mitochondria to TCVs is challenging, due to
 the limitation of in vivo imaging resolution. Thus, we performed a complementary experiment in
 vitro. We used transwell coculture system to separate bEnd.3 and MLO-Y4, and measured the
 transfer of mitochondria without dendrites contact (Fig. 3c-e). The confocal image analysis
 revealed that mitochondria transfer can occur through non-contact transferring (Fig. 3d-e).
 However, its transfer efficiency is lower than contacting transferring, indicating the TNT is the
 main routine for osteocyte mitochondria entering the endothelial cells (Fig. 3e). We have included
 the data in the revised manuscript. See line 203-209 in the main text.

(Line 203-209) “Moreover, we investigated the main routine of MLO-Y4 osteocytes transferring
 mitochondria to bEnd.3 endothelial cells. We used transwell coculture system to separate bEnd.3
 cells and MLO-Y4 cells, and measured the transfer of mitochondria without dendrites contact (Fig.
 3c). The confocal image analysis revealed that mitochondria transfer can also occur through non-
 contact transferring (Fig. 3d-e). However, its transfer efficiency is lower than contacting
 transferring, indicating osteocyte mitochondria entering the endothelial cells mainly based on
 directly contact.”

Fig. 3. (c) Schematic diagram for 2D-coculture and transwell coculture system of bEnd.3 cells
 and MLO-Y4 cells with number ratio at 3:1 (MLO-Y4: bEnd.3) for 48 hours. (d-e) Representative
 confocal images (d) and quantitative result (e) of bEnd.3 cells acquired with Mito-Dendra2

fluorescence as shown in (c), $n=3$, scale bars, $50\mu\text{m}$, yellow arrows represent transferred
mitochondria.

**Reviewer #3 major comments 2:** What is the gating strategy for Fig 3c?

**Response to Reviewer #3 major comments 2:**

The Fig. 3c in our first submission manuscript was a schematic diagram for the generation of the
$Dmp1^{Cre}\text{-Cox8}^{Dendra2}$ mouse line.

Regarding the gating strategy, we assume you may be referring to Fig. 4c instead. In that case,
the gating strategy used in Figure 4c can be found in Supplementary Fig. 4c.

**Supplementary Fig. 4. (c) Gate strategy to identify bEnd.3 endothelial cells acquired fluorescent**
**mitochondria from NC-MLO-Y4^{Mito-Dendra2} or Rhot1^{KD}-MLO-Y4^{Mito-Dendra2} after 24 hours of**
**coculture.**

**Reviewer #3 major comments 3:** Statistics are needed for Fig 5b.

**Response to Reviewer #3 major comments 3:**

In the seahorse assay, we measured four different indexes: basal respiration, maximal respiration,
ATP production, and spare respiratory capacity. To provide a clear and comprehensive
presentation of the statistical analysis, we have chosen to present the p-values for these four
indexes in Figure 5c. In Figure 5b, we focused on visualizing the line plot, which better illustrates
the trends of the data.

By presenting the p-values in a separate Figure (Fig. 5c) with independent bar plots, we aimed to
ensure readability and facilitate the interpretation of the results. This approach is in line with the
presentation strategies employed in other publications²³⁻²⁵.

**Reviewer #3 major comments 4:** Does exogenous Sph treatment affect mitochondria transfer
from osteocytes to Ecs?

**Response to Reviewer #3 major comments 4:**

In response to your comment, we supplemented the $b\text{End.3-MLO-Y4}^{Mito-Dendra2}$ coculture system
with 0nM , 12.5nM and 50nM of D-sphingosine. After 24 hours of coculture, we collected the cells
for flow cytometry analysis to measure the mitochondria transfer rate. The results showed that
exogenous sphingosine supplementation decreased the mitochondria transfer process. Moreover,
the transfer rate decreased with a higher concentration of sphingosine. (Supplementary Fig. 10g-
i). We have included the data in the revised manuscript. See line 410-413 in the main text.

(Line 410-413) “We also investigated the effect of D-sphingosine on mitochondria transfer. Flow
 cytometry analysis showed that exogenous D-sphingosine supplementation decreased the
 mitochondria transfer process (Supplementary Fig. 9g-i). Moreover, the transfer rate decreased
 with a higher concentration of D-sphingosine (Supplementary Fig. 9i).”

 *Supplementary Fig. 9. (g-i) The influence of D-sphingosine on bEnd.3 cells acquiring*
 *mitochondria from MLO-Y4^{Mito-Dendra2}. (g) Gate strategy to identify bEnd.3 endothelial cells*
 *acquired fluorescent mitochondria in 2D coculture system supplemented with 0nM, 12.5nM and*
 *50nM D-sphingosine (h-i) Representative dot plot (h) and quantitatively results (i) of the percentage*
 *of bEnd.3 cells acquired with Mito-Dendra2 fluorescence in entire bEnd.3 cells population that*
 *cocultured with MLO-Y4^{Mito-Dendra2} for 48 hours in present of 12.5nM, 25nM, 50nM D-sphingosine.*

**Reviewer #3 major comments 5:** Fig 6m-q: another group is required, specifically 4/4 A/R +
 PF543 (no mito)

**Response to Reviewer #3 major comments 5:**

*We apologize for any confusion caused. Our main objective was to compare the effects of normal*
 *MLO-Y4 mitochondria and PF543-pretreated MLO-Y4 mitochondria on bEnd.3. It is important to*
 *note that PF543, the SPHK1 inhibitor, did not directly react with bEnd.3 cells. Prior to and during*
 *the mitochondria isolation procedure (as shown in Figure 6m), the PF543 that was pretreated in*
 *MLO-Y4 media is removed and washed out.*

*In order to address any potential misunderstanding arising from the Figure label, we have revised*
 *the group name for bEnd.3 cells that received PF543-pretreated MLO-Y4 mitochondria as*
 *“(PF543-pretreated Mito)” in the tube formation assay and wound healing assay (Fig. 6o, q).*

 **Reviewer #3 major comments 6:** Does administration of mitochondria improve bone healing in
 the surgical model?

**Response to Reviewer #3 major comments 6:**

Thank you for your comment regarding the administration of mitochondria in bone healing. Please
 refer to the response to Reviewer #2 comments major 6 response for detailed information. Briefly,
 to investigate the effect of administration of osteocyte mitochondria on bone defect healing and
 callus angiogenesis in vivo, we performed mitochondria injection twice a week on bone defect site
 (Fig. 5l). The confocal image analysis of callus revealed the mice received osteocyte mitochondria
 exhibited an increasing number of blood vessels branches in healing area, suggesting the
 promoting effect of osteocyte mitochondria on blood vessel formation in cortical bone (Fig. 5m-
 n). We also observed a faster bone healing speed of defect area in mice injected with osteocyte
 mitochondria, which is supported by micro-CT analysis (Fig. 5o-p). In conclusion, our findings
 suggest that osteocyte mitochondria play a crucial role in blood vessel formation in cortical bone
 and have a significant impact on bone healing (Fig. 5l-p). We have added the data into the
 manuscript line 362-372.

(Line 362-372) “Angiogenesis is a key component of bone repair, which relies on new blood
 vessels to transport the nutrients and inflammatory cells as well as precursor cells¹¹. To examine
 the therapeutic effect of osteocyte-derived mitochondria on bone healing, we isolated
 mitochondria from MLO-Y4 in situ injected to the defected area of right femur in mice twice in
 one week (Fig. 5l). The right femurs were collected and performed micro-CT scanning to analyze
 bone healing and performed immunofluorescence staining of CD31 to analyze TCVs formation.
 Confocal imaging analyses of callus TCVs network showed higher speed of angiogenesis in bone
 defect area after the mice received osteocyte mitochondria (Fig. 5m-n). Consistently, the micro-
 CT analysis revealed improvement of bone healing quality in mitochondria injection group
 compared with the vehicle group (Fig. 5o-p). Our results validated both a positive role of osteocyte
 mitochondria in TCVs formation and intensive relationship between angiogenesis and
 osteogenesis.”

Fig. 5. (l) Workflow for bone defect surgery and mitochondrial injection. (m-n) Representative
 confocal images of femur callus in bone defected area (m) and quantitative result (n). n=3. (o-p)
 Representative micro-CT images of femur defects 1 week after surgery (o) and histomorphometric
 analysis of the regenerated bone. n=9.

 **Reviewer #3 major comments 7:** The authors have not really distinguished whether D-
 sphingosine or S1P are responsible for the phenotypes observed in ECs. Their PF543 data suggest

that perhaps SIP might be the responsible factor. The authors should clarify this experimentally,
 or they should temper the conclusions and text to more accurately state that mitochondria transfer
 delivers sphingolipids that promote EC responses, rather than making such specific claims about D-
 sphingolipid.

**Response to Reviewer #3 major comments 7:**

Yes, we revised our manuscript (line 390-391, line 430-431, line 447-448) according to your
 suggestion. It's interesting to dig down to the mechanism of sphingolipid in future. We have
 updated our conclusion that osteocyte mitochondria promote endothelial cells angiogenesis at
 least in part mediated via sphingosine pathway.

(Line 390-391) "These results suggest that osteocyte mitochondria may regulate angiogenesis
 through the upregulated sphingolipid pathway."

(Line 430-431) "In summary, these data indicated that osteocyte mitochondria promote
 endothelial cells angiogenesis at least in part mediated via sphingosine pathway."

(Line 447-448) "In summary, these results supported that sphingolipid pathway at least in part
 involves in the osteocyte mediated endothelial cells angiogenesis in cortical bone."

 **Reviewer #3 minor comments 1: Fig 6h: why does 12.5 nM Sph restore branch points after A/R**
 **but 50nM does not?**

**Response to Reviewer #3 minor comments 1:**

This a similar question with reviewer #2 comments minor 6. Please refer to our detailed answer
 to this question. Briefly, we found the sphingosine has a dose-dependent effect and cell density
 dependent effect on cells (Supplementary Fig. 9b-e). 12.5nM sphingosine is sufficient to promote
 tube formation of bEnd.3 cells. But 50nM sphingosine may be overdose for them and inhibit the
 tube formation. We have revised our results in line 394-400 to include this information.

(Line 394-400) "The effect of sphingosine is dose dependent^{12, 13}. Specifically, high dose of
 sphingosine induces apoptosis, while low dose of sphingosine promotes cell growth^{12, 13}. Thus, we
 investigated approximal concentration of D-sphingosine before further experiment. CCK8
 analysis were performed on different concentrations of D-sphingosine and bEnd.3 cell density and
 the results suggested D-sphingosine works positively lower than 50nM (Supplementary Fig. 9b-c)
 with cell density more than 12500 cell/cm² (Supplementary Fig. 9d-e)."

 **Supplementary Fig. 9. (b-c) Line plot (b) and quantitative result (c) of CCK-8 assay to investigate**
 **the effect of different D-sphingosine concentration on bEnd.3 with seeding density 12500 cell/cm².**

(d-e) Line plot (d) and quantitative result (e) of CCK-8 assay to investigate the reaction of bEnd.3
to 50nM D-sphingosine with cell seed density at 6250, 12500 and 18750 and 25000 cells/cm².

**Reviewer #3 minor comments 2:** Fig 5d and Fig 6c are not very convincing and underlies the
data in Fig 5e and 6d, respectively. I am not sure what value the ROS data add here given that the
functional readouts of cell counts and branch points are more prominent and relevant.

**Response to Reviewer #3 minor comments 2:**

*Thank you for your question. In Fig. 5d and Fig.6c, ROS-FITC mean fluorescence intensity was*
*presented with a 10-logarithmic x-axis to provide an overview of the entire dataset, which resulted*
*in the compression of the scale of data distribution in the x-axis and made the differences less*
*obvious (Fig. 5d and Figure 6c). However, for the statistical plots (Fig. 5e and Figure 6d), we*
*opted to present ROS-FITC mean fluorescence intensity using a linear y-axis, which allowed for*
*a clearer demonstration of differences. This presentation strategy aligns with similar approaches*
*used in other publications (Sci Transl Med. 2022²⁶, Sci Adv.2022²⁷, Nat Commun. 2023²⁸, Nat*
*Immunol 2021²⁹).*

*In comparisons with other experiments, the results of ROS assay demonstrated unique the*
*biological effects of osteocyte mitochondria and D-sphingosine on endothelial cells.in alleviating*
*oxidative stress. While results from CCK8, tube formation, and wound healing assays indicated*
*the role of osteocyte mitochondria and D-sphingosine in improving endothelial cells proliferation,*
*angiogenesis, and migration.*

**Reviewer #3 minor comments 3:** The discussion would be improved if the authors could
speculate about why is sphingosine delivered to ECs via mitochondria transfer.

**Response to Reviewer #3 minor comments 3:**

*We have added the information into the discussion (line 500-504). “Although our data showed*
*endothelial cells that received osteocytes-derived mitochondria had a higher level of D-*
*sphingosine, it is not clear why is sphingosine increased in ECs via mitochondria transfer. While*
*we speculated that transferred mitochondria may activate the production of sphingosine in ECs,*
*further investigation on the mechanism of upregulation of sphingosine in osteocyte mitochondria*
*transfer to endothelial cells is required.”*

**Reference**

- 1. Ding P, *et al.* Osteocytes regulate senescence of bone and bone marrow. *Elife* **11**, (2022).
- 2. Gruneboom A, *et al.* A network of trans-cortical capillaries as mainstay for blood
circulation in long bones. *Nat Metab* **1**, 236-250 (2019).
- 3. Grüneboom A, *et al.* A network of trans-cortical capillaries as mainstay for blood
circulation in long bones. *Nature Metabolism* **1**, 236 (2019).
- 4. Bonewald LF. The amazing osteocyte. *J Bone Miner Res* **26**, 229-238 (2011).
- 5. Frost HM. In vivo osteocyte death. *J Bone Joint Surg Am* **42-a**, 138-143 (1960).
- 6. Plotkin LI, Bellido T. Osteocytic signalling pathways as therapeutic targets for bone
fragility. *Nat Rev Endocrinol* **12**, 593-605 (2016).
- 7. Slungaard A. Platelet factor 4: a chemokine enigma. *Int J Biochem Cell Biol* **37**, 1162-1167
(2005).
- 8. Lin L, *et al.* SIRT2 regulates extracellular vesicle-mediated liver-bone communication. *Nat*
*Metab* **5**, 821-841 (2023).
- 9. Song Z, *et al.* Osteopontin Takes Center Stage in Chronic Liver Disease. *Hepatology* **73**,
1594-1608 (2021).
- 10. Liang CC, Park AY, Guan JL. In vitro scratch assay: a convenient and inexpensive method
for analysis of cell migration in vitro. *Nat Protoc* **2**, 329-333 (2007).
- 11. Hankenson KD, Dishowitz M, Gray C, Schenker M. Angiogenesis in bone regeneration.
*Injury* **42**, 556-561 (2011).
- 12. Zhang H, Buckley NE, Gibson K, Spiegel S. Sphingosine stimulates cellular proliferation
via a protein kinase C-independent pathway. *J Biol Chem* **265**, 76-81 (1990).
- 13. Iwata M, Herrington J, Zager RA. Sphingosine: a mediator of acute renal tubular injury
and subsequent cytoresistance. *Proc Natl Acad Sci U S A* **92**, 8970-8974 (1995).
- 14. Phinney DG, *et al.* Mesenchymal stem cells use extracellular vesicles to outsource
mitophagy and shuttle microRNAs. *Nat Commun* **6**, 8472 (2015).
- 15. Acquistapace A, *et al.* Human mesenchymal stem cells reprogram adult cardiomyocytes
toward a progenitor-like state through partial cell fusion and mitochondria transfer. *Stem*
*Cells* **29**, 812-824 (2011).
- 16. Liu D, *et al.* Intercellular mitochondrial transfer as a means of tissue revitalization. *Signal*
*Transduct Target Ther* **6**, 65 (2021).
- 17. Hayakawa K, *et al.* Transfer of mitochondria from astrocytes to neurons after stroke.
*Nature* **535**, 551-555 (2016).
- 18. Borcherdig N, *et al.* Dietary lipids inhibit mitochondria transfer to macrophages to divert
adipocyte-derived mitochondria into the blood. *Cell Metab* **34**, 1499-1513.e1498 (2022).
- 19. Brestoff JR, *et al.* Intercellular Mitochondria Transfer to Macrophages Regulates White
Adipose Tissue Homeostasis and Is Impaired in Obesity. *Cell Metab* **33**, 270-282.e278
(2021).
- 20. Zhang W, *et al.* Cancer cells reprogram to metastatic state through the acquisition of
platelet mitochondria. *Cell Rep* **42**, 113147 (2023).
- 21. Suh J, *et al.* Mitochondrial fragmentation and donut formation enhance mitochondrial
secretion to promote osteogenesis. *Cell Metab* **35**, 345-360.e347 (2023).
- 22. Huang T, *et al.* Efficient intervention for pulmonary fibrosis via mitochondrial transfer
promoted by mitochondrial biogenesis. *Nat Commun* **14**, 5781 (2023).
- 23. Giddings EL, *et al.* Mitochondrial ATP fuels ABC transporter-mediated drug efflux in
cancer chemoresistance. *Nat Commun* **12**, 2804 (2021).

- 24. Koopman M, *et al.* A screening-based platform for the assessment of cellular respiration
in *Caenorhabditis elegans*. *Nat Protoc* **11**, 1798-1816 (2016).
- 25. Kramer NJ, *et al.* Regulators of mitonuclear balance link mitochondrial metabolism to
mtDNA expression. *Nat Cell Biol* **25**, 1575-1589 (2023).
- 26. Mao C, *et al.* Delivery of an ectonucleotidase inhibitor with ROS-responsive nanoparticles
overcomes adenosine-mediated cancer immunosuppression. *Sci Transl Med* **14**, eabh1261
(2022).
- 27. Liu D, *et al.* ROS-responsive chitosan-SS31 prodrug for AKI therapy via rapid distribution
in the kidney and long-term retention in the renal tubule. *Sci Adv* **6**, (2020).
- 28. Wang Y, *et al.* The transcription factor Zeb1 controls homeostasis and function of type 1
conventional dendritic cells. *Nat Commun* **14**, 6639 (2023).
- 29. Gu M, *et al.* NF- κ B-inducing kinase maintains T cell metabolic fitness in antitumor
immunity. *Nat Immunol* **22**, 193-204 (2021).

REVIEWERS' COMMENTS

Reviewer #1 (Remarks to the Author):

This was a very substantial revised manuscript and improved substantially. I congratulate the authors to their work!!! All my concerns are excellently addressed.

Reviewer #2 (Remarks to the Author):

Enthusiasm for this study remains high, which has potential for high impact across fields of skeletal biology, the bone marrow microenvironment, and endothelial cell biology. The revised manuscript and additional data represents a substantial improvement versus the first version. The authors have thoughtfully addressed our comments, and at this point, only a few 'minor' points remain.

1. The labels in the legend below panel 1H still appear to be swapped.
2. In the new quantification in Figure 3H, please indicate standard deviation across biological replicates.
3. In Supplementary Figures 1C and 5A, ALP staining is performed to quantify osteoblasts. However, the cells indicated by yellow arrows are embedded osteocytes rather than surface osteoblasts. This quantification needs to be re-performed to appropriately quantify bone surface osteoblast number.
4. The new descriptions of osteocyte morphology in Rhot1 knockout mice in Supplementary Figure 5G-J are helpful but lack description of osteocyte dendrite morphology. Since the mechanism of mitochondrial transfer to endothelial cells likely involves dendrites (Figure 1), the length and number of osteocyte dendrites should be quantified in Dmp1-Cre;Rhot1^{fl/fl} mice compared to control.
5. The femur defect model, now shown in Figure 5 and in Figure 6, still lacks data to fully support the specific role of mitochondrial transfer in healing. Primarily, there is no histological evidence that endothelial cells near the bone defect are receiving osteocyte mitochondria, and effects of exogenous mitochondria and Sph on other bone cells are not ruled out. Further, full healing of the defect would be expected to take many weeks, so the endpoint of one week after injury (and just a few days after mitochondrial transfer / Sph administration) is not sufficient to demonstrate a meaningful effect on bone healing. We respectfully suggest that the effect of Sph on bone healing be removed from the abstract given its unclear relationship with the main focus of the study.

Reviewer #3 (Remarks to the Author):

The authors have done a reasonable job responding to my concerns, but some minor issues remain. They can be addressed with changes to the text without any additional experimental work, so I feel a minor revision is warranted so the authors have an opportunity to improve the manuscript.

Comment 1: I think the interpretation of the transwell experiment is problematic. First, as the authors point out, there is a substantial amount of mitochondria transfer (25%) across the transwell, which means that osteocytes do release cell-free mitochondria for transfer to endothelial cells. Second, the "lower efficiency" of mitochondria transfer with transwell separation could be explained by dilution of the released mitochondria into a relatively large volume of media. When the cells are cultured in the same chamber and in direct proximity of each other, the local concentration of released mitochondria is much higher, so transfer will be more efficient. Therefore the difference in mito transfer efficiency they are reporting could be explained entirely by a dilutional effect, which is an artifact of the system. Their conclusion is that the mito transfer is primarily contact-dependent, but I don't think they have convincingly shown that.

As far as I can tell, the most defensible conclusion is that osteocytes deliver mitochondria to their

dendritic endfeet in a MIRO-1-dependent manner and then transfer mitochondria to endothelial cells via unclear mechanisms that may involve release of mitochondria from the cell and/or direct cellular contact.

I do not expect the authors to distinguish between those mechanisms in this paper, but they should acknowledge alternative pathways without staking a claim on one particular mechanism.

Comment 2: The gating strategy in Supplemental Fig 4c (and in other gating examples they provide) is confusing for 3 reasons. (1) The authors have added gates that are not used for sequential gating, making it unclear which population of cells they are gating on. (2) The gating arrows are positioned on what appear to be the irrelevant cell populations. As an example, instead of placing the forward gating arrow on the live cells (ZombiePB450-negative population), the arrow emanates from the dead cells. (3) The authors are also using the same arrow type to define sequential gating and seemingly to highlight group comparisons. The gating should be clarified so not to add unnecessary confusion to an otherwise outstanding story.

REVIEWERS' COMMENTS

Reviewer #1 (Remarks to the Author):

This was a very substantial revised manuscript and improved substantially. I congratulate the authors to their work!!! All my concerns are excellently addressed.

Response:

We would like to sincerely thank you once again for your constructive comments on our initial manuscript, which significantly improved the quality of our manuscript.

Reviewer #2 (Remarks to the Author):

Enthusiasm for this study remains high, which has potential for high impact across fields of skeletal biology, the bone marrow microenvironment, and endothelial cell biology. The revised manuscript and additional data represents a substantial improvement versus the first version. The authors have thoughtfully addressed our comments, and at this point, only a few 'minor' points remain.

Response: We appreciate your recognition and encouraging words on our work. Your comments are valuable and very helpful for revising and improving our paper. We have made revisions according to your comments.

Reviewer #2 comments 1: The labels in the legend below panel 1H still appear to be swapped.

Response: Thank you for pointing it out. We apologize for our oversight, and we have corrected the legend now.

Fig. 1. (h) Quantitative percentage of dendrites in contact with vessels or in noncontact with vessels among all the vessel-nearside dendrites as described in (g).

Reviewer #2 comments 2: In the new quantification in Figure 3H, please indicate standard deviation across biological replicates.

Response: We have added the standard deviation (17.27%) in our manuscript. See line 233-235 in the main text.

(Line 233-235) “Quantification of cortical bones revealed that $79.95\% \pm 17.27\%$ (Mean \pm SD) of TCV fractions contain mitochondria from osteocytes (Fig. 3h).”

Reviewer #2 comments 3: In Supplementary Figures 1C and 5A, ALP staining is performed to quantify osteoblasts. However, the cells indicated by yellow arrows are embedded osteocytes rather than surface osteoblasts. This quantification needs to be re-performed to appropriately quantify bone surface osteoblast number.

Response: Thank you for pointing out. We have re-performed the quantification of osteoblasts on the bone surface. The results showed the number of osteoblasts were decreased in $Dmp1^{Cre}-DTA^{ki/wt}$ mice and unchanged in $Dmp1^{Cre}-Rhot1^{fl/fl}$ mice.

Supplementary Fig. 1. (c-d) Representative ALP staining images of femur cortical bone (c) and quantitative result of ALP positive cell number (d) from 4-week-old control $DTA^{ki/wt}$ mice and $Dmp1^{Cre}-DTA^{ki/wt}$ mice. $n=3$, scale bars, 100 μ m, yellow arrows represent ALP positive cells.

Supplementary Fig. 5. (a-b) Representative ALP staining images of femur cortical bone (a) and quantitative result of ALP positive cell number (b) from 6-week-old control $Rhot1^{fl/fl}$ and $Dmp1^{Cre}-Rhot1^{fl/fl}$ mice. $n=3$, scale bars, 100 μ m, yellow arrows represent ALP positive cells.

Reviewer #2 comments 4: The new descriptions of osteocyte morphology in Rhot1 knockout mice in Supplementary Figure 5G-J are helpful but lack description of osteocyte dendrite morphology. Since the mechanism of mitochondrial transfer to endothelial cells likely involves dendrites (Figure 1), the length and number of osteocyte dendrites should be quantified in $Dmp1^{Cre};Rhot1^{fl/fl}$ mice compared to control.

Response: Thank you for your suggestions, and we have quantified the number and length of osteocyte dendrites in $Dmp1^{Cre}-Rhot1^{fl/fl}$ based on confocal images. Our results indicate that there were no significant alterations in the number and length of dendrites in the experimental group (Supplementary Fig. 5k), and we have updated this information in the manuscript (Line 307-308). (Line 307-308) “Consistently, the number and length of osteocyte dendrites in $Dmp1^{Cre}-Rhot1^{fl/fl}$ mice were found to be unchanged (Supplementary Fig. 5k).”

Supplementary Fig. 5. (k) Quantitative result of dendrites number and length per osteocyte from 6-week-old control $Rhot1^{fl/fl}$ and $Dmp1^{Cre}-Rhot1^{fl/fl}$ mice. $n=3$.

Reviewer #2 comments 5: The femur defect model, now shown in Figure 5 and in Figure 6, still lacks data to fully support the specific role of mitochondrial transfer in healing. Primarily, there is no histological evidence that endothelial cells near the bone defect are receiving osteocyte mitochondria, and effects of exogenous mitochondria and Sph on other bone cells are not ruled out. Further, full healing of the defect would be expected to take many weeks, so the endpoint of one week after injury (and just a few days after mitochondrial transfer / Sph administration) is not sufficient to demonstrate a meaningful effect on bone healing. We respectfully suggest that the

effect of Sph on bone healing be removed from the abstract given its unclear relationship with the main focus of the study.

Response: *We thank you for your excellent suggestion. The complexity of mitochondrial transfer in fracture healing is indeed far beyond our understanding, and we have removed this information in abstract as suggested.*

Reviewer #3 (Remarks to the Author):

The authors have a done a reasonable job responding to my concerns, but some minor issues remain. They can be addressed with changes to the text without any additional experimental work, so I feel a minor revision is warranted so the authors have an opportunity to improve the manuscript.

Response: *We appreciate your previous comments and supports, and we have updated the manuscript according to your suggestions.*

Reviewer #3 comments 1: I think the interpretation of the transwell experiment is problematic. First, as the authors point out, there is a substantial amount of mitochondria transfer (25%) across the transwell, which means that osteocytes do release cell-free mitochondria for transfer to endothelial cells. Second, the “lower efficiency” of mitochondria transfer with transwell separation could be explained by dilution of the released mitochondria into a relatively large volume of media. When the cells are cultured in the same chamber and in direct proximity of each other, the local concentration of released mitochondria is much higher, so transfer will be more efficient. Therefore the difference in mito transfer efficiency they are reporting could be explained entirely by a dilutional effect, which is an artifact of the system. Their conclusion is that the mito transfer is primarily contact-dependent, but I don’t think they have convincingly shown that.

As far as I can tell, the most defensible conclusion is that osteocytes deliver mitochondria to their dendritic endfeet in a MIRO-1-dependent manner and then transfer mitochondria to endothelial cells via unclear mechanisms that may involve release of mitochondria from the cell and/or direct cellular contact.

I do not expect the authors to distinguish between those mechanisms in this paper, but they should acknowledge alternative pathways without staking a claim on one particular mechanism.

Response: *We appreciate your point on the transwell experiment, which supports the idea that a non-contact manner can serve as an alternative path for mitochondrial transfer in vitro. we have updated the corresponding results according to your suggestions (Line 210-212). We also added the “dilution effect” as a limitation in our discussion (Line 525-527). In addition, we have updated the suggested conclusion in our manuscript (Line 489-491).*

(Line 210-212) “The confocal image analysis revealed that mitochondria transfer may also occur through non-contact transferring in the local microenvironment (Fig. 3d-e).”

(Line 489-491) “Our study suggested osteocytes deliver mitochondria to their dendritic endfeet in a MIRO-1-dependent manner and then transfer mitochondria to endothelial cells. However, the mechanisms on how mitochondria are released and entered to endothelial cells are not clear.”

(Line 525-527) “Our second limitation is that “lower efficiency” of mitochondria transfer in transwell coculture system could potentially be attributed to the dilution of released mitochondria in vitro.”

Reviewer #3 comments 2: The gating strategy in Supplemental Fig 4c (and in other gating examples they provide) is confusing for 3 reasons. (1) The authors have added gates that are not used for sequential gating, making it unclear which population of cells they are gating on. (2) The

gating arrows are positioned on what appear to be the irrelevant cell populations. As an example, instead of placing the forward gating arrow on the live cells (ZombiePB450-negative population), the arrow emanates from the dead cells. (3) The authors are also using the same arrow type to define sequential gating and seemingly to highlight group comparisons. The gating should be clarified so not to add unnecessary confusion to an otherwise outstanding story.

Response: We appreciate your suggestion. We have now revised our gating strategies according to your suggestions.

Supplementary Fig. 2. **(a)** Gate strategy to identify Dendra2⁺ bEnd.3 cells cocultured with MLO-Y4^{Mito-Dendra2} with number ratio at 1:1 and 3:1 (MLO-Y4: bEnd.3) for 24 hours, 48 hours and 72 hours and 7 days.

Supplementary Fig. 4. **(c)** Gate strategy to identify bEnd.3 endothelial cells acquired fluorescent mitochondria from NC-MLO-Y4^{Mito-Dendra2} or Rhot1^{KD}-MLO-Y4^{Mito-Dendra2} after 24 hours of coculture.

Supplementary Fig. 7. **(b)** Gate strategy to identify bEnd.3 endothelial cells for measurement of fluorescence intensity of DFCH-DA after transplantation of MLO-Y4^{Mito-Dendra2} mitochondria.

Supplementary Fig. 8. **(a-d)** The influence of antimycin A and rotenone (A/R) on ROS levels in bEnd.3 cells. **(b)** Gate strategy. **(e-g)** The effect of MLO-Y4 cells mitochondria on ROS level on healthy bEnd.3 cells. **(e)** Gate strategy.

Supplementary Fig. 9. **(f)** Gate strategy to identify bEnd.3 endothelial cells for measurement of fluorescence intensity of DFCH-DA after being treated with different concentrations of D-sphingosine. **(g)** Gate strategy to identify bEnd.3 endothelial cells acquired fluorescent mitochondria in 2D coculture system supplemented with 0nM, 12.5nM and 50nM D-sphingosine.